# Can Diffusion Models Disentangle?
# A Theoretical Perspective

**Liming Wang**[1]    **Muhammad Jehanzeb Mirza**[1]    **Yishu Gong**[2]    **Yuan Gong**[1][*]

**Jiaqi Zhang**[1]    **Brian H. Tracey**[2]    **Katerina Placek**[2]    **Marco Vilela**[2]

**James R. Glass**[1]
[1]Massachusetts Institute of Technology, [2]Takeda
`limingw@csail.mit.edu`

## Abstract

This paper presents a novel theoretical framework for understanding how diffusion models can learn disentangled representations with commonly used weak supervision such as partial labels and multiple views. Within this framework, we establish identifiability conditions for diffusion models to disentangle latent variable models with *stochastic*, *non-invertible* mixing processes. We also prove *finite-sample global convergence* for diffusion models to disentangle independent subspace models. To validate our theory, we conduct extensive disentanglement experiments on subspace recovery in latent subspace Gaussian mixture models, image colorization, denoising, and voice conversion for speech classification. Our experiments show that training strategies inspired by our theory, such as style guidance regularization, consistently enhance disentanglement performance.

## 1 Introduction

Extracting hidden structure from raw sensory data is fundamental to progress in multimodal perception [1–5], scientific discovery [6–15], AI-assisted content creation [16–20], and many more. Autonomous vehicles must localize objects and auditory events while suppressing background noise to navigate safely in open-world conditions, and data-driven drug-discovery systems need to group and recombine functionally related chemical components to propose therapies for emerging diseases. Deep learning-based creative tools likewise hinge on isolating user-specified factors (e.g., speaker style or lighting) while leaving other aspects untouched.

Many latent factors are *disentangled* in the sense that they vary independently. For example, speech content persists regardless of the speaker, and object shape remains consistent under different lighting conditions. This intuition inspires breakthroughs in linear [21] and non-linear [22] independent component analysis (ICA), modern deep learning-based disentanglement [23, 24] and causal representation learning [25]. However, an impossibility result shows that fully *unsupervised* disentanglement is unattainable in general [26]. Recent work on disentanglement therefore relies on weak labels [27, 28] or multi-view supervision [29, 30].

Applications such as video editing and drug discovery often require both latent factor extraction and controlled synthesis of novel samples from the latent factors. Diffusion models (DMs) [31–33] based on learning score functions [34] of probability distributions, excel at generation and power state-of-the-art

---

[*]Work done at MIT, now with xAI

39th Conference on Neural Information Processing Systems (NeurIPS 2025).

editors and simulators [16, 17, 19]. However, standard DMs learn only the data *marginal*, encoding latent factors *implicitly*. To control data generation using the latent structure, *conditional* DM (CDM) inject side information into the score function [35–37] and achieve impressive empirical success in disentanglement tasks such as voice conversion and image editing [38, 39, 20, 35, 36]. Yet a principled understanding of *when* these models learn identifiably disentangled representations is still missing.

In our work, we broaden the disentanglement theory to diffusion models and provide the first learning-theoretic framework for DM-based disentanglement, which poses unique challenges. First, a sample from DM is generated by integrating a stochastic differential equation (SDE), so the generated sample becomes a *stochastic, non-invertible* mapping of the latent variables. The lack of an analytic inverse precludes the change-of-variables calculus and ICA-style arguments that underpin classical disentanglement theory [26, 40]. Further, the extra uncertainty injected by the stochastic drift can make it more difficult to leverage the commonly used weak supervision. We tackle these challenges by (i) proposing notions of approximate disentanglement applicable to stochastic, non-invertible settings, (ii) marrying information-regularised score matching with recent finite-sample analyses of score-based models [41]. The resulting framework lays the groundwork for the following contributions:

1. We show that, under mild Lipschitz assumptions, DMs can recover approximately disentangled representations of two latent factors (e.g., content and style), when given either partial supervision or multi-view inputs. For independent subspace models, we further prove global convergence in the finite-sample regime using gradient descent.

2. Building on the theory, we introduce a novel style-guided score matching loss that attains a global optimum for the independent-subspace case and improves disentanglement in practice.

3. Extensive experiments on Gaussian mixture models, image editing, and voice conversion for speech classification demonstrate that our theory-inspired training strategies consistently enhance disentanglement quality and downstream classification accuracy.

The rest of this paper is organized as follows: In Section 2, we provide the background on diffusion models. Section 3 formalizes the problem of content-style disentanglement, and later in Section 4 we present the main theoretical results and Section 5 details empirical evaluations on synthetic, image and speech data, demonstrating and supporting the theoretical findings. Finally, Section 7 concludes the paper.

## 2 Background: diffusion models

Diffusion models (DMs) [31–33] approximate the pdf $p_X =: p_0$ of an r.v. $X \sim p_0$ via a two-stage process: *noising* and *denoising*. In the noising stage, data is progressively corrupted using an SDE:

$$\mathrm{d}X_t = \mu(X_t, t)\mathrm{d}t + \xi(t)\mathrm{d}B_t, \quad X_0 \sim p_0, \tag{1}$$

where $B_t$ is a Brownian motion. We adopt the choices $\mu(X_t, t) := -X_t$ and $\xi(t) \equiv \sqrt{2}$, leading to the Ornstein–Uhlenbeck (OU) process:

$$\mathrm{d}X_t = -X_t \mathrm{d}t + \sqrt{2}\mathrm{d}B_t, \quad X_0 \sim p_0, \tag{2}$$

which converges to a standard Gaussian [42, 41]. Let $p_t$ denote the marginal of $X_t$ and $p_{t|s}$ the conditional pdf of $X_t$ given $X_s$. In the denoising stage, the goal is to recover $X$ from noisy versions $X_t, t \geq t_0$ by simulating the time-reversed process $X_t^{\leftarrow} := X_{T-t}$:

$$\mathrm{d}X_t^{\leftarrow} = [X_t^{\leftarrow} + 2\nabla_x \log p_{T-t}(X_t^{\leftarrow})]\mathrm{d}t + \sqrt{2}\mathrm{d}B_t^{\leftarrow}, \quad X_0^{\leftarrow} \sim p_T, \tag{3}$$

which converges back to $p_0$ [43]. Since the score function $\nabla_x \log p_t(X_t)$ is unknown, DM learns a *score estimator* $s_\theta : \mathbb{R}^{d_X} \times [0,T] \mapsto \mathbb{R}^{d_X}$ by minimizing the *score matching* objective:

$$L(\theta) := \mathbb{E}_{t, p_t} \| s_\theta(X_t, t) - \nabla_x \log p_t(X_t) \|^2,$$

This objective is equivalent to a *conditional* score matching objective involving $p_{t|0}$:

$$L_c(\theta) := \mathbb{E}_{t, p_0 p_{t|0}} \left\| s_\theta(X_t, t) - \nabla_x \log p_{t|0}(X_t | X) \right\|^2 = \mathbb{E}_{t, p_0 p_{t|0}} \left\| s_\theta(X_t, t) + \frac{N_t}{\sigma(t)} \right\|^2, \tag{4}$$

where $N_t$ is standard Gaussian and $\sigma(t) := \sqrt{1 - \exp(-2t)}$. During inference, new samples are generated by simulating an estimated SDE, with $\hat{X}_0^{\leftarrow} \sim p_T$ and discretized time steps $t \in [k\eta, (k+1)\eta]$:

$$\mathrm{d}\hat{X}_t^{\leftarrow} = [\hat{X}_t^{\leftarrow} + 2s_\theta(\hat{X}_{k\eta}^{\leftarrow}, T - k\eta)]\mathrm{d}t + \sqrt{2}\mathrm{d}B_t^{\leftarrow}. \tag{5}$$

# 3 Content-style disentanglement

The task of content-style disentanglement can be formalized through a *latent variable model* (LVM). For clarity, we refer to all random entities – scalars, vectors, or matrices – as random variables (r.v.'s), and focus on the *continuous* case. For any r.v. $X$, let $p_X$ denote its probability density function (pdf). We assume two latent factors: the *content* $Z \sim p_Z$ taking values in $\mathbb{R}^{d_Z}$ and the *style* $G \sim p_G$ taking values in $\mathbb{R}^{d_G}$, which jointly generate an observable *sample* $X \in \mathbb{R}^{d_X}$ through the noisy, non-invertible and nonlinear transformation with an invertible *mixing function* $f : \mathbb{R}^{d_Z} \times \mathbb{R}^{d_G} \mapsto \mathbb{R}^{d_X}$:

$$X = \sqrt{1-\delta^2} f(Z,G) + \delta N, \tag{6}$$

where $\delta$ is the noise level and $N$ is an independent, standard Gaussian noise. We assume that $Z$ and $G$ are statistically independent, as is common in disentanglement [24, 44, 26] and ICA literature [21, 40]. Although exact independence between $Z$ and $G$ is assumed for clarity, our framework naturally extends to settings where independence holds only approximately, as will be discussed subsequently.

This partition of latent factors arises in several settings. In *controllable generation*, $Z$ encodes persistent attributes (e.g., object identity) while $G$ governs editable factors (e.g., pose or lighting). In *self-supervised learning* (SSL), $Z$ captures invariant content across views or modalities, and $G$ captures modality- or augmentation-specific variations. In the low-noise regime $\delta \to 0$, our goal is to recover the latent variables $Z$ and $G$ from observations of $X$. A common notion of recovery is *block identifiability* [45–48, 29, 30], which ensures that subgroups of scalar latent variables can be recovered up to an invertible transformation. However, exact block identifiability is not achievable in the presence of noise $\delta > 0$ due to the non-invertibility of the mixing process, motivating a need for *approximate disentanglement*. To this end, we propose two complementary criteria: (1) approximate information-theoretic disentanglement and (2) editability.

**$(\epsilon, \nu)$-disentanglement.** To quantify how well the learned content and style representations are separated and informative, we define an information-theoretic notion of approximate disentanglement.

**Definition 3.1** (($(\epsilon,\nu)$-disentanglement)**.** *Let $(\hat{Z}, \hat{G})$ be content and style encodings inferred from an observed sample $X$. They are $(\epsilon,\nu)$-disentangled if, for some $\epsilon, \nu \geq 0$, (i) $I(\hat{Z}; \hat{G}) \leq \epsilon$; (ii) $I(\hat{Z}, \hat{G}; X) \geq I(Z,G;X) - \nu$, where $I(A;B)$ denotes the mutual information (MI) between r.v.'s $A$ and $B$.*

These conditions ensure that the learned latent factors are (i) nearly independent (as the true $Z$ and $G$ are), and (ii) retain most of the information about the observed data $X$. The definition remains meaningful even as $\delta \to 0$ and $I(Z,G;X) \to \infty$, since it is based on a bounded *difference* in mutual information.

**$\epsilon$-editability.** In many applications, it is desirable to modify style while preserving content. This motivates the following notion of editability based on conditional sample generation.

**Definition 3.2.** *($\epsilon$-editability) Let $(\hat{Z}, \hat{G})$ be encodings inferred from $X$, and let $\hat{G}' \sim p_{\hat{G}}$ be an independent style encoding. $(\hat{Z}, \hat{G})$ are $\epsilon$-editable if there exists a generative model $q(\cdot \mid \hat{Z}, \hat{G}')$ such that the generated sample $\hat{X} \sim q(\cdot \mid \hat{Z}, \hat{G}')$ satisfies $\mathbb{E}_{Z \sim p_Z} d_{\mathrm{TV}} \left( p_{\tilde{X}|Z}, p_{\hat{X}|Z} \right) \leq \epsilon$, where $\tilde{X} := \sqrt{1-\epsilon^2} X + \epsilon N$ is a* smoothed *version of $X$, and $d_{\mathrm{TV}}$ is the total variation distance.*

This definition captures the ability to recombine content and style encodings to generate new samples that are consistent with the original content. For example, in a facial image editing task, the encoding $\hat{Z}$ may represent identity while $\hat{G}$ captures facial expression. By swapping $\hat{G}$ with a new expression encoding $\hat{G}'$, we can generate a new image that preserves identity but alters the expression. Deterministic decoders are allowed as a special case with $q(\cdot \mid \hat{Z}, \hat{G}) = \delta_{\hat{f}(\hat{Z}, \hat{G})}$ for some function $\hat{f}$.

The notions of $(\epsilon, \nu)$-disentanglement and $\epsilon'$-editability are complimentary but not equivalent. In particular, disentanglement in the mutual information sense does not guarantee editability. The following example illustrates this by constructing encodings that are perfectly disentangled (i.e., $(0,0)$-disentangled) yet fail the editability criterion due to a hidden ambiguity introduced during recombination. The proof is provided in Appendix A.

**Example 3.1.** *Suppose content $Z \sim \mathcal{N}(0,1)$ and $G \sim \mathcal{N}(0,1)$ are independent standard Gaussian r.v.'s and consider the noiseless setting with $\delta = 0$. Further, suppose $p_{f(Z,G)|Z} \not\equiv p_{f(-Z,G)|Z}$. Then we can choose the content/style encodings to be $\hat{Z} = Z\,sgn(G)$, $\hat{G} = G$ and the decoder $\hat{f}(\hat{Z},\hat{G}) = f(\hat{Z}sgn(\hat{G}),\hat{G})$, where $sgn(x)$ denotes the sign of $x$. Then $(\hat{Z},\hat{G})$ are (0,0)-disentangled but not $\epsilon$-editable for some $\epsilon > 0$.*

Intuitively, the encoder flips the sign of the content variable depending on the style. This transformation preserves both independence and informativeness, but causes ambiguity when recombining $\hat{Z}$ with a new style $\hat{G}'$, since the decoder cannot distinguish whether the original sign should be restored. As a result, samples generated from new combinations do not match the original content distribution.

**Weakly supervised disentanglement.**    When neither $Z$ nor $G$ is observed, disentanglement is in general impossible for nonlinear mixing functions: the observable pdf $p_X$ alone does not identify whether the data is generated from a disentangled or entangled LVM [26]. We consider two practical settings where *side information* is available to help resolve this ambiguity.

**Definition 3.3** (Content disentanglement). *Assume Eq. 6, and suppose a known* style function *$g : \mathbb{R}^{d_X} \mapsto \mathbb{R}^{d_G}$ is given such that $g(f(z,G)) = G$ for all $z$. The goal of* content disentanglement *is to learn encodings $(\hat{Z},\hat{G})$ from $X$ that are $(\epsilon,\nu)$-disentangled and $\epsilon'$-editable.*

This setting appears in applications such as image editing and voice conversion [49, 38, 50]. For example, in voice conversion, $g(X)$ is a speaker embedding extracted from a pre-trained speaker recognition model (e.g. speaker embeddings from ECAPA-TDNN, which achieves almost perfect speaker verification error rate [51]). This assumption mirrors common practice in the voice conversion literature and is not unique to our framework. In image editing, $g(\cdot)$ could represent learned text embeddings of editing instructions.

**Definition 3.4** (Multi-view disentanglement). *Assume there are multiple views $X^1,\cdots,X^{n_V}$, where each $X^i$ is generated as $X^i = \sqrt{1-\delta_i^2} f_i(Z,G^i) + \delta_i N^i, 1 \leq i \leq n_V$, with i.i.d view-specific styles $G^i$'s, i.i.d standard Gaussian noise $N^i$'s, view-specific noise levels $\delta_i$'s and invertible view-specific mixing functions $f_i : \mathbb{R}^{d_{G_i}} \times \mathbb{R}^{d_{Z_i}} \mapsto \mathbb{R}^{d_{X_i}}$. Then the task of* multi-view disentanglement *is to learn encodings $(\hat{Z},\hat{G}^i)$ for all $i$ such that $(\hat{Z},\hat{G}^i)$ are $(\epsilon,\nu)$-disentangled and $\epsilon'$-editable.*

This setting is prevalent in SSL (e.g., [52–58]). When $f_i \equiv f_1$ (e.g. multiple camera views), each $X^i$ may correspond to a different augmentation. When $f_i$ differ (e.g., across sensory modalities), the views may represent distinct but semantically aligned representations. For clarity, we focus on the unimodal two-view case ($n_V = 2$), which readily generalizes to multimodal scenarios with $n_V > 2$. To facilitate our theoretical analysis of DM-based disentanglement, we adopt the following mild assumptions, which are common in the analysis of DMs [59, 42, 41].

**Assumption 3.5.** *The sample $X$ is sub-gaussian with second moment $\sigma_X^2 d_X$.*

**Assumption 3.6.** *The score function of the sample pdf $p_X$ is $\lambda_s$-Lipschitz.*

Assumption 3.5 ensures that sample values do not exhibit heavy tails, which could destabilize the diffusion process. Assumption 3.6 ensures that the score function does not change too abruptly, preventing discontinuities that could hinder accurate recovery of content and style during denoising.

# 4   Theory: diffusion model-based disentanglement

In this section, we first present our theoretical results for DM-based disentanglement, and then discuss multi-view disentanglement, and finally provide results for disentanglement of independent subspaces.

## 4.1   Content disentanglement with diffusion models

**Overview.**    This section analyzes the ability of DMs to achieve approximate disentanglement in the content disentanglement task defined in Definition 3.3. First, we introduce a conditional DM trained with a regularized score matching objective tailored for this task. Second, we show that this model can learn content and style encodings $(\hat{Z},\hat{G})$ that are $(\epsilon,\nu)$-disentangled with arbitrarily small $\epsilon,\nu$ as the noise level $\delta \to 0$. Finally, we discuss why such encodings may still fail to be $\epsilon'$-editable for some $\epsilon' > 0$. Formal proofs are provided in Appendix B.

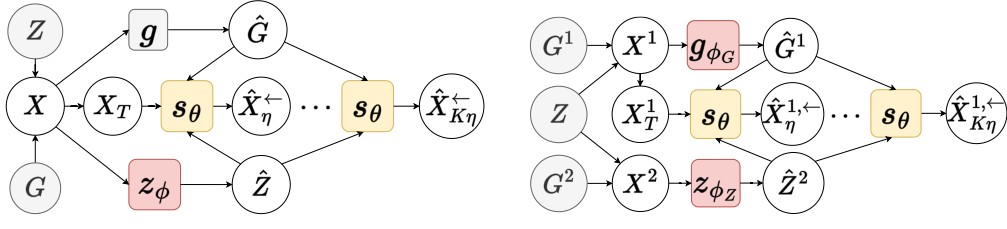

(a) Content disentanglement        (b) Multi-view disentanglement ($n_V = 2$)

Figure 1: **Graphical model of diffusion-based disentanglement under different types of weak supervision**. Shaded nodes denote latent variables; clear nodes denote observed variables. Learnable components are marked in red and orange, while the frozen component is shown in gray. **Left: Content disentanglement**. The style encoder $g(\cdot)$ is known and fixed. The model learns a content encoder $z_\phi$ and a score estimator $s_\theta$ to estimate the score of $p_{X|\hat{Z},\hat{G}}$, thereby disentangle content $Z$ from style. **Right: Multi-view disentanglement**. Given paired views $(X^1, X^2)$ sharing content $Z$, the model learns to recover $Z$ and style $G^1$ of view 1 by estimating the score of $p_{X|\hat{Z}^2,\hat{G}^1}$ using $s_\theta$.

The model architecture is illustrated in Figure 1a. Style is represented by a fixed encoder $\hat{G} := g(X_{t_0})$, where $g(\cdot)$ is assumed known. Content is learned through a trainable encoder yielding $\hat{Z} = z_\phi(X_{t_0})$. To enable controllable generation, both $\hat{Z}$ and $G$ are fed as inputs to a conditional score estimator $s_\theta : \mathbb{R}^{d_X} \times \mathbb{R}^{d_Z} \times \mathbb{R}^{d_G} \times [0,T] \mapsto \mathbb{R}^{d_X}$, which estimates the score of the *conditional* pdf $p_{X|\hat{Z},\hat{G}}$. We train this model using the following *regularized* score matching objective: hyperparameters $\gamma, \rho > 0$:

$$L_c^{\gamma,\rho}(\theta,\phi) := \underbrace{\mathbb{E}_{t,p_0 p_{t_0|0} p_{t|t_0}} \left\| s_\theta(X_t, \hat{Z}, \hat{G}, t) + \frac{N_t}{\sigma(t)} \right\|^2}_{\text{Conditional score matching loss}} + \underbrace{\gamma (I(z_\phi(X_{t_0}); X) - \rho)_+}_{\text{Content information regularizer}}, \qquad (7)$$

where $(x)_+ := \max\{x, 0\}$. The score matching loss encourages the model to match the conditional score, while the regularizer limits the information that the content encoder can extract from the input, preventing overfitting via direct copying. In practice, mutual information is often approximated using tractable variational bounds [60–62], which serve as surrogates for the content information regularizer. Using the above steup, we prove the following theorem.

**Theorem 4.1.** *Suppose Assumption 3.5-3.6 hold, and $(\theta^*, \phi^*)$ be a minimizer of $L_c^{\gamma,\rho}$ defined in Eq. 7 with $\rho = I(Z;X) + C_1 \delta, \gamma \geq C_2/T$ for some $C_1, C_2 > 0$. Set $t_0 = -\log(1-\delta^2)^{1/2}$ and $\hat{Z} := z_{\phi^*}(X)$. Then for any $\delta < \min\left\{ \frac{1}{2}, \frac{1}{\sqrt{d_X}} \right\}$, the encodings $(\hat{Z}, \hat{G}) = (z_{\phi^*}(X_{t_0}), g(X_{t_0}))$ are $(\epsilon, \nu)$-disentangled with $\epsilon = C_3 \lambda_s \sigma_X^2 d_X \delta, \nu = C_4 \sigma_X^2 \delta^2$ for some constants $C_3, C_4 > 0$.*

> **Intuition.** Theorem 4.1 shows that under sub-gaussian tail assumptions and Lipschitz-continuous scores, DM can achieve $(\epsilon, \nu)$-disentanglement with $(\epsilon, \nu) \to 0$ as the noise level $\delta \to 0$. The rate at which disentanglement improves depends inversely on the Lipschitz constant — higher sensitivity in the score function slows disentanglement by amplifying noise-induced variations. The regularized objective mitigates this trading off between predictive power and the amount of content information retained in the content encoding $\hat{Z}$.

Theorem 4.1 extends to cases where content and style are only approximately independent, i.e., $I(Z;G) \leq \epsilon_1$ for some $\epsilon_1 > 0$, by treating the dependency as a perturbation (Appendix B.7). Assumption 3.5 can also be relaxed to bounded variances, as the proof relies on moment control. However, it is important to note that approximate disentanglement in the MI sense does not imply editability. As illustrated in Example 3.1, the model may still leak style information into the content encoding. This limitation, which we refer to as *content distortion*, prevents achieving vanishing $\epsilon'$-editability even as $\delta \to 0$.

## 4.2   Multi-view disentanglement with diffusion models

**Overview.** This section analyzes the ability of DMs to achieve editability in the multi-view disentanglement setting defined in Definition 3.4. We introduce a DM trained with a modified score matching objective, analogous to the content disentanglement case. We then show that the learned

encodings $(\hat{Z},\hat{G})$ are $\epsilon$-editable with $\epsilon \to 0$ as $\delta \to 0$. Finally, we discuss how the result generalizes to more than two views and non i.i.d styles, and what it reveals about the role of different types of weak supervision. Full proofs are in Appendix C.

The model architecture is illustrated in Figure 1b. We define the encoders $\hat{Z}^i := z_{\phi_Z}(X^i_{t_0})$ and $\hat{G}^i := g_{\phi_G}(X^i_{t_0})$ for each view $i$. The score estimator $s_\theta(X_t, \hat{Z}^2, \hat{G}^1, t)$ is trained with the following loss:

$$L_m(\theta,\phi) := \underbrace{\mathbb{E}_{t,p_0 p_{t_0|0} p_{t|0}} \left\| s_\theta(X^1_t, \hat{Z}^2, \hat{G}^1, t) + \frac{N^1_t}{\sigma(t)} \right\|^2}_{\text{Conditional score matching loss}} + \underbrace{I(\hat{G}^1; X^2)}_{\text{Style information regularizer}}, \tag{8}$$

where $\phi = [\phi_Z, \phi_G]$. The regularizer encourages $\hat{G}^1$ to encode only the style of $X^1$. At inference time, we sample from the estimated reverse process Eq. 5 but use the *conditional* score estimator to guide generation. Under this setup, We prove the following theorem.

**Theorem 4.2.** *Under Assumption 3.5-3.6 and additional regularity conditions (Assumption C.2-C.4). Consider the two-view, unimodal setting with $\delta_1 =: \delta$ and $\delta_2 = 0$. Let $(\theta^*, \phi^*_Z, \phi^*_G)$ be a minimizer of $L_m$ and $\hat{Z}^i = z_{\phi^*_Z}(X^i), \hat{G}^i = g_{\phi^*_G}(X^i)$ for each view $i$. Set $t_0 = -\log(1-\delta^2)^{1/2}$ and the diffusion step size $\eta := C_5 \frac{\delta^3}{\lambda^2_{\hat{\Theta}} T}$ for some constant $C_5 > 0$. Then for some constants $C_6, C_7 > 0$: (i) $I(\hat{Z}^2; \hat{G}^1) \leq C_6 \lambda_s \sigma_X d_x \delta$; (ii) the content encoding $\hat{Z}^2$ and the style encoding $\hat{G}^1$ are $C_7 \sqrt{\lambda_s \sigma^2_X d_X \delta}$-editable.*

**Intuition.** Theorem 4.2 establishes that in the multi-view setting, DMs can achieve $(\epsilon, \nu)$-disentanglement and $\epsilon$-editability with vanishingly small $\epsilon$, even though $\nu$ cannot be made arbitrarily small. In other words, perfect separation of content and style is theoretically achievable in terms of independence and editability, even if some information loss is unavoidable. The editability error depends on the Lipschitz constant of the decoder, the intrinsic data variance, and the latent dimensionality—factors that all amplify sensitivity to noise. To mitigate this, we introduce a style information regularizer, which suppresses content leakage into the style encoder. This prevents entanglement and enables reliable "mix-and-match" generation. Without this regularization, residual content in the style embedding can undermine both disentanglement and editing performance.

While Theorem 4.2 is stated for i.i.d. styles across views, it readily extends to nearly independent styles, i.e., those satisfying $I(G^1; G^2 \mid Z) \leq \epsilon_I$ for some small $\epsilon_I$ (see Appendix C.4). Our framework also handles more than two views and heterogeneous view-specific mixing functions. A key challenge arises when content and style are highly correlated or when the sample size is small. In these settings, noise can break delicate statistical dependencies, leading to spurious entanglement, especially under limited data, which can mask the true latent structure further due to overfitting. These issues are not covered by prior identifiability results [48, 29], which assume noiseless generation and invertible mixing functions.

### 4.3 Disentanglement of independent subspaces with diffusion models

**Overview.** This section considers a *special case* of content disentanglement – when the underlying distribution follows an *independent subspace model* (ISM) [45–47, 63]. ISMs have served as a foundation in classical disentanglement literature and are also supported empirically in recent self-supervised learning studies [64], with applications in both discriminative and generative disentanglement tasks [65, 66, 39, 50, 67]. Under the ISM setting, we prove that a carefully constructed DM trained with gradient descent can simultaneously achieve $(\epsilon, \nu)$-disentanglement and $\epsilon'$-editability even in the finite-sample setting – offering

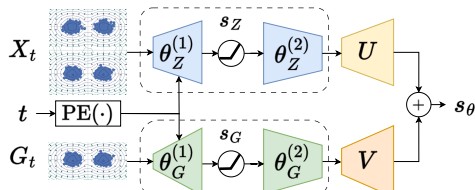

Figure 2: **Dual-encoder score network** for content-style disentanglement of ISM.

stronger theoretical guarantees than those in the general content disentanglement case (Section 4.1). We also outline how these results can be extended to the multi-view case in the discussion. Full proofs are provided in Appendix D. To begin, we formally define an ISM below.

**Definition 4.3.** *An* independent subspace model *(ISM) is an LVM defined by*

$$Z \sim p_Z, G \sim p_G, X = A_Z Z + A_G G, \tag{9}$$

where $A_Z \in \mathbb{R}^{d_X \times d_Z}$, $A_G \in \mathbb{R}^{d_X \times d_G}$ are orthogonal matrices such that their column spaces are orthogonal and span $\mathbb{R}^{d_X}$. That is, $R(A_Z)^\perp = R(A_G)$ with $d_Z + d_G = d_X$, where $R(A)$ is the column space of matrix $A$. Let $X_t$ denote the noisy version of $X$ at time $t$ in the diffusion process. Then we define $Z_t := A_Z^\top X_t =: z(X_t)$, $G_t := A_G^\top X_t =: g(X_t)$.

This model generalizes the setting in [41], which assumes Gaussian noise for $G_t$. A key property of ISMs is that the score function of the marginal $p_t(X_t)$ is *decomposable* as shown below.

**Lemma 4.4.** *For any $t \geq 0$, the score of the pdf $p_t(X_t)$ under the ISM satisfies*

$$s^*(X_t, t) := \nabla_x \log p_t(X_t) = A_Z \nabla_z \log p_{Z_t}(z(X_t)) + A_G \nabla_g \log p_{G_t}(g(X_t)).$$

Lemma 4.4 shows that for ISM, the score function is a linear combination of the score functions of content and style. Motivated by Lemma 4.4, we propose a *dual encoder network* for learning $s^*(x,t)$, as shown in Figure 2:

$$s_\theta(x,t) := U s_Z^{\theta_Z}(x,t) + V s_G^{\theta_G}(g(x),t) := U \texttt{NN}([x, \texttt{PE}(t)]) + V \texttt{NN}([g(x), \texttt{PE}(t)]). \tag{10}$$

where $\texttt{NN}(\cdot)$ denotes a two-layer ReLU neural net and $\texttt{PE}(t)$ is a time position encoding. The first branch of the dual network computes the content score $s_Z(X_t, t)$ of the noisy sample $X_t$, while the second branch computes the style score $s_G(G_t, t)$ from the noisy style $G_t$. The two scores are combined to produce the final score function $s_\theta(X_t, t)$. Unlike earlier sections, $s_\theta$ here is *unconditional*. To train this network, we use a regularized score matching loss:

$$L_n^{\lambda_r}(\theta) = 2 \underbrace{\mathbb{E}_{t, \hat{p}_t^n(x)} \| s_\theta(x,t) - s^*(x,t) \|_2^2}_{L_{0,n}: \text{ score matching loss}} + 2\lambda_r \underbrace{\mathbb{E}_{t, \hat{p}_t^n(x)} \| V s_G(g(x),t) - s^*(x,t) \|_2^2}_{L_{r,n}: \text{ style guidance regularizer}} + \underbrace{\frac{1}{2} L_{b,n}(\theta)}_{\text{Balancing loss}}, \tag{11}$$

where $\lambda_r > 0$ is the style guidance weight and $\hat{p}_t^n$ is the empirical pdf. The style guidance regularizer encourages style separation by **forcing the score network to explain part of the data using only style**, thereby implicitly **discouraging it from encoding style information redundantly in the content branch**; The balancing loss, defined in Appendix D, helps prevent poor local minima. For general latent variable models, this is a heuristic regularization rather than a formal MI minimization, and we will add a clearer explanation and acknowledge. The following theorem analyzes training dynamics by studying the *gradient flow*:

$$[\dot{U}, \dot{V}] = [-\nabla_U L_n^{\lambda_r}(\theta), -\nabla_V L_n^{\lambda_r}(\theta)], \quad [\dot{\theta}_Z, \dot{\theta}_G] = [-\nabla_{\theta_Z} L_n^{\lambda_r}(\theta), -\nabla_{\theta_G} L_n^{\lambda_r}(\theta)]. \tag{12}$$

**Theorem 4.5.** *Under Assumption 3.5-3.6, for some $t_0$ dependent on $n$ and let $\min\{d_T, d_H\} \to \infty$, and let the positional encoding $\texttt{PE}(\cdot)$ be bounded and linearly independent over $t \in [t_0, T]$. Define $P_M$ to be the projection matrix onto $R(M)$, and $\sigma_i(s)$ to be the $i$-th largest singular value of the operator $s$. Then for some $\lambda_r$, Eq. 12 converges to a critical point $\hat{\theta} := [\hat{\theta}_Z, \hat{\theta}_G, \hat{U}, \hat{V}]$ such that with probability at least $1 - O(\frac{1}{n})$: 1) Encodings $\hat{Z} := P_{\hat{U}} X$ and $\hat{G} := g(X)$ are $\left( O\left( \frac{d_X^{5/4} \log^{3/4} n}{\sigma_{d_Z}(s_Z^*) n^{1/4}} \right), 0 \right)$-disentangled; 2) Let $\hat{Z}_{t_0} := \hat{Z} + \sigma(t_0) N_{t_0}$. Then $(\hat{Z}_{t_0}, \hat{G})$ are $O\left( \frac{d_X^{7/4} \log^{9/16} n}{T \min\{\sigma_{d_Z}^{1/2}(s_Z^*), \sigma_{d_G}^{1/2}(s_G^*)\} n^{1/16}} \right)$-editable.*

> **Intuition.** Theorem 4.5 shows that for ISMs, a DM trained with gradient descent can recover the content and style subspaces as the number of samples $n \to \infty$. This enables both approximate $(\epsilon, \nu)$-disentanglement and $\epsilon'$-editability. A key component is the style guidance regularizer, which encourages the model to separate content and style by encouraging the model to use the style information during score matching. Importantly, the speed and quality of subspace recovery depend on the strength of the signal in the score function. In particular, recovery is faster when the content and style score functions have larger minimum singular value. This highlights a practical consideration: well-separated or high-contrast latent factors lead to more reliable disentanglement, while near-degenerate cases may require additional regularization or supervision.

Theorem 4.5 extends to the *multi-view* setting using the dual encoder network $s_{\theta,\phi}(X_t^1, t) = U s_Z(X_t^2, t) + V s_G(X_t^1, t)$, where $s_Z$ extracts content from the second view. In this case, we use a *content guidance loss* $L'_{r,n}$ applied to $(U, s_Z)$ instead of $(V, s_G)$, to encourage reliance on content information from the second view and suppress residual content in the style encoder $s_G$. Our bound depends on the data dimension $d_X$, but extensions to lower-dimensional latent representations are

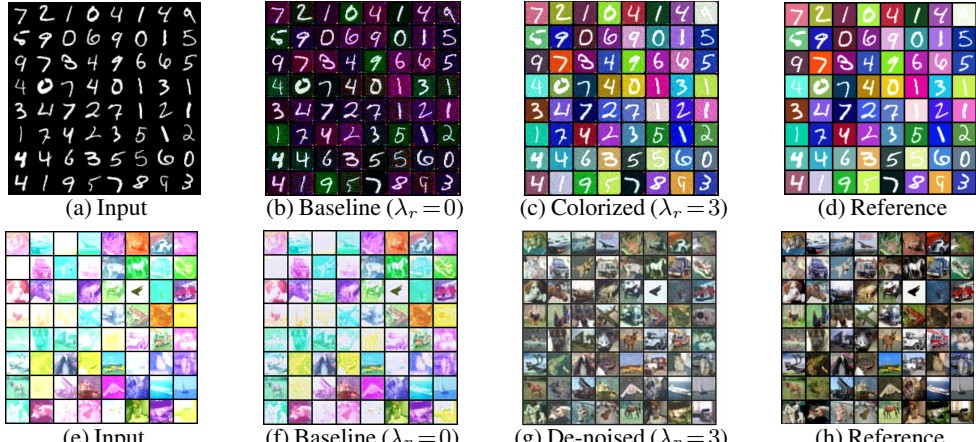

| (a) Input | (b) Baseline ($\lambda_r = 0$) | (c) Colorized ($\lambda_r = 3$) | (d) Reference |
| (e) Input | (f) Baseline ($\lambda_r = 0$) | (g) De-noised ($\lambda_r = 3$) | (h) Reference |

Figure 3: **Disentanglement results on MNIST and CIFAR-10**. Top: **Disentanglement results on MNIST**. The content is the gray-scale digit image, and the style is the background color. Bottom: **Disentanglement results on CIFAR10**. The content is the clean image and the style is the corruption on the image. Both 3b-3c and 3f-3g suggest disentanglement is achieved with the style guidance loss.

Table 1: **Quantitative results for image denoising and face editing on CelebA test sets for various models.** Conditional (cond.) InfoGAN and cond. $\beta$-VAE stand for InfoGAN [69] and $\beta$-VAE [70] adapted to the content disentanglement setting by conditioning on the observed style variable. DDPM stands for denoising diffusion probabilistic model (DDPM) [33].

|  | Denoising | | Face Editing | |
| --- | --- | --- | --- | --- |
|  | LPIPS($\downarrow$) | SSIM($\uparrow$) | CLIP Score($\uparrow$) | LPIPS($\downarrow$) |
| cond. InfoGAN | 0.83 | 0.16 | 0.24 | 0.39 |
| cond. $\beta$-VAE | 0.57 | 0.17 | 0.25 | 0.37 |
| DDPM (ours), $\lambda_r = 0$ | 0.53 | 0.18 | 0.25 | 0.35 |
| DDPM (ours), $\lambda_r = 0.03$ | 0.47 | 0.19 | 0.26 | **0.30** |
| DDPM (ours), $\lambda_r = 0.3$ | 0.35 | 0.23 | 0.26 | 0.32 |
| DDPM (ours), $\lambda_r = 3$ | **0.33** | **0.24** | **0.26** | 0.38 |

possible via residual connections in the score network in Figure 2. While our focus is on unconditional score matching, our results extend to conditional settings with appropriate corrections for cross terms. Lastly, although our analysis assumes infinite-width two-layer ReLU networks, similar convergence behavior may hold for deeper or finite-width networks [68].

# 5 Experiments

This section presents empirical evaluation of our theoretical framework by testing whether DMs can achieve approximate disentanglement under settings in Section 4.1-4.3. We begin with synthetic datasets generated from Gaussian mixture models (GMM), which instantiate the ISM framework from Section 4.3 to validate guarantees in Theorem 4.5. We then move to more realistic settings using standard image datasets. Specifically, we apply our DM-based disentanglement method to two tasks: image colorization on MNIST [71] and image denoising on CIFAR10 [72]. Finally, we validate Theorem 4.1-4.2 on a real-world speech task, voice conversion (VC) adaptation, by adopting the DM-based VC framework from [38].

**Implementation details.** For the GMM dataset, we use a two-layer ReLU network consistent with Theorem 4.5. For MNIST and CIFAR, we use a U-Net [73] architecture, following common DM design [74]. We optimize a regularized score-matching objective, inspired by Eq. 11:

$$L_c^{\lambda_r}(\theta, \phi) := \mathbb{E}_{t, p_0 p_{t_0|0} p_{t|t_0}} \left\| s_\theta(X_t, \hat{Z}, G, t) + \frac{N_t}{\sigma(t)} \right\|^2 + \lambda_r \mathbb{E}_{t, p_t} \left\| s_\theta(X_t, \mathbf{0}_{d_Z}, G, t) + \frac{N_t}{\sigma(t)} \right\|^2, \quad (13)$$

where $\lambda_r$ controls the style guidance weight. Note that we omit the balancing term for empirical simplicity, and style guidance is applied by zeroing out content, equivalent to the style regularizer in Eq. 11 when the input data has ISM structure.

**GMM disentanglement results.** First, we conduct subspace recovery experiments on *latent subspace GMM*s (LS-GMM), a class of LSMs where each subspace follows a GMM. Figure 4 shows the subspace recovery error as a function of the style guidance weight $\lambda_r$ and the sample size $n$. Consistent with the predictions of Theorem 4.5, the LSGMM achieves the smallest subspace reconstruction error when the style guidance weight is sufficiently large, and the result is consistent across different noise schedulers. Moreover, since all score networks are wide, two-layer MLPs trained using gradient-based methods, these results provide further support for Theorem 4.5.

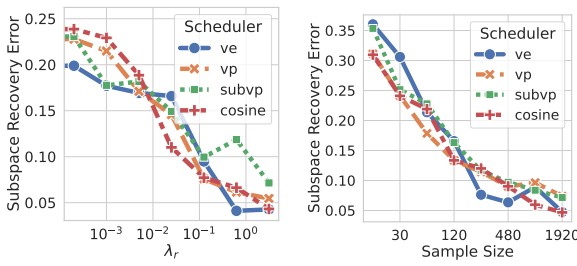

Figure 4: **GMM disentanglement results with the score estimator in Eq. 10**. The subspace recovery error (defined in Appendix E.1) is normalized between [0,1]. In four random trials, DM consistently recovers (error $< 0.1$) the correct content subspace and achieves disentanglement with sufficiently large style guidance weight $\lambda_r$ and sample size.

Figure 4 also reveals a sublinear decay rate of the subspace recovery error as the sample size increases, aligning with Theorem 4.5.

**Image disentanglement results.** Next, we validate our theoretical findings on image data from MNIST and CIFAR-10. For MNIST, we perform *colorization*, treating digit shape as content and background color as style. The results are visualized in Figure 3. Without regularization, the model fails to disentangle these factors, simply copying the input (Figure 3b). With the proposed regularizer, it successfully disentangles color from shape (Figure 3c). On CIFAR10, we test image *denoising* where the content is the clean image and the style is an independent noise. In this setup, we introduce a random color shift as the noise, though our approach can, in principle, be extended to other types of independent noise. The results show similar improvements with regularization, consistent with Theorem 4.5. We have further verified our theory on two more disentanglement tasks on the CelebA image dataset. First, we perform denoising with the same setting as CIFAR 10 and compared our approach with two standard baselines, infoGAN [69] and beta-VAE [70] adapted to the content disentanglement setting by conditioning on the observed style variable. As shown in the first two columns of Table 1, we observe the same trend as our previous experiments. Further, we perform face editing with *gender* as the style variable. As shown in the last two columns of Table 1, our method is superior to the conditional GAN baseline in terms of style transfer (CLIP score) and content preservation (LPIPS [75] score). Additional examples and quantitative results using different regularization weights for multiple metrics are provided in Appendix E.2.

**Speech disentanglement results.** Lastly, we apply DM-based disentanglement to a real-world application: *voice conversion adaptation* (VCA). In this setup, the style corresponds to speaker identity, while the content captures attributes like emotional state or health condition. Our goal is to learn representations that disentangle these factors, enabling robust classification under speaker shifts. We use speech emotion recognition on the IEMOCAP dataset [76] as a testbed, where generalization across unseen speakers serves as a proxy for successful disentanglement. This task instantiates *multi-view* disentanglement (Definition 3.4): different speakers provide style views, and the shared emotion across recordings is the content. We are also actively developing additional vision-domain multi-view experiments.

As shown in Figure 5, our DM-based approach achieves higher classification accuracy than

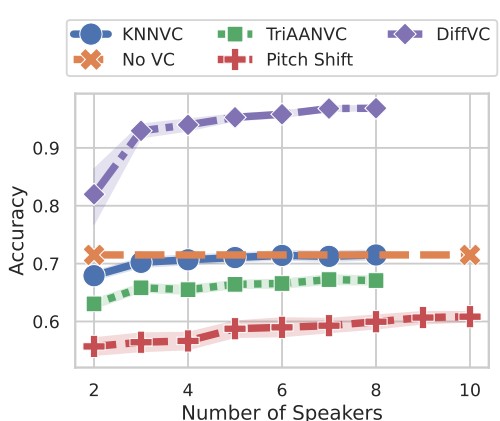

Figure 5: **Emotion recognition results on IEMO-CAP as a probing task for speech disentanglement**. DM-based disentanglement between emotion (content) and speaker (style) outperforms other methods. Data augmentation using multiple speakers further improves disentanglement.

several baselines, including no conversion, pitch shifting, and recent VCA models [65, 77]. Performance improves as more target speakers are used during training, consistent with our theoretical prediction of Theorem 4.1 that DMs can indeed achieve approximate disentanglement for speech data. It also validates Theorem 4.2 by demonstrating that multi-style data enhances content-style disentanglement. Further, more speech metrics and additional experiments beyond emotion classification (e.g., Alzheimer detection [78], Amyotrophic Lateral Sclerosis severity [79]) are detailed in Appendix E.3. Further, we also note that our current experiments focus on controlled settings aligned with the theory. Extensions to large-scale experimentation are left as an exciting future avenue of research.

## 6 Related works

**Diffusion model theory.** Early theoretical works on DMs analyzed their ability to learn data distributions, under different assumptions like log-Sobelev inequality [59], bounded moments [80, 42] and score approximation in $L^\infty$ [81] and $L^2$ [59, 42] norms. Later works compared DMs with likelihood-based models [82], and studied their capability to recover Gaussian mixtures [83–86], Ising models [87], low-dimensional subspaces [41, 66, 88, 89] and manifold structures [90]. Recently, [91, 92] analyzed the convergence of conditional DMs and the role of classifier-free guidance. Others have analyzed the training [93–95] and sampling [95–97] dynamics of DMs.

**Disentangled representation learning.** Disentanglement is defined via factorized representation [98, 21–23], group equivariance [99] or approximate independence [100]. It underpins many advances in self-supervised learning [52, 54, 101, 56], multimodal learning [102–104] and controllable generation [49, 38, 35, 36, 105]. Theoretically, disentanglement traces back to independent component analysis (ICA) [106], later extended to correlated factors [48, 29] and studied through the lens of modern SSL techniques such as data augmentation, contrastive loss [107] and self-distillation [108]. A key result [26] shows that unsupervised disentanglement is impossible without additional inductive biases, leading to weakly supervised methods using multiple views [29, 30], auxiliary labels [27, 109, 28], temporal cues [110, 111] and isometric constraints [112]. Other works analyze the disentangling capacity of (variational) auto-encoders [49, 113–116, 100, 117, 118].

## 7 Conclusion

We presented the first learning-theoretic framework for DM-based disentanglement, addressing unsolved challenges of prior works that focused primarily on deterministic, invertible mixing processes. Our framework introduces two notions of approximate disentanglement that generalize classical formulations to stochastic, non-invertible settings and shows how DMs can achieve them with partial labels or multi-view inputs. Moreover, in the special case of ISMs, we derive stronger guarantees, including finite-sample global convergence for gradient-based training. Our experiments across several domains, spanning Gaussian mixture recovery, image colorization, denoising, and voice conversion, show that theory-guided training methods, such as style-guidance regularization, lead to improved disentanglement and better downstream performance. Our framework lays the foundation for principled disentanglement with DMs and future works on more complex latent structures and modalities.

## Acknowledgments and Disclosure of Funding

This research was supported by Takeda Development Center Americas, INC. (successor in interest to Millennium Pharmaceuticals, INC.) This research could not be possible without the immense contributions from people living with ALS who have participated for months and years in the ALS Research Collaborative Study. We thank the ALS patient and caregiver community, in particular Augie's Quest for a Cure, for financially supporting ALS TDI's data collection work.

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

# Contents

# A Proof of Example 3.1

**Example A.1.** *(Example 3.1, restated) Suppose content $Z \sim \mathcal{N}(0,1)$ and $G \sim \mathcal{N}(0,1)$ are independent standard Gaussian r.v.'s and consider the noiseless setting with $\delta = 0$. Further, suppose $p_{f(Z,G)|Z} \not\equiv p_{f(-Z,G)|Z}$. Then we can choose the content/style encodings to be $\hat{Z} = Z\mathrm{sgn}(G)$, $\hat{G} = G$ and the decoder $\hat{f}(\hat{Z},\hat{G}) = f(\hat{Z}\mathrm{sgn}(\hat{G}),\hat{G})$, where $\mathrm{sgn}(x)$ denotes the sign of $x$. Then $(\hat{Z},\hat{G})$ are $(0,0)$-disentangled but not $\epsilon$-editable for some $\epsilon > 0$.*

*Proof.* First of all, notice that $\hat{Z} \sim \mathcal{N}(0,1)$ regardless of the value of $\hat{G}$ owing to the even symmetry of the standard Gaussian distribution, which implies $\hat{Z} \perp\!\!\!\perp \hat{G}$, or equivalently $I(\hat{Z};\hat{G}) = 0$. Second, by construction, $\hat{f}(\hat{Z},\hat{G}) = f(Z\mathrm{sgn}^2(G),G) = f(Z,G) = X$, and thus $I(\hat{Z},\hat{G};X) - I(Z,G;X) = 0$. Therefore by definition, $(\hat{Z},\hat{G})$ are $(0,0)$-disentangled. However, for an i.i.d sample $\hat{G}' \sim p_G$, the conditional distribution $p_{\hat{f}(\hat{Z},\hat{G}')|Z} = \frac{1}{2}p_{f(-Z,\hat{G}')|Z} + \frac{1}{2}p_{f(Z,\hat{G}')|Z} \not\equiv p_{f(Z,G)|Z}$ due to the fact that $\mathrm{sgn}(\hat{G})\mathrm{sgn}(\hat{G}')$ is a symmetric Bernoulli variable and $p_{f(Z,G)|Z} \not\equiv p_{f(-Z,G)|Z}$. Therefore, $(\hat{Z},\hat{G})$ is not $\epsilon$-editable for $\epsilon = d_{\mathrm{TV}}(p_{\hat{f}(\hat{Z},\hat{G}')|Z}, p_{X|Z}) > 0$. $\square$

# B Proof of Theorem 4.1

For clarity, we restate the theorem below.

**Theorem B.1.** *Suppose Assumption 3.5-3.6 hold, and $(\theta^*,\phi^*)$ be a minimizer of $L_c^{\gamma,\rho}$ defined in Eq. 7 with $\rho = I(Z;X) + C_1\delta, \gamma \geq C_2/T$ for some $C_1, C_2 > 0$. Set $t_0 = -\log(1-\delta^2)^{1/2}$ and $\hat{Z} := z_{\phi^*}(X)$. Then for any $\delta < \min\left\{\frac{1}{2}, \frac{1}{\sqrt{d_X}}\right\}$, the encodings $(\hat{Z},\hat{G}) = (z_{\phi^*}(X_{t_0}), g(X_{t_0}))$ are $(\epsilon,\nu)$-disentangled with $\epsilon = C_3\lambda_s\sigma_X^2 d_X\delta, \nu = C_4\sigma_X^2\delta^2$ for some constants $C_3, C_4 > 0$.*

## B.1 Main proof

Our result relies crucially on the following lemma.

**Lemma B.2.** *There exists $(\theta_1,\phi_1)$ such that for*

$$\delta < \min\left\{\frac{1}{2}, \frac{1}{\sqrt{d_X}}\right\}, t_0 = -\log(1-\delta^2)^{1/2}, t_1 = -\log(1-\delta)^{1/2},$$

*the followings hold:*

1. *The followings hold for the MIs $I(z_{\phi_1}(X_{t_0});X), I(g(X_{t_0});X)$ and $I(z_\phi(X_{t_0});g(X_{t_0})|X)$:*

$$I(z_{\phi_1}(X_{t_0});X) = I(Z;X) + O(\lambda_s\sigma_X d_X\delta),$$
$$I(g(X_{t_0});X) = I(G;X) + O(\lambda_s\sigma_X d_X\delta),$$
$$I(z_{\phi_1}(X_{t_0});g(X_{t_0})|X) = I(Z;G|X) + O(\lambda_s\sigma_X d_X\delta);$$

2. *The conditional score matching loss satisfies*

$$L_c(\theta_1,\phi_1) \leq \frac{(1+\sigma_X^2\delta^2)\delta^2 d_X(e^{2T}-e^{2t_1})}{2(T-t_1)(e^{2t_1}-1)(e^{2T}-1)} = O\left(\frac{\delta d_X}{T}\right).$$

First, we assume Lemma B.2 to be true and defer its proof to Section B.2. Therefore, by the property of $(\theta_1,\phi_1)$ and the optimality of $(\theta^*,\phi^*)$,

$$
\begin{aligned}
L_c^{\gamma,\rho}(\theta^*,\phi^*) &= L_c(\theta^*,\phi^*) + \gamma(I(z_{\phi^*}(X_{t_0});X) - \rho)_+ \\
&\leq L_c(\theta_1,\phi_1) + \gamma(I(z_{\phi_1}(X_{t_0});X) - I(Z;X)) + \gamma(I(Z;X) - \rho)_+ \\
&\leq \frac{(1+\sigma_X^2\delta^2)\delta d_X}{2(T-t_1)} + C_1\gamma\delta + \gamma(I(Z;X) - \rho)_+,
\end{aligned}
$$

for some $C_1 = O(\lambda_s \sigma_X d_X)$ and $C_2 := 1 + \sigma_X^2 \delta^2$. Since both $L_c$ and $I(Z;X)$ are nonnegative, this implies

$$L_c(\theta^*, \phi^*) \leq \frac{(1+\sigma_X^2\delta^2)\delta d_X}{2(T-t_1)} + C_1\gamma\delta + \gamma(I(Z;X)-\rho)_+,$$

$$I(z_{\phi^*}(X_{t_0});X) \leq I(Z;X) + \frac{(1+\sigma_X^2\delta^2)\delta d_X}{2\gamma(T-t_1)} + C_1\delta.$$

Choose $\gamma = \frac{1+\sigma_X^2\delta^2}{2(T-t_1)}$ and $\rho = I(Z;X) + O(\delta)$, then we have

$$I(z_{\phi^*}(X_{t_0});X) \leq I(Z;X) + C_1'\delta,$$

where $C_1' = O(\lambda_s \sigma_X d_X)$.

Let $h(X) := -\int_{\mathcal{X}} p(x)\log p(x)\mathrm{d}x$ denote the differential entropy of continuous random variable $X$. Then by definition, we have

$$\begin{aligned}
I(Z,G;X) &= h(Z,G) - h(Z,G|X) \\
&= h(Z) - h(Z|X) + h(G) - h(G|X) + I(Z;G|X) \\
&= I(Z;X) + I(G;X) + I(Z;G|X),
\end{aligned}$$

$$\begin{aligned}
I(X_T,\hat{Z},\hat{G};X) &= I(\hat{Z},\hat{G};X) + I(X_T;X|\hat{Z},\hat{G}) \\
&= h(\hat{Z},\hat{G}) - h(\hat{Z},\hat{G}|X) + I(X_T;X|\hat{Z},\hat{G}) \\
&= I(\hat{Z};X) + I(\hat{G};X) + I(\hat{Z};\hat{G}|X) - I(\hat{Z};\hat{G}) + I(X_T;X|\hat{Z},\hat{G}).
\end{aligned}$$

As a result,

$$I(\hat{Z};\hat{G}) = I(\hat{Z};X) + I(\hat{G};X) + I(\hat{Z},\hat{G}|X) + I(X_T;X|\hat{Z},\hat{G}) - I(X_T,\hat{Z},\hat{G};X)$$

$$= \underbrace{I(\hat{Z};X)-I(Z;X)}_{(i)} + \underbrace{I(\hat{G};X)-I(G;X)}_{(ii)} + \underbrace{I(\hat{Z};\hat{G}|X)-I(Z;G|X)}_{(iii)} +$$

$$\underbrace{I(Z,G;X)-I(X_T,\hat{Z},\hat{G};X)}_{(iv)} + \underbrace{I(X_T;X|\hat{Z},\hat{G})}_{(v)} \tag{14}$$

where terms $(i)(ii)(iii)$ can be upper bounded by item 1 as $3C_1\delta$. To bound the term $(iv)$, use the definition $I(Z,G;X)$:

$$I(Z,G;X) = h(X) - h(N) = h(X) - \frac{1}{2}\log(2\pi e\delta^2 d_X),$$

and apply the maximum entropy inequality on $I(X_T,z(X_{t_0}),G;X)$:

$$\begin{aligned}
I(X_T,z(X_{t_0}),\hat{G};X) &\geq h(X) - \frac{1}{2}\log 2\pi e\mathbb{E}\left\| (e^T - e^{-T})s_{\theta^*}(X_T,\hat{Z},\hat{G},T) + e^T X_T - X \right\|^2 \\
&\geq h(X) - \frac{1}{2}\log 2\pi e^{2T+1}(1-e^{-2T})^2 \lim_{t_1\to T} L_c(\theta^*,\phi^*) \\
&\geq h(X) - \frac{1}{2}\log 2\pi e C_2\delta^2 d_X = I(Z,G;X) - \frac{1}{2}\log C_2.
\end{aligned}$$

The last inequality uses the optimality of $(\theta^*,\phi^*)$ and therefore $s_\theta(x,z,g,t)$'s needs to achieve minimal loss at any $t \in [t_1,T]$, which is upper-bounded by item 2 of Lemma B.2 as

$$\lim_{t_1\to T} L_c(\theta_1,\phi_1) \leq \lim_{t_1\to T} \frac{C_2\delta^2 d_X(e^{2T}-e^{2t_1})}{2(T-t_1)(e^{2t_1}-1)(e^{2T}-1)} = \frac{C_2\delta^2 d_X}{e^{2T}(1-e^{-2T})^2}.$$

To bound term $(v)$ of the RHS of Eq. 14, notice that $(\hat{Z},\hat{G})$ and $X_{t_0}$ are invertible, and thus

$$I(X_T;X|\hat{Z},\hat{G}) = I(X_T;X|X_{t_0}) = 0.$$

Combining the bounds and choose $T := \Omega(\log\frac{1}{\delta})$, we conclude that

$$I(z_\phi(X_{t_0});\hat{G}) \leq \frac{1}{2}\log C_2 + 3C_1\delta = O\big(\sigma_X^2\delta^2 + \lambda_s\sigma_X d_X\delta\big) = O(\lambda_s\sigma_X^2 d_X\delta).$$

## B.2 Proof of Lemma B.2

To prove the lemma, we need the following technical lemmas, whose proofs are deferred to B.3, B.4 and B.5 respectively.

**Lemma B.3.** *Suppose random variable $Y = \alpha X + N \sim p_Y$, for independent random variables $X \sim p_X$ and $N \sim \mathcal{N}(0, \sigma^2 I_d)$, where $\alpha := \sqrt{1-\sigma^2}$. Then, for any distribution $q(x|z)$ with an L-Lipschitz score function in $x$ for any $z \in \mathcal{X}$, then the following inequality holds:*

$$\mathbb{E}_{p_{XYZ}(x,y,z)} \log \frac{q(y|z)}{q(x|z)} \leq CL[\sigma^2(1+\sigma^2)\mathbb{E}\|X\|^2 + \sigma^2 d + \sigma\sqrt{\mathbb{E}\|X\|^2 d}],$$

*for some constant $C > 0$ independent of $\sigma$, $\mathbb{E}\|X\|^2$ and $d$.*

**Lemma B.4.** *For $\delta < 1/2$ and $\alpha := \sqrt{1-\delta^2}$, the KL divergence between two Gaussian distributions $\mathcal{N}(\alpha\mu, \delta^2 I_d)$ and $\mathcal{N}(\frac{1}{\alpha}\mu, \frac{\delta^2}{\alpha^2} I_d)$ is upper-bounded as*

$$D_{\mathrm{KL}}\left(\mathcal{N}(\alpha\mu, \delta^2 I_d) \middle\| \mathcal{N}\left(\frac{1}{\alpha}\mu, \frac{\delta^2}{\alpha^2} I_d\right)\right) \leq \left(\frac{\|\mu\|^2}{2} + \frac{d}{6}\right)\delta^2.$$

**Lemma B.5.** *Suppose random variable $Y = \alpha X + N \sim p_Y$, for independent random variables $X \sim p_X$ and $N \sim \mathcal{N}(0, \sigma^2 I_d)$, where $\alpha := \sqrt{1-\sigma^2}$. Further, suppose $p_X(x) > 0$ for any $x \in \mathbb{R}^d$ and the score function of $p_X$ is L-Lipschitz, then the following holds*

$$\max\{D_{\mathrm{KL}}(p_X\|p_Y), D_{\mathrm{KL}}(p_Y\|p_X)\} \leq CL(\mathbb{E}\|X\|^2)^{1/2}\delta d,$$

*for some constant $C > 0$.*

Assuming Lemma B.3-B.5, define $\bar{X} := f(Z, G)$ and choose $t_0 := \frac{1}{2}\log\frac{1}{1-\delta^2}$. Then by the property of the OU process,

$$X_{t_0} = \alpha X + \delta N_{t_0},$$

where $\alpha := \sqrt{1-\delta^2}$. Next, since the mixing function $f$ is invertible, we can write its inverse $f^{-1} : \mathbb{R}^{d_X} \mapsto \mathbb{R}^{d_Z} \times \mathbb{R}^{d_G}$ as $f^{-1}(x) =: [z(x), g(x)]$, where we call $z : \mathbb{R}^{d_X} \mapsto \mathbb{R}^{d_Z}$ the *true content encoder* and $g : \mathbb{R}^{d_X} \mapsto \mathbb{R}^{d_G}$ the *true style encoder* such that $z(\bar{X}) = Z, g(\bar{X}) = G$. By Definition 3.3, we have access to the true style encoder $g(\cdot)$. Set $\phi_1$ such that $z_{\phi_1}(x) = z(x)$. Further, define

$$q(z|x) := p_{z(X_{t_0})|X}(z|x),$$
$$q(z) := p_{z(X_{t_0})}(z),$$
$$p(z|x) := p_{Z|X}(z|x),$$
$$p(z) := p_Z(z),$$

then by definition,

$$\begin{aligned}
&I(z(X_{t_0}); X) - I(Z; X) \\
&= \mathbb{E}_{p_X}[D_{\mathrm{KL}}(q(Z|X)\|q(Z)) - D_{\mathrm{KL}}(p(Z|X)\|p(Z))] \\
&= \mathbb{E}_{q_X}[D_{\mathrm{KL}}(q(Z|X)\|p(Z)) - D_{\mathrm{KL}}(q(Z)\|p(Z)) - D_{\mathrm{KL}}(p(Z|X)\|p(Z))] \\
&\leq \mathbb{E}_{p_X}[D_{\mathrm{KL}}(q(Z|X)\|p(Z)) - D_{\mathrm{KL}}(p(Z|X)\|p(Z))] \\
&\leq \mathbb{E}_{p_X} D_{\mathrm{KL}}(q(Z|X)\|p(Z|X)),
\end{aligned}$$

where the first inequality uses the non-negativity of $D_{\mathrm{KL}}$ and the second inequality uses the triangle inequality of $D_{\mathrm{KL}}$: $D_{\mathrm{KL}}(p\|q) \leq D_{\mathrm{KL}}(p\|r) + D_{\mathrm{KL}}(r\|q)$ for any pdfs $p, q, r$. Further, by the data processing inequality,

$$\begin{aligned}
&\mathbb{E}_{p_X} D_{\mathrm{KL}}(q(Z|X)\|p(Z|X)) \\
&= \mathbb{E}_{p_X} D_{\mathrm{KL}}(p_{z(X_{t_0})|X}\|p_{z(\bar{X})|X}) \\
&\leq \mathbb{E}_{p_X} D_{\mathrm{KL}}(p_{X_{t_0}|X}\|p_{\bar{X}|X}) \\
&= \mathbb{E}_{p_{XX_{t_0}}(x,y)} \log \frac{\mathcal{N}(y|\alpha_{t_0} x, \delta^2 I_{d_X})}{p_{\bar{X}}(y)\mathcal{N}(x|\alpha y, \delta^2 I_{d_X})/p_X(x)} \\
&= \mathbb{E}_{p_{XX_{t_0}}(x,y)} \log \frac{p_X(x)}{p_{\bar{X}}(y)} + O((d_X + \mathbb{E}\|X\|)\delta^2) \\
&= D_{\mathrm{KL}}(p_X\|p_{\bar{X}}) + \mathbb{E}_{p_{XX_{t_0}}(x,y)} \log \frac{p_{\bar{X}}(x)}{p_{\bar{X}}(y)} + O((d_X + \mathbb{E}\|X\|)\delta^2),
\end{aligned}$$

where the second-to-last equality uses Lemma B.4:

$$\mathbb{E}_{p_{XX_{t_0}}(x,y)}\log\frac{\mathcal{N}(y|\alpha x,\delta^2 I_{d_X})}{\mathcal{N}(x|\alpha y,\delta^2 I_{d_X})}$$

$$=\mathbb{E}_{p_{XX_{t_0}}(x,y)}\log\frac{\mathcal{N}(y|\alpha_{t_0}x,\delta_{t_0}^2 I_{d_X})}{\mathcal{N}\left(y|\frac{1}{\alpha}x,\frac{\delta^2}{\alpha^2}I_{d_X}\right)}+d_X\log(1/\alpha)$$

$$=O((d_X+\mathbb{E}\|X\|)\delta^2). \tag{15}$$

To proceed, notice further that

$$D_{\mathrm{KL}}(p_X\|p_{\bar{X}})+\mathbb{E}_{p_{XX_{t_0}}(x,y)}\log\frac{p_{\bar{X}}(x)}{p_{\bar{X}}(y)}$$

$$=D_{\mathrm{KL}}(p_X\|p_{\bar{X}})+O(\lambda_s\sigma_X d_X\delta)$$

$$=O(\lambda_s d_X\sigma_X\delta), \tag{16}$$

where the first equality uses Lemma B.3 with $\delta<1/\sqrt{d_X}$, and the second inequality uses Lemma B.5. Combining this with Eq. 15 yields

$$I(z(X_{t_0});X)-I(Z;X)=O(\lambda_s\sigma_X d_X\delta)=C_1\delta,$$

where $C_1:=O(\lambda_s\sigma_X d_X)$. Using a similar strategy, we can prove that

$$I(\hat{G};X)-I(G;X)=O(\lambda_s\sigma_X d_X\delta)=C_1\delta.$$

For the last MI $I(\hat{Z};\hat{G}|X)$, define

$$q(z,g|x):=p_{\hat{Z}\hat{G}|X}(z,g|x),p(z,g|x):=p_{ZG|X}(z,g|x),$$

and notice that

$$I(\hat{Z};\hat{G}|X)-I(Z;G|X)$$
$$=\mathbb{E}_{p_X}[D_{\mathrm{KL}}(q(Z,G|X)\|q(Z|X)q(G|X))-D_{\mathrm{KL}}(p(Z,G|X)\|p(Z|X)p(G|X))]$$
$$=\mathbb{E}_{p_X}[D_{\mathrm{KL}}(q(Z,G|X)\|p(Z|X)p(G|X))-D_{\mathrm{KL}}(p(Z,G|X)\|p(Z|X)p(G|X))]-$$
$$\quad\mathbb{E}_{p_X}[D_{\mathrm{KL}}(p(Z|X)\|q(Z|X))+D_{\mathrm{KL}}(p(G|X)\|q(G|X))]$$
$$\leq\mathbb{E}_{p_X}[D_{\mathrm{KL}}(q(Z,G|X)\|p(Z|X)p(G|X))-D_{\mathrm{KL}}(p(Z,G|X)\|p(Z|X)p(G|X))]$$
$$\leq\mathbb{E}_{p_X}D_{\mathrm{KL}}(q(Z,G|X)\|p(Z,G|X))$$
$$\leq\mathbb{E}_{p_X}D_{\mathrm{KL}}(p_{X_{t_0}|X}\|p_{\bar{X}|X})=O(\lambda_s d_X\sigma_X\delta)=C_1\delta,$$

where we again apply non-negativity of $D_{\mathrm{KL}}$ on the first inequality and triangle inequality of $D_{\mathrm{KL}}$ on the second inequality. For the last equality, we combine Eq. 15 and Eq. 16.

To prove item 2, set $\theta_1$ so that the score function

$$s_{\theta_1}(X_t,\hat{Z},\hat{G},t)=\frac{\exp(-t)f(\hat{Z},\hat{G})-X_t}{1-\exp(-2t)}.$$

Then by setting $t_1:=\frac{1}{2}\log\frac{1}{1-\delta}$, the loss $L_c$ becomes

$$L_c(\theta_1,\phi_1)=\frac{1}{T-t_1}\int_{t_1}^T\frac{\exp(-2t)\mathbb{E}\|f(\hat{Z},\hat{G})-X\|^2}{(1-\exp(-2t))^2}\mathrm{d}t$$

$$=:\frac{\tilde{L}_c(\theta_1,\phi_1)}{T-t_1}\int_{e^{t_1}}^{e^T}\frac{\mathrm{d}\tau}{2(\tau-1)^2}$$

$$=\frac{\tilde{L}_c(\theta_1,\phi_1)}{2(T-t_1)}\left(\frac{1}{e^{2t_1}-1}-\frac{1}{e^{2T}-1}\right)\leq\frac{\tilde{L}_c(\theta_1,\phi_1)(e^{2T}-e^{2t_1})}{2(T-t_1)(e^{2t_1}-1)(e^{2T}-1)},$$

Further, notice that

$$\tilde{L}_c(\theta_1,\phi_1):=\mathbb{E}\|f(\hat{Z},\hat{G})-X\|^2$$
$$=\mathbb{E}\|f(z(X_{t_0}),g(X_{t_0}))-f(z(X),g(X))\|^2$$
$$=\mathbb{E}\|X_{t_0}-X\|^2$$
$$\leq C_2\delta^2 d_X,$$

where $C_2 := 1 + \delta^2\sigma_X^2$. The second equality uses the definition of inverse functions so that $f(z(x),g(x))=x,\forall x\in\mathbb{R}^{d_X}$. The last inequality uses the fact that

$$
\begin{aligned}
\mathbb{E}\|X_{t_0}-X\|^2 &= \mathbb{E}\|X-\bar{X}\|^2 \\
&= \mathbb{E}\|(\alpha-1)X+N_t\|^2 \\
&= (1-\sqrt{1-\delta^2})^2\mathbb{E}\|X\|^2+\mathbb{E}\|N_{t_0}\|^2 \\
&\leq \delta^4\sigma_X^2 d_X+\delta^2 d_X.
\end{aligned}
$$

As a result, item 2 follows from

$$
L_c(\theta_1,\phi_1)\leq\frac{C_2\delta^2 d_X(e^{2T}-e^{2t_1})}{2(T-t_1)(e^{2t_1}-1)(e^{2T}-1)}\leq\frac{C_2\delta d_X}{2(T-\log(1-\delta)^{1/2})}=O\left(\frac{\delta d_X}{T}\right)
$$

with the choice of $t_1$ and $\delta$ and the fact that $0<\delta<\min\{T,1\}$.

## B.3  Proof of Lemma B.3

To begin, we make use of the following lemma proved in Section B.6.

**Lemma B.6.** *Suppose the score function $s_q(x|z):=\nabla_x\log q(x|z)$ of the probability density $q(x|z)$ is $L$-Lipschitz as a function of $x$, then the following inequality holds:*

$$
\log\frac{q(y|z)}{q(x|z)}\leq(L\|x\|+\|s_q(\mathbf{0}_d|z)\|)\|y-x\|+\frac{L\|y-x\|^2}{2}.
$$

Set $s_q(x|z):=\nabla_x\log q(x|z)$, then by Lemma B.6,

$$
\mathbb{E}_{p_{XYZ}(x,y,z)}\log\frac{q(y|z)}{q(x|z)}
$$

$$
\leq\mathbb{E}_{p_{XYZ}(x,y,z)}(L\|x\|+\|s_q(\mathbf{0}_d|z)\|)\|y-x\|+\frac{L}{2}\mathbb{E}_{p_{XY}(x,y)}\|y-x\|^2.
$$

For the first term of the RHS, notice that

$$
\begin{aligned}
&\mathbb{E}_{p_{XYZ}(x,y,z)}(L\|x\|+\|s_q(\mathbf{0}_d|z)\|)\|y-x\| \\
&\leq\sqrt{\mathbb{E}(L\|X\|+\|s_q(\mathbf{0}_d|Z)\|)^2\mathbb{E}\|Y-X\|^2} \\
&=\sqrt{\mathbb{E}(L\|X\|+\|s_q(\mathbf{0}_d|Z)\|)^2}\sqrt{(1-\alpha)^2\mathbb{E}\|X\|^2+\sigma^2 d} \\
&\leq\sqrt{2L^2\mathbb{E}\|X\|^2+C_1}\sqrt{(1-\alpha)^2\mathbb{E}\|X\|^2+\sigma^2 d} \\
&\leq C_2 L(\sigma^2\mathbb{E}\|X\|^2+\sigma\sqrt{\mathbb{E}\|X\|^2 d}),
\end{aligned}
$$

where $C_1:=2\sup_z\mathbb{E}\|s_q(\mathbf{0}_d|z)\|^2$ and $C_2$ large enough. To bound the second term of the RHS, notice that

$$
\begin{aligned}
\frac{L}{2}\mathbb{E}_{p_{XY}(x,y)}\|y-x\|^2 &= \frac{L}{2}[(1-\alpha)^2\mathbb{E}\|X\|^2+\mathbb{E}\|N\|^2] \\
&\leq\frac{L\sigma^2}{2}(\sigma^2\mathbb{E}\|X\|^2+d).
\end{aligned}
$$

Combining the two terms yields

$$
\mathbb{E}_{p_{XYZ}(x,y,z)}\log\frac{q(y|z)}{q(x|z)}\leq CL[\sigma^2(1+\sigma^2)\mathbb{E}\|X\|^2+\sigma^2 d+\sigma\sqrt{\mathbb{E}\|X\|^2 d}],
$$

for some $C>0$ large enough.

## B.4 Proof of Lemma B.4

Use the formula for the KL divergence between Gaussians:

$$D_{\mathrm{KL}}(\mathcal{N}(\alpha\mu,\delta^2 I_d)||\mathcal{N}(\mu/\alpha,(\delta/\alpha)^2 I_d))$$

$$=\left(\frac{\delta^2}{(\delta/\alpha)^2}-1\right)d/2+\frac{\|\alpha\mu-\mu/\alpha\|^2}{2(\delta/\alpha)^2}+d\log\frac{1}{\alpha}$$

$$=-\frac{d}{2}\delta^2+\frac{\delta^2\|\mu\|^2}{2}+d\log\frac{1}{\alpha}$$

$$=\frac{(\|\mu\|^2-d)\delta^2}{2}+\frac{d}{2}\log\left(1+\frac{\delta^2}{1-\delta^2}\right)$$

$$\leq\frac{(\|\mu\|^2-d)\delta^2}{2}+\frac{2d\delta^2}{3}=\left(\frac{\|\mu\|^2}{2}+\frac{d}{6}\right)\delta^2.$$

## B.5 Proof of Lemma B.5

By Jensen's inequality,

$$D_{\mathrm{KL}}(p_Y||p_X)$$

$$=\mathbb{E}_{\int p_X(x)\mathcal{N}(y|\alpha x,\sigma^2 I_d)\mathrm{d}x}\log\frac{\int p_X(x)\mathcal{N}(y|\alpha x,\sigma^2 I_d)\mathrm{d}x}{p_X(y)}$$

$$\leq\mathbb{E}_{p_X(x)\mathcal{N}(y|\alpha x,\sigma^2 I_d)}\log\frac{p_X(x)\mathcal{N}(y|\alpha x,\sigma^2 I_d)}{p_X(y)\mathcal{N}(x|y/\alpha,(\sigma/\alpha)^2 I_d)}$$

$$=\mathbb{E}_{p_X(x)\mathcal{N}(y|\alpha x,\sigma^2 I_d)}\log\frac{p_X(x)}{p_X(y)}+d\log\alpha$$

$$\leq\mathbb{E}_{p_X(x)\mathcal{N}(y|\alpha x,\sigma^2 I_d)}\log\frac{p_X(x)}{p_X(y)}-\frac{\sigma^2 d}{2}$$

$$\leq CL(\sigma^2\mathbb{E}\|X\|^2+\sigma^2 d+\sigma\sqrt{d}\mathbb{E}\|X\|),$$

for some $C>0$, where the last inequality uses Lemma B.3.

Similarly, apply Jensen's inequality and Lemma B.3,

$$D_{\mathrm{KL}}(p_X||p_Y)=\mathbb{E}_{p_X(x)}\log\frac{p_X(x)}{\int p_X(y)\mathcal{N}(x|\alpha y,\sigma^2 I_d)\mathrm{d}y}$$

$$\leq\mathbb{E}_{p_X(x)\mathcal{N}(y|x/\alpha,(\sigma/\alpha)^2 I_d)}\log\frac{p_X(x)\mathcal{N}(y|x/\alpha,(\sigma/\alpha)^2 I_d)}{p_X(y)\mathcal{N}(x|\alpha y,\sigma^2 I_d)}$$

$$=\mathbb{E}_{p_X(x)\mathcal{N}(y|x/\alpha,(\sigma/\alpha)^2 I_d)}\log\frac{p_X(x)}{p_X(y)}+d\log\alpha$$

$$\leq CL(\sigma^2\mathbb{E}\|X\|^2+\sigma^2 d+\sigma\sqrt{d}\mathbb{E}\|X\|).$$

## B.6 Proof of Lemma B.6

by the Lipschitz property of the score function of $s_q$, for any $(x,y)\in\mathcal{X}^2$,

$$\log\frac{q(y|z)}{q(x|z)}\leq|\langle s_q(x|z),y-x\rangle|+\frac{L\|y-x\|^2}{2}$$

$$\leq\|s_q(x|z)\|\|y-x\|+\frac{L\|y-x\|^2}{2}.$$

Apply the Lipschitz property of $s_q$ again,

$$\log\frac{q(y|z)}{q(x|z)}\leq(L\|x\|+\|s_q(\mathbf{0}_d|z)\|)\|y-x\|+\frac{L\|y-x\|^2}{2}. \tag{17}$$

### B.7 Extension to correlated content and style

To extend Theorem 4.1 to correlated content and style, we can add an additional term to the right-hand side of Eq. 14 Theorem 4.2 as

$$I(\hat{Z};\hat{G}) = (i)+(ii)+(iii)+(iv)+(v)+I(Z;G) \le (i)+(ii)+(iii)+(iv)+(v)+\epsilon_1, \qquad (18)$$

where (i)-(v) are defined and bounded as in Appendix B.1.

## C  Proof of Theorem 4.2

For clarity, we restate the theorem below.

**Theorem C.1.** *Under Assumption 3.5-3.6 and additional regularity conditions (Assumption C.2-C.4). Consider the two-view, unimodal setting with $\delta_1 =: \delta$ and $\delta_2 = 0$. Let $(\theta^*, \phi_Z^*, \phi_G^*)$ be a minimizer of $L_m$ and $\hat{Z}^i = z_{\phi_Z^*}(X^i)$, $\hat{G}^i = g_{\phi_G^*}(X^i)$ for each view $i$. Set $t_0 = -\log(1-\delta^2)^{1/2}$ and the diffusion step size $\eta := C_5 \frac{\delta^3}{\lambda_\Theta^2 T}$ for some constant $C_5 > 0$. Then for some constants $C_6, C_7 > 0$: (i) $I(\hat{Z}^2; \hat{G}^1) \le C_6 \lambda_s \sigma_X d_x \delta$; (ii) the content encoding $\hat{Z}^2$ and the style encoding $\hat{G}^1$ are $C_7 \sqrt{\lambda_s \sigma_X^2 d_X \delta}$-editable.*

### C.1  Main proof

Let $\bar{X}^i := f_i(Z, G^i) = f(Z, G^i)$, where the last equality assumes the unimodal setting. By the invertibility of the mixing function, there exist functions $z : \mathbb{R}^{d_X} \mapsto \mathbb{R}^{d_z}$ such that $z(\bar{X}^i) = Z$ and $g : \mathbb{R}^{d_X} \mapsto \mathbb{R}^{d_G}$ such that $g(\bar{X}^i) = G^i$, $i \in \{1,2\}$. To prove this theorem, we need additional assumptions below.

**Assumption C.2.** *For any $\theta \in \Theta, \phi \in \Phi$, the estimated score function $s_\theta(x,z,g,t)$ is $\frac{\lambda_\Theta}{\sigma(t)^2}$-Lipschitz in all arguments.*

**Assumption C.3.** *The style encoder $g_{\phi_G} : \mathbb{R}^{d_X} \mapsto \mathbb{R}^{d_G}$ is $\lambda_g$-Lipschitz in $x \in \mathbb{X}$ and the content encoder $z_{\phi_Z} : \mathbb{R}^{d_X} \mapsto \mathbb{R}^{d_z}$ is $\lambda_z$-Lipschitz. Further, there exists $\phi_{1,G} \in \Phi_G$ such that $z_{\phi_{1,Z}}(\cdot) = z(\cdot)$ and $g_{\phi_{1,G}}(\cdot) = g(\cdot)$.*

**Assumption C.4.** *The mixing function $f$ is $\lambda_f$-Lipschitz.*

Assumption C.2 ensures the function class for the bottleneck and the score function are Lipschitz, analogous to the setting in [41]. Assumption C.3 assumes that the content and style encoders are both Lipschitz and the true content and style functions are realizable by the function classes $\Phi_Z$ and $\Phi_G$ respectively. This is a mild assumption since the style and content encoders are assumed to be NNs and common choices of neural architectures are either already Lipschitz or constrained to be so through regularizations [119–121]. For Assumption C.4, note that the invertibility of $f_\theta$ is not strictly necessary and can be relaxed to injectivity by partitioning the domain.

Again, we start by introducing a couple helpful lemmas and postponing their proofs to Section C.2 and C.3.

**Lemma C.5.** *There exists $(\theta_1, \phi_{Z,1}, \phi_{G,1})$ such that for*

$$\delta < \min\left\{\frac{1}{2}, \frac{1}{\sqrt{d_X}}\right\}, \quad t_0 = -\log(1-\delta^2)^{1/2}, \quad t_1 = -\log(1-\delta)^{1/2},$$

*the followings hold:*

1. *The followings hold for MIs $I(\hat{G}^1; X^2)$ and $I(\hat{G}^1; Z)$:*

$$I(\hat{G}^2; X) = O(\lambda_s \sigma_X d_X \delta),$$
$$I(\hat{G}^1; Z) = O(\lambda_s \sigma_X d_X \delta);$$

2. *The regularized score matching loss for the multi-view disentanglement in Eq. 8 satisfies*

$$L_m(\theta_1, \phi_1) \le \frac{\lambda_f \lambda_g (\sigma_X^2 \delta^2 + 2) \delta^2 d_X (e^{2T} - e^{2t_1})}{2(T-t_1)(e^{2t_1}-1)(e^{2T}-1)} = O\left(\frac{\lambda_f \lambda_g d_X \delta}{T}\right),$$

*where $\phi_1 := [\phi_{Z,1}, \phi_{G,1}]$.*

**Lemma C.6.** *(Novikov's condition) The following bound holds:*

$$\mathbb{E}_{X^1,(X_t^{1,\leftarrow})_t}\exp\left(\frac{1}{2}\int_0^{T-t}\|s_\theta(X_t^{1,\leftarrow},\hat{Z}^2,\hat{G}^1,T-t)-\nabla_x\log p_{T-t_0}(X_t^{1,\leftarrow}|X^1)\|^2\mathrm{d}t\right)<\infty$$

Let $z:\mathbb{R}^{d_X}\mapsto\mathbb{R}^{d_Z}$ and $g:\mathbb{R}^{d_X}\mapsto\mathbb{R}^{d_G}$ be the true content and style encoders defined in Section B.1, and define $\bar{X}^i:=f(Z^i,G^i)$, $i\in\{1,2,3\}$. By Definition 3.4, we have $X^2=\bar{X}^2$. Further, define $\hat{X}^{21}:=\hat{X}^{21,\leftarrow}_{T-t_1}$ to be a sample from the following estimated backward process:

$$\mathrm{d}\hat{X}_t^{21,\leftarrow}=[\hat{X}_t^{21,\leftarrow}+2s_{\theta^*}(\hat{X}_{k\eta}^{21,\leftarrow},\hat{Z}^2,\hat{G}^1,T-k\eta)]\mathrm{d}t+\sqrt{2}\mathrm{d}B_t^{\leftarrow},\hat{X}_0^{21,\leftarrow}\sim p_T,t\in[k\eta,(k+1)\eta].\tag{19}$$

We begin by proving item (i) of the theorem. To this end, we apply the first part of Lemma C.5 and data processing inequality:

$$I(\hat{G}^1;\hat{Z}^2)\leq I(\hat{G}^1;X^2)=O(\lambda_s\sigma_X d_X\delta).\tag{20}$$

Next, we prove item (ii). Then apply triangle inequality:

$$\mathbb{E}_{p_Z}d_{\mathrm{TV}}(p_{X_{t_1}^1|Z},p_{\hat{X}^{23}|Z})\leq\underbrace{\mathbb{E}_{p_Z}d_{\mathrm{TV}}(p_{X_{t_1}^1|Z},p_{\hat{X}^{21}|Z})}_{(a)}+\underbrace{\mathbb{E}_{p_Z}d_{\mathrm{TV}}(p_{\hat{X}^{21}|Z},p_{\hat{X}^{23}|Z})}_{(b)}.\tag{21}$$

To bound term $(a)$, we check that Novikov's condition holds by Lemma C.6, and apply Girsanov's theorem on the estimated backward process Eq. 19 with $t_1:=-\log(1-\delta)^{1/2}$ followed by data processing inequality:

$$\mathbb{E}_{p_Z}d_{\mathrm{TV}}(p_{X_{t_1}^1|Z},p_{\hat{X}^{21}|Z})\leq\mathbb{E}_{p_Z}d_{\mathrm{TV}}(p_{X_{t_1}^1|X^1},p_{\hat{X}^{21}|X^1})$$

$$=O\left((\sqrt{d_X}\eta+\sigma_X^2\eta)\lambda_\Theta\sqrt{T}/\delta+\sqrt{L_c(\theta^*,\phi_Z^*,\phi_G^*)T}\right)$$

$$=O\left((\sqrt{d_X}\eta+\sigma_X^2\eta)\lambda_\Theta\sqrt{T}/\delta+\sqrt{\lambda_f\lambda_g\delta d_X}\right),$$

where the last equality applies Lemma C.5. Set $\eta:=\frac{\delta^3}{\lambda_\Theta^2 T}$, we obtain

$$\mathbb{E}_{p_Z}d_{\mathrm{TV}}(p_{X_{t_1}^1|Z},p_{\hat{X}^{21}|Z})=O\left(\sqrt{\lambda_f\lambda_g\sigma_X^2 d_X\delta}\right).$$

To bound the second term $(b)$, applying data processing inequality, Pinsker's inequality, triangle inequality and data processing inequality again in that order,

$$d_{\mathrm{TV}}(p_{\hat{X}^{11}|Z},p_{\hat{X}^{13}|Z})$$

$$\leq d_{\mathrm{TV}}(p_{\hat{Z}^2\hat{G}^1|Z},p_{\hat{Z}^2\hat{G}^3|Z})\leq d_{\mathrm{TV}}(p_{\hat{Z}^2\hat{G}^1|Z},p_{\hat{Z}^2|Z}p_{\hat{G}^1|Z})+d_{\mathrm{TV}}(p_{\hat{Z}^2\hat{G}^2|Z},p_{\hat{Z}^2\hat{G}^3|Z})$$

$$=d_{\mathrm{TV}}(p_{\hat{Z}^2\hat{G}^1|Z},p_{\hat{Z}^2|Z}p_{\hat{G}^1|Z})+d_{\mathrm{TV}}(p_{\hat{Z}^2|Z}p_{\hat{G}^1|Z},p_{\hat{Z}^2|Z}p_{\hat{G}^3})$$

$$\leq\sqrt{\frac{I(\hat{Z}^2;\hat{G}^1|Z)}{2}}+\sqrt{\frac{I(\hat{G}^1;Z)}{2}}\leq\sqrt{\frac{I(X^2;X^1|Z)}{2}}+O(\sqrt{\lambda_s\sigma_X d_X\delta})=O(\sqrt{\lambda_s\sigma_X d_X\delta}),\tag{22}$$

where the last inequality uses Lemma C.5 and the last equality using the conditional independence $X^1\perp\!\!\!\perp X^2|Z$.

Combining the bounds on $(i)(ii)$, we obtain

$$\mathbb{E}_{p_Z}d_{\mathrm{TV}}(p_{X_{t_1}^1|Z},p_{\hat{X}^{23}|Z})=O\left(\sqrt{\sigma_X^2 d_X\delta}+\sqrt{\lambda_s\sigma_X d_X\delta}\right)=O\left(\sqrt{(\lambda_s+\lambda_f\lambda_g)\sigma_X^2 d_X\delta}\right).$$

## C.2   Proof of Lemma C.5

We start by proving item 1. By Definition 3.4, the styles of different views are independent, i.e. $G^1\perp\!\!\!\perp X^2$ and thus $I(G^1;X^2)=0$. Choose parameters $(\theta_1,\phi_{Z,1},\phi_{G,1})$ such that

$$\hat{f}_{\theta_1}(x)=f(x),\quad z_{\phi_{Z,1}}(x)=z(x),\quad g_{\phi_{G,1}}(x)=g(x).$$

Further, define
$$q(g|x^2):=p_{\hat{G}^1|X^2}(g|x^2), \quad p(g|x^2):=p_{G^1|X^2}(g|x^2),$$
$$q(g|x^1):=p_{\hat{G}^1|X^1}(g|x^1), \quad p(g|x^1):=p_{G^1|X^1}(g|x^1),$$
$$p(x^1|x^2):=p_{X^1|X^2}(x^1|x^2),$$

and by a similar argument as in the proof of Lemma B.2 in Section B.2,
$$I(\hat{G}^1;X^2)=I(\hat{G}^1;X^2)-I(G^1;X^2)\leq \mathbb{E}_{X^2}D_{\mathrm{KL}}(q(g|x^2)||p(g|x^2)).$$

Further, by data processing inequality,
$$\mathbb{E}_{p_{X^2}}D_{\mathrm{KL}}(q(g|x^2)||p(g|x^2))\leq \mathbb{E}_{p_{X^2}}D_{\mathrm{KL}}(p_{\hat{G}^1,X^1|X^2}||p_{G^1,X^1|X^2})$$
$$=\mathbb{E}_{X^2}D_{\mathrm{KL}}(p(x^1|x^2)q(g|x^1)||p(x^1|x^2)p(g|x^1))$$
$$=\mathbb{E}_{p_{X^1}}D_{\mathrm{KL}}(q(g|x^1)||p(g|x^1)), \tag{23}$$

where the first equality uses the fact that $(\hat{G}^1,G^1)\perp\!\!\!\perp X^2|X^1$ and thus
$$p_{\hat{G}^1|X^1,X^2}(g|x^1,x^2)=q(g|x^1), \quad p_{G^1|X^1,X^2}(g|x^1,x^2)=p(g|x^1).$$

Again by a similar argument as in Section B.2,
$$\mathbb{E}_{p_{X^1}}D_{\mathrm{KL}}(q(g|x^1)||p(g|x^1))=O(\lambda_s\sigma_X d_X\delta_1)$$
$$\Longrightarrow I(\hat{G}^1;X^2)=O(\lambda_s\sigma_X d_X\delta_1).$$

As a result, by the chain rule of MI,
$$I(\hat{G}^1;Z,X^2)=I(\hat{G}^1;Z)=I(\hat{G}^1;X^2)+I(\hat{G}^1;Z|X^2)$$
$$=I(\hat{G}^1;Z|X^2)+O(\lambda_s\sigma_X d_X\delta)$$
$$=O(\lambda_s\sigma_X d_X\delta), \tag{24}$$

where the first inequality uses the conditional independence $\hat{G}^1\perp\!\!\!\perp X^2|Z$ and the last equality uses the conditional independence $\hat{G}^1\perp\!\!\!\perp Z|\bar{X}^2=X^2$. This concludes the proof of item 1.

To prove item 2, consider the score function of the form
$$s_{\theta_1}(X_t^1,\hat{Z}^2,\hat{G}^1,t)=\frac{\exp(-t)f(\hat{Z}^2,\hat{G}^1)}{1-\exp(-2t)}=\frac{\exp(-t)f(Z,\hat{G}^1)}{1-\exp(-2t)},$$

and set $t_1:=\frac{1}{2}\log\frac{1}{1-\delta}$, the loss $L_c$ becomes
$$L_m(\theta_1,\phi_1)=\frac{1}{T-t_1}\int_{t_1}^{T}\frac{\exp(-2t)\mathbb{E}\|f(Z,\hat{G}^1)-X^1\|^2}{(1-\exp(-2t))^2}\mathrm{d}t$$
$$=\frac{\tilde{L}_m(\theta_1,\phi_1)}{2(T-t_1)}\left(\frac{1}{e^{2t_1}-1}-\frac{1}{e^{2T}-1}\right)\leq\frac{\tilde{L}_m(\theta_1,\phi_1)(e^{2T}-e^{2t_1})}{2(T-t_1)(e^{2t_1}-1)(e^{2T}-1)}.$$

By Assumption C.3,
$$\tilde{L}_m(\theta_1,\phi_1):=\mathbb{E}\|f(Z,\hat{G}^1)-X^1\|^2=\mathbb{E}\|f(Z,\hat{G}^1)-f(Z,G^1)\|^2$$
$$\leq\lambda_f\lambda_g\|X_{t_0}^1-\bar{X}^1\|^2\leq\lambda_f\lambda_g(\sigma_X^2\delta^2+2)\delta^2 d_X,$$

where the inequality uses the fact that
$$\mathbb{E}\|X_{t_0}^1-\bar{X}^1\|^2=\mathbb{E}\|\alpha^2\bar{X}^1+\alpha N^1+\delta N_{t_0}^1-\bar{X}^1\|^2$$
$$=\delta^4\mathbb{E}\|\bar{X}\|^2+(1-\delta^2)\delta^2 d_X+\delta^2 d_X$$
$$=((\sigma_X^2-1)\delta^2+2)\delta^2 d_X.$$

Plugging this into Eq. 24 yields
$$L_m(\theta_1,\phi_1)\leq\frac{\lambda_f\lambda_g(\sigma_X^2\delta^2+2)\delta^2 d_X(e^{2T}-e^{2t_1})}{2(T-t_1)(e^{2t_1}-1)(e^{2T}-1)}$$
$$=O\left(\frac{\lambda_f\lambda_g\delta d_X}{T}\right).$$

This proves item 2 of the lemma.

## C.3 Proof of Lemma C.6

First, by the equivalence of the forward and reverse process,

$$\mathbb{E}_{X^1,(X^{1,\leftarrow}_t)_t}\exp\left(\frac{1}{2}\int_0^{T-t_1}\|s_\theta(X^{1,\leftarrow}_t,\hat{Z}^2,\hat{G}^1,T-t)-\nabla_x\log p_{T-t}(X^{1,\leftarrow}_t|X^1)\|^2\mathrm{d}t\right)$$

$$=\mathbb{E}_{p_{X^1}}\mathbb{E}_{t,p_{t|0}(\cdot|X^1)}\exp\left(\frac{1}{2}\int_{t_1}^T\|s_\theta(X^1_t,\hat{Z}^2,\hat{G}^1,t)-\nabla_x\log p_t(X^1_t|X^1)\|^2\mathrm{d}t\right) \tag{25}$$

$$=\mathbb{E}_{p_{X^1}}\mathbb{E}_{t,p_{t|0}(\cdot|X^1)}\exp\left(\int_{t_1}^T\frac{1}{2\sigma(t)^4}\left\|\sigma(t)^2 s_\theta(X^1_t,\hat{Z}^2,\hat{G}^1,t)+\sigma(t)N^1_t\right\|^2\mathrm{d}t\right), \tag{26}$$

where the second-to-last equality uses the closed-form formula of the conditional score function of an OU process and $\sigma(t):=\sqrt{1-e^{-2t}}$. Define $\lambda:=2\lambda_\Theta(1+\lambda_z+\lambda_g)$, and by Assumption C.2 again, we have

$$\mathbb{E}\exp\left(\frac{1}{2\sigma(t)^4}\int_{t_1}^T\left\|\sigma(t)^2 s_\theta(X^1_t,\hat{Z}^2,\hat{G}^1,t)+\sigma(t)N^1_t\right\|^2\mathrm{d}t\right)$$

$$\lesssim\mathbb{E}\exp\left(\int_{t_1}^T\frac{\lambda^2(\|X^1_t\|^2+\|X^2_{t_0}\|^2+\|X^1_{t_0}\|^2+t^2)+\sigma(t)^2\|N^1_t\|^2}{\sigma(t)^4}\mathrm{d}t\right)$$

$$\lesssim\mathbb{E}\exp\left(2\lambda^2(2\|\bar{X}^1\|^2+\|\bar{X}^2\|^2)\int_{t_1}^T\frac{\mathrm{d}t}{\sigma(t)^4}+2\lambda^2\int_{t_1}^T\frac{\|N^1_{t_0}\|^2+\|N^2_{t_0}\|^2+\frac{\lambda^2+\sigma(t)^2}{\lambda^2}\|N^1_t\|^2}{\sigma(t)^4}\mathrm{d}t+\int_{t_1}^T\frac{\lambda^2 t^2\mathrm{d}t}{2\sigma(t)^4}\right)$$

$$\lesssim\mathbb{E}\exp\left(\frac{2\lambda^2 T(2\|\bar{X}^1\|^2+\|\bar{X}^2\|^2+(3+\delta)d_X\|\sup_{t_1\leq t\leq T}N^1_{t,i}\|^2)}{\delta^2}+\frac{\lambda^2 T^3}{2\delta^2}\right)<\infty,$$

where the last inequality uses the fact that $\bar{X}$ is sub-gaussian while $\sup_{t_1\leq t\leq T}N_{t,i},\forall 1\leq i\leq d_X$ is sub-gaussian [122, 41] and independent of $\bar{X}$, and the second-to-last inequality uses the bounds

$$\int_{t_1}^T\frac{\mathrm{d}t}{\sigma(t)^4}\leq\frac{T}{\sigma(t_1)^4}=\frac{T}{(1-e^{-2t_1})^2}=\frac{T}{\delta^2}$$

$$\int_{t_1}^T\frac{\mathrm{d}t}{\sigma(t)^2}\leq\frac{T}{\sigma(t_1)^2}=\frac{T}{1-e^{-2t_1}}=\frac{T}{\delta}.$$

## C.4 Extension to correlated view-specific styles

To extend Theorem 4.2 to correlated view-specific styles with $I(G^1;G^2|Z)\leq\epsilon_I$, we can replace every step in the proof of Theorem 4.2 that involves the conditional independence $X^1\perp\!\!\!\perp X^2|Z$ and thus $I(X^1;X^2|Z)=0$ (e.g., Eq. 22 and Eq. 24) with

$$I(X^1;X^2|Z)\leq I(G^1;G^2|Z)\leq\epsilon_I,$$

by data processing inequality. The rest of the proofs follow as before.

## D  Proof of Theorem 4.5

For clarity, we restate the theorem below.

**Theorem D.1.** *Under Assumption 3.5-3.6, for some $t_0$ dependent on $n$ and let $\min\{d_T,d_H\}\to\infty$, and let the positional encoding $PE(\cdot)$ be bounded and linearly independent over $t\in[t_0,T]$. Define $P_M$ to be the projection matrix onto $R(M)$, and $\sigma_i(s)$ to be the $i$-th largest singular value of the operator $s$. Then for some $\lambda_r$, Eq. 12 converges to a critical point $\hat{\theta}:=[\hat{\theta}_Z,\hat{\theta}_G,\hat{U},\hat{V}]$ such that with probability at least $1-O(\frac{1}{n})$: 1) Encodings $\hat{Z}:=P_{\hat{U}}X$ and $\hat{G}:=g(X)$ are $\left(O\left(\frac{d_X^{5/4}\log^{3/4}n}{\sigma_{d_Z}(s^*_Z)n^{1/4}}\right),0\right)$-disentangled; 2) Let $\hat{Z}_{t_0}:=\hat{Z}+\sigma(t_0)N_{t_0}$. Then $(\hat{Z}_{t_0},\hat{G})$ are $O\left(\frac{d_X^{7/4}\log^{9/16}n}{T\min\{\sigma_{d_Z}^{1/2}(s^*_Z),\sigma_{d_G}^{1/2}(s^*_G)\}n^{1/16}}\right)$-editable.*

## D.1   Main proof

To prove the theorem, we need Lemma 4.4 proved in Section D.2 and the following theorems, whose proofs are postponed to Section D.3-D.4. Note that in the proofs, we drop the subscript $n$ and use the population versions of the losses when $n \to \infty$. In subsequent analysis, we use the following balancing loss:

$$L_{b,n} := \mathbb{E}_t \|\tilde{U}^\top \tilde{U} - \mathbb{E}_{\hat{p}_t^n(x)} \tilde{s}(x,t)^\top \tilde{s}(x,t)\|^2 + \lambda_r \mathbb{E}_t \|V^\top V - \mathbb{E}_{\hat{p}_t^n(x)} s_G(g(x),t) s_G(g(x),t)^\top\|^2,$$

where $\tilde{U} := [U,V]$ and $\tilde{s}(x,t) := [s_Z(x,t)^\top, s_G(x,t)^\top]^\top$

**Theorem D.2.** *For the linear subspace model in Definition 4.3 and the objective in Eq. 11 with $n \to \infty$, then any minimizer $(U^*, V^*)$ of Eq. 11 satsify $R(U^*) = R(A_Z)$ and $R(V^*) = R(A_G)$.*

**Theorem D.3.** *Suppose $\min\{n, d_T, d_H\} \to \infty$, and the neural network weights are initialized by standard Gaussians. Further, choose $\text{PE}(\cdot)$ such that $\text{PE}(t)$'s are bounded and linearly independent for all $t \in [t_0, T]$. Then for $\lambda_r = 3$, the system of gradient flow equations in Eq. 12 converges to a critical point $(\hat{U}, \hat{V})$ such that $R(\hat{U}) = R(A_Z), R(\hat{V}) = R(A_G)$.*

Recall that the two branches of the dual encoder network $s_Z^{\theta_Z}$ and $s_G^{\theta_G}$ are defined as

$$s_Z^{\theta_Z}(x,t) =: s_Z(x,t) := \frac{1}{\sqrt{d_H}} \sum_{j=1}^{d_H} \theta_Z^{(2),j} (\theta_Z^{(1),j\top} [x^\top, \text{PE}(t)]^\top)_+, \tag{27}$$

$$s_G^{\theta_G}(g,t) =: s_G(g,t) := \frac{1}{\sqrt{d_H}} \sum_{j=1}^{d_H} \theta_G^{(2),j} (\theta_G^{(1),j\top} [g^\top, \text{PE}(t)]^\top)_+. \tag{28}$$

Further, define the *neural tangent kernels* (NTKs) [123] for the score functions $s_Z$ and $s_G$ as

$$K_Z(x,t,x',t') := J_{\text{vec}(\theta_Z)} s_Z(x,t)^\top J_{\text{vec}(\theta_Z)} s_Z(x',t'),$$

$$K_G(x,t,x',t') := J_{\text{vec}(\theta_G)} s_G(g(x),t)^\top J_{\text{vec}(\theta_G)} s_G(g(x'),t'), \forall (x,t,x',t') \in \mathcal{X} \times [t_0,T] \times \mathcal{X} \times [t_0,T],$$

where $\text{vec}(\theta)$ denotes the flattened version of the parameter $\theta$.

For i.i.d samples $[x^1, \cdots, x^n]$, let the Hilbert space spanned by the NTKs $K_Z(x,t,\cdot,\cdot)$'s and $K_G(x,t,\cdot,\cdot)$'s be $\mathcal{H}_{K_Z}$ and $\mathcal{H}_{K_G}$ respectively. Further, for any $f(x,t) := \int_{t_0}^T \int_{\mathcal{X}} K(x,t,x',t') c(x',t') \mathrm{d}x' \mathrm{d}t' \in \mathcal{H}_K, c(x,t) \in \mathbb{R}^d, \forall x,t$, define the *NTK norm* as

$$\|f\|_K := \langle f, f \rangle_K := \sqrt{\mathbb{E}_{t,t',p_t(x)p_t(x')} c(x,t)^\top K(x,t,x',t') c(x',t)}. \tag{29}$$

Further, define the *subspace score matching losses* as

$$L_Z(U,s_Z) := \mathbb{E}_{t,p_t(x)} \|U s_Z(x,t) - A_Z s_Z^*(x,t)\|^2 =: \mathbb{E}_{t,p_t(x)} \ell_Z(x,t;U,s_Z)$$

$$L_G(V,s_G) := \mathbb{E}_{t,p_t(x)} \|V s_G(g(x),t) - A_G s_G^*(x,t)\|^2 =: \mathbb{E}_{t,p_t(x)} \ell_G(x,t;V,s_G),$$

and their empirical versions as

$$\hat{L}_Z(U,s_Z) := \frac{1}{n} \sum_{i=1}^n \ell_Z(\tilde{x}^i; U, s_Z)$$

$$\hat{L}_G(V,s_G) := \frac{1}{n} \sum_{i=1}^n \ell_G(\tilde{x}^i; V, s_G).$$

We also need the following lemma proved in Section D.9.

**Lemma D.4.** *(generalization error bound) Let $\min\{d_T, d_H\} \to \infty$, and for i.i.d samples $[x^1, \cdots, x^n]$, let the reproducing kernel Hilbert space (RKHS) spanned by the NTKs $K_Z(x,t,\cdot,\cdot)$'s and $K_G(x,t,\cdot,\cdot)$'s be $\mathcal{H}_{K_Z}$ and $\mathcal{H}_{K_G}$ respectively, and denote $K_{Z,(x,t)} := K_Z(x,t,\cdot,\cdot)$. Further, define the function classes*

$\mathcal{S}_Z, \mathcal{U}, \mathcal{S}_G, \mathcal{V}$ as

$$\mathcal{S}_Z := \left\{ f = \sum_{i=1}^{N} c_i K_{Z,\tilde{x}_i} \,\middle|\, \forall N \in \mathbb{N}, \|f\|_K \leq C_Z \sqrt{\frac{\sigma_1(s_Z^*) d_Z^{1/2}}{\lambda_{\min}(K_Z^*)}}, \forall [\tilde{x}_1, \cdots, \tilde{x}_N] \in (\mathcal{X} \times [t_0, T])^N \right\},$$

$$\mathcal{U} := \left\{ U : \|U\|_F \leq C_Z \sqrt{\sigma_1(s_Z^*) d_Z^{1/2}} \right\},$$

$$\mathcal{S}_G := \left\{ f = \sum_{i=1}^{N} c_i K_{G,\tilde{x}_i} \,\middle|\, \forall N \in \mathbb{N}, \|f\|_K \leq C_G \sqrt{\frac{\sigma_1(s_G^*) d_G^{1/2}}{\lambda_{\min}(K_G^*)}}, \forall [\tilde{x}_1, \cdots, \tilde{x}_N] \in (\mathcal{X} \times [t_0, T])^N \right\},$$

$$\mathcal{V} := \left\{ V : \|V\|_F \leq C_G \sqrt{\sigma_1(s_G^*) d_G^{1/2}} \right\},$$

where $\sigma_i(A)$ is the $i$-th largest singular value of the operator $A$. Then with probability at least $1 - O(\frac{1}{n})$,

$$L_Z(\hat{U}, \hat{s}_Z) \leq \min_{(U, s_Z) \in \mathcal{U} \times \mathcal{S}_Z} L_Z(U, s_Z) + O\left( \sqrt{\frac{d_X^5 \log^3 n}{n}} \right),$$

$$L_G(\hat{V}, \hat{s}_G) \leq \min_{(V, s_G) \in \mathcal{V} \times \mathcal{S}_G} L_G(V, s_G) + O\left( \sqrt{\frac{d_X^5 \log^3 n}{n}} \right).$$

Now we are ready to prove the Theorem 4.5. To this end, we first prove the following statements:

1. The population score matching loss satisfies: $L_0(\hat{\theta}) = O\left( \sqrt{\frac{d_X^5 \log^3 n}{n}} \right)$;

2. The subspace recovery errors are bounded by:

$$\|P_{\hat{U}} - A_Z A_Z^\top\|_F^2 = O\left( \sqrt{\frac{d_X^5 \log^3 n}{\sigma_{d_Z}^4(s_Z^*) n}} \right), \quad \|P_{\hat{V}} - A_G A_G^\top\|_F^2 = O\left( \sqrt{\frac{d_X^5 \log^3 n}{\sigma_{d_G}^4(s_G^*) n}} \right).$$

First, by the universal approximation theorem of neural networks (e.g. [124]) and the Lipschitzness of the true score functions, for $C_Z, C_G$ large enough, $s_Z^* \in \mathcal{S}_Z$ and $s_G^* \in \mathcal{S}_G$ and therefore

$$\min_{(U, s_Z) \in \mathcal{U} \times \mathcal{S}_Z} L_Z(U, s_Z) = L_Z(A_Z, s_Z^*) = 0,$$

$$\min_{(V, s_G) \in \mathcal{V} \times \mathcal{S}_G} L_G(V, s_G) = L_G(A_G, s_G^*) = 0.$$

As a result, we have with probability at least $1 - O(\frac{1}{n})$,

$$L_Z(\hat{U}, \hat{s}_Z) \leq \epsilon(n), \quad L_G(\hat{V}, \hat{s}_G) \leq \epsilon(n),$$

where $\epsilon(n) = O\left( \sqrt{\frac{d_X^5 \log^3 n}{n}} \right)$. Therefore, we prove item 1 by noticing

$$L_0(\hat{U}, \hat{V}, \hat{s}_Z, \hat{s}_G) \leq 2 L_Z(\hat{U}, \hat{s}_Z) + 2 L_G(\hat{V}, \hat{s}_G) \leq 2\epsilon(n) = O\left( \sqrt{\frac{d_X^5 \log^3 n}{n}} \right).$$

We proceed to prove item 2. Let $Q_Z \in \mathbb{R}^{d_X \times d_Z}, Q_G \in \mathbb{R}^{d_X \times d_G}$ be orthogonal matrices such that

$$P_{R(\hat{U})} = Q_Z Q_Z^\top, P_{R(\hat{V})} = Q_G Q_G^\top,$$

then we have $\hat{U}\hat{s}_Z := Q_Z Q_Z^\top \hat{U}\hat{s}_Z$ and $\hat{V}\hat{s}_G := Q_G Q_G^\top \hat{V}\hat{s}_G$,

Then applying Lemma 7 of [41] yields with probability at least $1 - 4\delta$,

$$\|P_{R(\hat{U})} - A_Z A_Z^\top\|^2 \leq \frac{\epsilon(n)}{\sigma_{d_Z}(s_Z^*)^2}$$

$$\|P_{R(\hat{V})} - A_G A_G^\top\|^2 \leq \frac{\epsilon(n)}{\sigma_{d_G}(s_G^*)^2}.$$

To prove $(\epsilon,\nu)$-disentanglement, define $Z_\| := f_1(Z) := P_U A_Z Z$ and $Z_\perp := f_2(G) := P_U A_G G$. Then by definition,

$$\hat{Z} = Z_\| + Z_\perp = f_1(Z) + f_2(G).$$

Since $Z_\|$ is a function of $Z$ and $\hat{G} = G$, we have the independence relations $Z_\| \perp\!\!\!\perp \hat{G}$ and $\hat{Z} \perp\!\!\!\perp \hat{G}|Z_\perp$, and therefore by data processing inequality,

$$
\begin{aligned}
I(\hat{Z};\hat{G}) &\leq I(\hat{Z};Z_\perp) \\
&= \mathbb{E}_{p_{\hat{Z}Z_\perp}(z,y)} \log \frac{p_{\hat{Z}Z_\perp}(z,y)}{p_{\hat{Z}}(z)p_{Z_\perp}(y)} \\
&= \mathbb{E}_{p_{\hat{Z}Z_\perp}(z,y)} \log \frac{p_{Z_\|}(z-y)}{p_{\hat{Z}}(z)} \\
&= \mathbb{E}_{p_{\hat{Z}Z_\perp}(z,y)} \log \frac{p_{Z_\|}(z)}{p_{\hat{Z}}(z)} + \mathbb{E}_{p_{\hat{Z}Z_\perp}(z,y)} \log \frac{p_{Z_\|}(z-y)}{p_{Z_\|}(z)} \\
&= \mathbb{E}_{p_{\hat{Z}}(z)} \log \frac{p_{Z_\|}(z)}{p_{\hat{Z}}(z)} + \mathbb{E}_{p_{\hat{Z}Z_\perp}(z,y)} \log \frac{p_{Z_\|}(z-y)}{p_{Z_\|}(z)} \\
&= -D_{\mathrm{KL}}(p_{\hat{Z}}||p_{Z_\|}) + \mathbb{E}_{p_{\hat{Z}Z_\perp}(z,z-y)} \log \frac{p_{Z_\|}(z-y)}{p_{Z_\|}(z)} \\
&\leq \mathbb{E}_{p_{\hat{Z}Z_\|}(z,z-y)} \log \frac{p_{Z_\|}(z-y)}{p_{Z_\|}(z)} = \mathbb{E}_{p_{Z_\perp}(y)p_{Z_\|}(z-y)} \log \frac{p_{Z_\|}(z-y)}{p_{Z_\|}(z)} \\
&= O\left( \mathbb{E}_{p_{Z_\|}(z-y)p_{Z_\perp}(y)} \|z-y\|\|y\| + \mathbb{E}_{p_{Z_\perp}(y)} \|y\|^2 \right) = O(\|P_U A_G\| + \|P_U A_G\|^2),
\end{aligned}
$$

where the last inequality uses the nonnegativity of KL divergence and the second-to-last equality combines Lemma B.6 with the fact that $Z_\| = P_U A_Z Z$ is a 1-Lipschitz function of $Z$ and therefore its score function along the column space of $P_U A_Z$ is $\lambda_s$-Lipschitz. To upper bound $\|P_U A_G\|^2$, we utilize item 2 by noticing that

$$\|P_U A_G\|^2 = \frac{1}{2}\left(\|P_U - A_Z A_Z^\top\|^2 + \mathrm{rank}(U) - d_Z\right) \leq \frac{\epsilon(n)}{2\sigma_{d_Z}(s_Z^*)^2},$$

where the last inequality uses the fact that $\mathrm{rank}(U) \leq d_Z$. As a result,

$$I(\hat{Z};\hat{G}) = O\left(\frac{\sqrt{\epsilon(n)}}{\sigma_{d_Z}(s_Z^*)}\right) = O\left(\frac{d_X^{5/4} \log^{3/4} n}{\sigma_{d_Z}(s_Z^*) n^{1/4}}\right).$$

Further, we would like to bound the mutual information gap $I(\hat{Z},\hat{G};X) - I(Z,G;X)$ by finding an estimator of the sample $X$ from $(\hat{Z},\hat{G})$. To this end, we show that $[P_U, P_V]$ is invertible with high probability. This is the case since by item 2,

$$\|P_{\hat{U}} + P_{\hat{V}} - I_{d_X}\| \leq \|P_{\hat{U}} - A_Z A_Z^\top\| + \|P_{\hat{V}} - A_G A_G^\top\| = O\left(\frac{\log^{3/4} n}{n^{1/4}}\right)$$

with high probability. Therefore, by matrix perturbation inequality,

$$\sigma_{d_X}([P_{\hat{U}}, P_{\hat{V}}]) \geq 1 - O\left(\frac{\log^{3/4} n}{n^{1/4}}\right) > 0$$

for sufficiently large $n$. As a result, we conclude there exists an $\hat{f}(P_{\hat{U}} X, P_{\hat{V}} X) := X$ that can reconstruct $X$ and therefore preserve its information perfectly. Therefore, $(\hat{Z},\hat{G})$ are $\left(O\left(\frac{d_X^{5/4} \log^{3/4} n}{\sigma_{d_Z}(s_Z^*) n^{1/4}}\right), 0\right)$-disentangled.

To prove $\epsilon$-editability, choose $t_0 := \frac{1}{2}\log\frac{1}{1-\delta^2}$ and $t_1 := \frac{1}{2}\log\frac{1}{1-\delta}$ for some $\delta > 0$ and let $q(\cdot|\hat{Z}_{t_0},\hat{G})$ be the pdf of the following estimated reverse diffusion process at time $T - t_1$:

$$d\hat{X}_t^\leftarrow = [\hat{X}_t^\leftarrow + 2s_{\hat{\theta}}(\hat{X}_{k\eta}^\leftarrow, \hat{Z}_{t_0}, \hat{G}, T - k\eta)]dt + \sqrt{2}dB_t^\leftarrow, \hat{X}_0^\leftarrow \sim p_T, t \in [k\eta, (k+1)\eta],$$

where

$$s_{\hat{\theta}}(x,z,g,t) := \frac{e^{-t}x - P_{\hat{U}}z - P_{\hat{V}}g}{\sigma(t)^2}.$$ (30)

For generated samples $\hat{X} \sim q(\cdot|\hat{Z}_{t_0}, \hat{G})$ and $\hat{X}' \sim q(\cdot|\hat{Z}_{t_0}, \hat{G}')$, where $\hat{G}' \sim p_G$ is an i.i.d sample of $\hat{G}$, we have

$$\mathbb{E}_{pz}d_{\mathrm{TV}}(p_{X_{t_1}|Z}, p_{\hat{X}'|Z}) \leq \underbrace{\mathbb{E}_{pz}d_{\mathrm{TV}}(p_{X_{t_1}|Z}, p_{\hat{X}|Z})}_{(i)} + \underbrace{\mathbb{E}_{pz}d_{\mathrm{TV}}(p_{\hat{X}|Z}, p_{\hat{X}'|Z})}_{(ii)}.$$

To bound $(i)$, notice that the score estimation error of $s_{\theta^*}$ is

$$
\begin{aligned}
L_c &:= \mathbb{E}_{t,\hat{Z}_{t_0},\hat{G},X_t} \left\| s_{\hat{\theta}}(X_t, \hat{Z}_{t_0}, \hat{G}, t) + \frac{N_t}{\sigma(t)^2} \right\|^2 \\
&= \frac{1}{T-t_1} \int_{t_1}^T \frac{e^{-2t}\mathbb{E}\left\| \hat{Z}_{t_0} + \hat{G} - X \right\|^2}{\sigma(t)^4} \mathrm{d}t \\
&= \frac{(e^{2T} - e^{2t_1})}{2(T-t_1)(e^{2t_1}-1)(e^{2T}-1)} \mathbb{E}\left\| P_U \hat{Z}_{t_0} + P_V \hat{G} - X \right\|^2 \\
&= O\left( \frac{d_X}{\delta T} \mathbb{E}\left\| P_U \hat{Z}_{t_0} + P_V \hat{G} - X \right\|^2 \right) \\
&= O\left( \frac{d_X^{7/2}\log^{9/8}n}{T\min\{\sigma_{d_Z}^2(s_Z^*),\sigma_{d_G}^2(s_G^*)\}n^{7/16}} + \frac{d_X}{Tn^{1/16}} \right),
\end{aligned}
$$

where the last equality uses $\delta := \frac{\log^{3/8}n}{n^{1/16}}$ and the fact that

$$
\begin{aligned}
&\mathbb{E}\left\| P_U \hat{Z}_{t_0} + P_V \hat{G} - X \right\|^2 \\
&\leq 2\mathbb{E}\left\| (P_U A_Z A_Z^\top - A_Z A_Z^\top)X \right\|^2 + 2\mathbb{E}\left\| (P_V A_G A_G^\top - A_G A_G^\top)X \right\|^2 + 2\sigma(t_0)^2 \\
&= O\left( \frac{d_X^{5/2}\log^{3/2}n}{\min\{\sigma_{d_Z}^2(s_Z^*),\sigma_{d_G}^2(s_G^*)\}n^{1/2}} + \frac{\log^{3/8}n}{n^{1/8}} \right).
\end{aligned}
$$

Further, using an argument similar to Section C.3, we can check that the Novikov's condition holds for $s_{\hat{\theta}}$, and using an argument similar to Section C.2, we can apply Girsanov's theorem to show that

$$(i) = O\left( \frac{d_X^{7/4}\log^{9/16}n}{T\min\{\sigma_{d_Z}(s_Z^*),\sigma_{d_G}(s_G^*)\}n^{7/32}} + \frac{d_X}{Tn^{1/32}} \right).$$

Further, notice that

$$
\begin{aligned}
&I(\hat{Z}_{t_0}; \hat{G}|Z) \\
&= \mathbb{E}_{p_{\hat{Z}_{t_0}G|Z}} \log \frac{p_{\hat{Z}_{t_0}|G,Z}(\hat{z}|g,z)}{p_{\hat{Z}_{t_0}|Z}(\hat{z}|z)} \\
&= \mathbb{E}_{p_{\hat{Z}_{t_0}G|Z}} \log \frac{p_{\delta N_{t_0}}(\hat{z} - f_1(z) - f_2(g))}{p_{f_2(g)+\delta N_{t_0}}(\hat{z} - f_1(z))} \\
&= -D_{\mathrm{KL}}(p_{\delta N_{t_0}} \| p_{f_2(g)+\delta N_{t_0}}) + \mathbb{E}_{p_{\hat{Z}G|Z}} \log \frac{p_{\delta N_{t_0}}(\hat{z} - f_1(z) - f_2(g))}{p_{\delta N_{t_0}}(\hat{z} - f_1(z))} \\
&\leq \frac{1}{\delta^2} O\left( \mathbb{E}_{p_{\hat{Z}G|Z}} \|\hat{z} - f_1(z)\| \|f_2(g)\| + \mathbb{E}_{p_G(g)} \|f_2(g)\|^2 \right) \\
&= O\left( \frac{1}{\delta^2} \|P_U A_G\| \right) = O\left( \frac{d_X^{5/4}\log^{3/4}n}{\delta^2\sigma_{d_Z}(s_Z^*)n^{1/4}} \right) = O\left( \frac{d_X^{5/4}}{\sigma_{d_Z}(s_Z^*)n^{1/8}} \right),
\end{aligned}
$$

where the inequality uses Lemma B.6. By Pinsker's inequality and data processing inequality similar to the proof of Theorem 4.2, we have

$$d_{\mathrm{TV}}(p_{X_{t_1}|Z},p_{\hat{X}'|Z})=O\left(\frac{d_X^{5/8}}{\sigma_{d_Z}(s_Z^*)^{1/2}n^{1/16}}\right).$$

Combining the bounds on $(i)(ii)$, we have $\hat{Z}_{t_0}$ and $\hat{G}$ are $O\left(\frac{d_X^{7/4}\log^{9/16}n}{T\min\{\sigma_{d_Z}^{1/2}(s_Z^*),\sigma_{d_G}^{1/2}(s_G^*)\}n^{1/16}}\right)$-editable.

## D.2   Proof of Lemma 4.4

Let $J_x f(x)\in\mathbb{R}^{d_1\times d_2}$ denote the Jacobian matrix of vector function $f:\mathbb{R}^{d_1}\mapsto\mathbb{R}^{d_2}$ with respect to vector $x\in\mathbb{R}^{d_1}$. Use the change of variables formula for pdf on $X_t=A_ZZ_t+A_GG$, we have

$$p_t(x)=p_{Z_tG_t}(z(x),g(x))|\det(J_x[z(x)^\top,g(x)^\top]^\top)|$$
$$=p_{Z_tG_t}(z(x),g(x))|\det(A)|=p_{Z_tG_t}(z(x),g(x)),$$

where the last equality uses the fact that the matrix $A=[A_Z,A_G]$ is orthogonal and thus $|\det(A)|=1$. Further, by Definition 4.3, $Z_t\perp\!\!\!\perp G_t,\forall t\geq 0$ since $Z\perp\!\!\!\perp G$ and $A_Z^\top N_t\perp\!\!\!\perp A_G^\top N_t$ due to the orthogonality of $R(A_Z)$ and $R(A_G)$. Therefore,

$$p_t(x)=p_{Z_tG_t}(z(x),g(x))=p_{Z_t}(z(x))p_{G_t}(g(x)).$$

Plugging this into the score of $p_t(X)$ and applying the chain rule yields

$$\nabla_x\log p_t(x)=J_xz(x)\nabla_z\log p_Z(z(x))+J_xg(x)\nabla_g\log p_G(g(x))$$
$$=A_Z\nabla_z\log p_Z(z(x))+A_G\nabla_g\log p_G(g(x)).$$

## D.3   Proof of Theorem D.2

Let $s_Z^*(x):=\nabla_z\log p_{Z_t}(z(x))$ and $s_G^*(g):=\nabla_g\log p_{G_t}(g)$ and $P_A$ be the projection matrix onto $R(A)$, then for any $(s_z,s_G,U,V)$,

$$L_0(s_Z,s_G,U,V)$$
$$:=\mathbb{E}_{t,p_t(x)}\|(P_{A_Z}(Us_Z(x,t)+Vs_G(g(x),t)-A_Zs_Z^*(x,t))+(P_{A_G}(Us_Z(x,t)+Vs_G(g(x,t)))-A_Gs_G^*(g(x,t)))\|^2$$
$$=\mathbb{E}_{t,p_t(x,t)}[\|P_{A_Z}(Us_Z(x,t)+Vs_G(g(x),t)-A_Zs_Z^*(x,t)\|^2+$$
$$\|P_{A_G}(Us_Z(x,t)+Vs_G(g(x),t)-A_Gs_G^*(g(x),t)\|^2]\geq 0=L_0(s_Z^*,s_G^*,A_Z,A_G).\quad(31)$$

To analyze the equality condition, notice by the fact that minimizing over a larger set leads to smaller loss and the independence between $Z_t$ and $G_t$,

$$\mathbb{E}_{t,p_t(x)}\|P_{A_Z}(Us_Z(x,t)+Vs_G(g(x),t)-A_Zs_Z^*(x,t)\|^2$$
$$\geq\mathbb{E}_{t,p_t(x)}\|\mathbb{E}_{p_t(x)}[A_Zs_Z^*(x,t)|Us_Z(x,t)+Vs_G(g(x),t)]-A_Zs_Z^*(x,t)\|^2$$
$$\geq\mathbb{E}_{t,p_t(x)}\|\mathbb{E}_{p_t(x)}[A_Zs_Z^*(x,t)|Us_Z(x,t),Vs_G(g(x),t)]-A_Zs_Z^*(x,t)\|^2$$
$$=\mathbb{E}_{t,p_t(x)}\|\mathbb{E}_{p_t(x)}[A_Zs_Z^*(x,t)|Us_Z(x,t)]-A_Zs_Z^*(x,t)\|^2,\quad(32)$$

with equality if and only if

$$\mathbb{E}_{p_t(x)}(s_Z^*(x,t)-P_{A_Z}Vs_G(g(x),t))s_G(g(x),t)^\top=0,\forall t\in[0,T]$$
$$\Rightarrow P_{A_Z}Vs_G(g(x),t)=0,\text{a.s.},\forall x,t.$$

by the orthogonality principle. As a result, the equality of Eq. 32 is achieved if and only if

$$P_{A_Z}Us_Z(x,t)=A_Zs_Z^*(x,t),\quad(33)$$
$$P_{A_Z}Vs_G(g(x),t)=0,\quad(34)$$
$$P_{A_G}Us_Z(x,t)+Vs_G(g(x),t)=A_Gs_G^*(x),\text{a.s.},\forall x,t.\quad(35)$$

Now, we turn our attention to the regularizer $L_r$ and notice that for any $(s_Z,U)$ that satisfies Eq. 33-35,

$$L_r(s_G,V) \geq \mathbb{E}_{t,p_t(x)}\|Vs_G(g(x),t) - \nabla_x\log p_t(x)\|^2 = \mathbb{E}_{t,p_t(x)}\|Us_Z(x,t)\|^2$$
$$\geq \mathbb{E}_{t,p_t(x)}\|P_{A_Z}Us_Z(x,t)\|^2 = \mathbb{E}_{t,p_t(x)}\|A_Z s_Z^*(x,t)\|^2,$$

where both equalities are achieved if and only if

$$V^\top V = \mathbb{E}s_G(g,t)s_G(g,t)^\top,$$
$$\|P_{A_G}Us_Z(x,t)\| = 0, \text{a.s.}, \forall x,t.$$

Finally, we shall show that for any $(A_Z,A_G,s_Z^*,s_G^*)$, there exists some optimal solution $(U,V,s_Z,s_G)$ such that

$$Us_Z = A_Z s_Z^*, Vs_G = A_G s_G^*, \tag{36}$$
$$V^\top V = \mathbb{E}s_G(g,t)s_G(g,t)^\top, L_b(s_Z,s_G,U,V) = 0. \tag{37}$$

To this end, consider the SVD of the operator $A_Z s_Z^*(x,t) + A_G s_G^*(g(x),t)$ as

$$\forall x,t, \quad A_Z s_Z^*(x,t) + A_G s_G^*(g(x),t) = [\Phi_Z,\Phi_G]\begin{bmatrix} \Sigma_Z & \mathbf{0}_{d_Z\times d_G} \\ \mathbf{0}_{d_G\times d_Z} & \Sigma_G \end{bmatrix}\begin{bmatrix} f_Z(x,t) \\ f_G(g(x),t) \end{bmatrix}$$

$$\text{s.t.} \quad \Sigma_Z = \text{diag}(\sigma_1(s_Z^*),\cdots,\sigma_{d_Z}(s_Z^*)), \Sigma_Z = \text{diag}(\sigma_1(s_G^*),\cdots,\sigma_{d_G}(s_G^*)),$$

$$\begin{bmatrix} \Phi_Z^\top \\ \Phi_G^\top \end{bmatrix}[\Phi_Z,\Phi_G] = \mathbb{E}\begin{bmatrix} f_Z(x,t) \\ f_G(g(x),t) \end{bmatrix}[f_Z(x,t)^\top, f_G(g(x),t)^\top] = I_{d_X}$$

Set $U := \Phi_Z\Sigma_Z^{1/2}, s_Z^*(x,t) = \Sigma_Z^{1/2}f_Z(x,t), V := \Phi_G\Sigma_G^{1/2}, s_G^*(x,t) = \Sigma_G^{1/2}f_G(x,t)$, we have $(U,V,s_Z,s_G)$ satisfies Eq. 36. Further, notice that

$$U^\top U = \Sigma_Z^{1/2}\Phi_Z^\top\Phi_Z\Sigma_Z^{1/2} = \Sigma_Z = \Sigma_Z^{1/2}\mathbb{E}f_Z(x,t)f_Z(x,t)^\top\Sigma_Z^{1/2} = \mathbb{E}s_Z(x,t)s_Z(x,t)^\top,$$
$$U^\top V = \Sigma_Z^{1/2}\Phi_Z^\top\Phi_G\Sigma_G^{1/2} = 0 = \Sigma_Z^{1/2}\mathbb{E}f_Z(x,t)f_G(x,t)^\top\Sigma_G^{1/2} = \mathbb{E}s_Z(x,t)s_G(g(x),t)^\top,$$
$$V^\top V = \Sigma_G^{1/2}\Phi_G^\top\Phi_G\Sigma_G^{1/2} = \Sigma_G = \Sigma_G^{1/2}\mathbb{E}f_G(g,t)f_G(g,t)^\top\Sigma_G^{1/2} = \mathbb{E}s_G(g,t)s_G(g,t)^\top.$$

Therefore, $(U,V,s_Z,s_G)$ also satisfies Eq. 37.

### D.4 Proof of Theorem D.3

Define matrices

$$W(x,t) := \begin{bmatrix} U & V \\ s_Z(x,t)^\top & s_G(g(x),t)^\top \end{bmatrix} =: \begin{bmatrix} \tilde{U} \\ s(x,t)^\top \end{bmatrix},$$

$$\tilde{W}(x,t) := \begin{bmatrix} W(x,t) \\ \mathbf{0}_{(d_X+1)\times d_Z}, \sqrt{\frac{\lambda_r}{3}}W_G(x,t) \end{bmatrix},$$

$$W^*(x,t) := \begin{bmatrix} A_Z & A_G \\ s_Z^*(x,t)^\top & s_G^*(g(x),t)^\top \end{bmatrix} =: \begin{bmatrix} A \\ s^*(x,t)^\top \end{bmatrix},$$

$$\tilde{W}^*(x,t) := \begin{bmatrix} W^*(x,t) \\ \mathbf{0}_{(d_X+1)\times d_Z}, \sqrt{\frac{\lambda_r}{3}}W_G^*(x,t) \end{bmatrix},$$

$$N(x,t) := W(x,t)W(x,t)^\top, N^*(x,t) := W^*(x,t)W^*(x,t)^\top,$$
$$\tilde{N}(x,t) := \tilde{W}(x,t)\tilde{W}(x,t)^\top, \tilde{N}^*(x,t) := \tilde{W}^*(x,t)\tilde{W}^*(x,t)^\top.$$

Further, define the *direction of improvement* of $W(x,t)$'s and $\tilde{W}(x,t)$'s respectively as

$$\Delta(x,t) := W(x,t) - W^*(x,t)R,$$
$$\tilde{\Delta}(x,t) := \tilde{W}(x,t) - \tilde{W}^*(x,t)R,$$
$$R := \underset{R:R^\top R=RR^\top=I_{d_X}}{\text{argmin}} \mathbb{E}_{t,p_t(x)}[\|\tilde{W}(x,t) - \tilde{W}^*(x,t)R\|^2]$$
$$:= \underset{R:R^\top R=RR^\top=I_{d_X}}{\text{argmin}} \mathbb{E}_{t,p_t(x)}\left[\|W(x,t) - W^*(x,t)R\|^2 + \frac{\lambda_r}{3}\|W_G(x,t) - W_G^*(x,t)R_G\|^2\right],$$

where for any matrices $M$, define $M_Z$ to be its first $d_Z$ columns and $M_G$ to be its $(d_Z+1)$-th through $d_X$-th columns.

For any set of matrices $\{C(y,t)\}_{y\in\mathcal{Y},t\in[0,T]}$ and probability measures $q_t(y)$'s, define random matrix $\mathbf{C}$ as

$$\mathbf{C}=C(y,t) \quad \text{w.p.} \quad p(t)q_t(y),$$

where $p(t)$ is some fixed distribution of the diffusion time $t$. Further, define a blockwise representation of $\mathbf{C}$ as

$$\mathbf{C}=\begin{bmatrix} \mathbf{C}_0 & \mathbf{c}_2 \\ \mathbf{c}_1 & c_{11} \end{bmatrix},$$

where $\mathbf{C}_0$ is $\mathbf{C}$ deleting the last row and column.

For a pair of random matrices $(\mathbf{C}_1,\mathbf{C}_2)$, define

$$\mathbf{C}_1\mathbf{C}_2=C_1(y,t)C_2(y',t) \quad \text{w.p.} \quad p(t)q_t(y)q_t(y').$$

Next, let $[\mathbf{A},\mathbf{B}]_\mathcal{K}$ denote the *bilinear form* between random matrices $(\mathbf{A},\mathbf{B})$ weighted by the operator $\mathcal{K}$, and $\langle\mathbf{A},\mathbf{B}\rangle_\mathcal{K}=[\mathbf{A},\mathbf{B}]_\mathcal{K}$ be an *inner product* between random matrices $\mathbf{A}$ and $\mathbf{B}$ if $\mathcal{K}$ is positive definite. Then we define the following bilinear forms between random matrices $(\mathbf{A},\mathbf{B})$ with weight operators $\mathcal{I}$, $\mathcal{G}$ and $\mathcal{H}_0$ respectively as

$$\langle\mathbf{A},\mathbf{B}\rangle_\mathcal{I}:=\mathbb{E}_{t,q_t(y)}\langle A(y,t),B(y,t)\rangle$$
$$[\mathbf{A},\mathbf{B}]_\mathcal{G}:=\mathbb{E}_{t,q_t(y)}[\langle A_0(y,t),B_0(y,t)\rangle+a_{11}(y,t)b_{11}(y,t)-$$
$$\langle a_1(y,t),b_1(y,t)\rangle-\langle a_2(y,t),b_2(y,t)\rangle],$$
$$[\mathbf{A},\mathbf{B}]_{\mathcal{H}_0}:=\mathbb{E}_{t,q_t(y)}[\langle a_1(y,t),b_1(y,t)\rangle+\langle a_2(y,t),b_2(y,t)\rangle].$$

It can be verified that $\langle\cdot,\cdot\rangle_\mathcal{I}$ is indeed an inner product satisfying properties such as conjugate symmetry, linearity in the first argument and positive definiteness, and therefore we can define the norm $\|\mathbf{A}\|_\mathcal{I}:=\langle\mathbf{A},\mathbf{A}\rangle_\mathcal{I}$. In addition, it can be checked that $[\cdot,\cdot]_\mathcal{G}$ and $[\cdot,\cdot]_{\mathcal{H}_0}$ are conjugate symmetric and linear in the first argument. An important relation we use repeatedly later is the fact that

$$2[\mathbf{C},\mathbf{C}]_{\mathcal{H}_0}+[\mathbf{C},\mathbf{C}]_\mathcal{G}=[\mathbf{C},\mathbf{C}]_{2\mathcal{H}_0+\mathcal{G}}=\|\mathbf{C}\|_\mathcal{I}^2, \tag{38}$$

since

$$[\mathbf{C},\mathbf{C}]_{2\mathcal{H}_0+\mathcal{G}}=2(\|\mathbf{c}_1\|_\mathcal{I}^2+\|\mathbf{c}_2\|_\mathcal{I}^2)+\|\mathbf{C}_0\|_\mathcal{I}^2+\|c_{11}\|_\mathcal{I}^2-\|\mathbf{c}_1\|_\mathcal{I}^2-\|\mathbf{c}_2\|_\mathcal{I}^2=\|\mathbf{C}\|_\mathcal{I}^2.$$

Under these definitions, we prove the following Lemma in Section D.5.

**Lemma D.5.** *Let $\tilde{L}^{\lambda_r}(\mathbf{W}):=L^{\lambda_r}(\theta)$ with $\lambda_r=3$, and $\mathbf{W}$ be an approximate critical point of $\tilde{L}(\mathbf{W})$ so that $\langle\nabla\tilde{L}(\mathbf{W}),\mathbf{\Delta}'\rangle_\mathcal{I}\leq\epsilon\|\mathbf{\Delta}'\|_\mathcal{I}$ for any random matrix $\mathbf{\Delta}'$. Then the following holds*

$$[\mathbf{\Delta},\mathbf{\Delta}]_{\nabla^2\tilde{L}^{\lambda_r}(\mathbf{W})}\leq\|\tilde{\mathbf{\Delta}}\tilde{\mathbf{\Delta}}^\top\|_\mathcal{I}^2-3\|\tilde{\mathbf{N}}-\tilde{\mathbf{N}}^*\|_\mathcal{I}^2+\epsilon\|\mathbf{\Delta}\|_\mathcal{I}, \tag{39}$$

*where $\nabla^2 f(\mathbf{W})$ denotes the Hessian operator of the functional $f$ at $\mathbf{W}$.*

To proceed, we use the following lemma for random matrices analogous to Lemma 40 and 41 in [125] and defer its proof to Section D.6.

**Lemma D.6.** *If $\mathbb{E}_{p(\mathbf{X})}[U(\mathbf{X})^\top Y(\mathbf{X})]$ is a positive semi-definite (PSD) matrix, then for independent, identically distributed random variables $\mathbf{X},\mathbf{X}'$,*

$$\mathbb{E}\|U(\mathbf{X})U(\mathbf{X}')^\top-Y(\mathbf{X})Y(\mathbf{X}')^\top\|^2\geq$$
$$\max\left\{\frac{1}{2}\mathbb{E}\|(U(\mathbf{X})-Y(\mathbf{X}))(U(\mathbf{X}')-Y(\mathbf{X}'))^\top\|^2,2(\sqrt{2}-1)\mathbb{E}\|(U(\mathbf{X})-Y(\mathbf{X}))U(\mathbf{X}')^\top\|^2\right\}.$$

Further, we prove in Section D.7 the following lemma showing that gradient descent converges to a local optimum of the objective $\tilde{L}$.

**Lemma D.7.** *The gradient flow equation Eq. 12 converges in probability to a solution $(\hat{U},\hat{V},\hat{\theta}_Z,\hat{\theta}_G)$ such that for some $\epsilon>0$ and random matrix $\mathbf{\Delta}'(t)$:*

$$\langle\nabla\tilde{L}^{\lambda_r}(\mathbf{W}),\mathbf{\Delta}'\rangle_\mathcal{I}\leq\epsilon,$$
$$[\mathbf{\Delta}',\mathbf{\Delta}']_{\nabla^2\tilde{L}^{\lambda_r}}\geq0.$$

To prove the theorem, first consider the singular value decomposition $\mathbb{E}_{t,p_t(x)}\tilde{W}^*(x,t)^\top\tilde{W}(x,t)=:\Phi\Sigma f^\top$, and by the definition of the direction of improvement,

$$
\begin{aligned}
R:=&\operatorname*{argmin}_{RR^\top=R^\top R=I_{d_X}}\|\tilde{\mathbf{W}}-\tilde{\mathbf{W}}^*R\|_{\mathcal{I}}^2\\
=&\operatorname*{argmax}_{RR^\top=R^\top R=I_{d_X}}\langle\tilde{\mathbf{W}},\tilde{\mathbf{W}}^*R\rangle_{\mathcal{I}}=\operatorname*{argmax}_{RR^\top=R^\top R=I_{d_X}}\langle\mathbb{E}_{t,p_t(x)}\tilde{W}^*(x,t)^\top\tilde{W}(x,t),R\rangle\\
=&\operatorname*{argmax}_{RR^\top=R^\top R=I_{d_X}}\langle\Sigma,\Phi^\top Rf\rangle=\Phi f^\top,
\end{aligned}
$$

where the last equality holds since $R':=\Phi^\top Rf$ is orthogonal, $|R'_{ii}|\le 1$ and

$$
\langle\Sigma,R'\rangle\le\sum_{i=1}^{d_X}\Sigma_{ii},
$$

with equality iff $R'=I_{d_X}$. As a result,

$$
\mathbb{E}_{t,p_t(x)}\tilde{W}^\top(x,t)\tilde{W}^*(x,t)R=f\Sigma f^\top=\mathbb{E}_{p_t(x),t}R^\top\tilde{W}^{*\top}(x,t)\tilde{W}(x,t)
$$

is PSD. Applying Lemma D.6 on $U(\mathbf{X})=\tilde{\mathbf{W}}^*R$ and $Y(\mathbf{X})=\tilde{\mathbf{W}}$ yields

$$
\begin{aligned}
&\|\tilde{\boldsymbol{\Delta}}\tilde{\boldsymbol{\Delta}}^\top\|_{\mathcal{I}}^2-3\|\tilde{\mathbf{N}}-\tilde{\mathbf{N}}^*\|_{\mathcal{I}}^2\\
&\le 2\|\tilde{\mathbf{N}}-\tilde{\mathbf{N}}^*\|_{\mathcal{I}}^2-3\|\tilde{\mathbf{N}}-\tilde{\mathbf{N}}^*\|_{\mathcal{I}}^2=-\|\tilde{\mathbf{N}}-\tilde{\mathbf{N}}^*\|_{\mathcal{I}}^2\le -2(\sqrt{2}-1)\|\tilde{\boldsymbol{\Delta}}\|_{\mathcal{I}}^2\le -0.8\|\tilde{\boldsymbol{\Delta}}\|_{\mathcal{I}}^2.
\end{aligned}
$$

Combining this with Lemma D.5 yields

$$
[\boldsymbol{\Delta},\boldsymbol{\Delta}]_{\nabla^2\tilde{L}^{\lambda_r}}\le -0.8\|\tilde{\boldsymbol{\Delta}}\|_{\mathcal{I}}^2,
$$

Therefore, we have the LHS to be positive only if $\|\tilde{\boldsymbol{\Delta}}\|_{\mathcal{I}}=0$. Set $\epsilon=0$ and applying Theorem D.2 yields $R(U)=R(A_Z),R(V)=R(A_G)$.

## D.5   Proof of Lemma D.5

Define $\tilde{L}_b(\mathbf{W}):=[\mathbf{N},\mathbf{N}]_{\mathcal{G}}^2$, then it can be verified that

$$
\begin{aligned}
L_0(\theta)&=\frac{1}{2}[\mathbf{N}-\mathbf{N}^*,\mathbf{N}-\mathbf{N}^*]_{\mathcal{H}_0}^2=:\tilde{L}_0(\mathbf{W}),\\
L_b(\theta)&=[\mathbf{N},\mathbf{N}]_{\mathcal{G}}^2+\lambda_r[\tilde{\mathbf{N}},\tilde{\mathbf{N}}]_{\mathcal{G}}^2=\tilde{L}_b(\mathbf{W})+\lambda_r\tilde{L}_b(\tilde{\mathbf{W}}).
\end{aligned}
$$

Define $\tilde{L}_r(\mathbf{W}):=L_r(\theta)+\lambda_r\tilde{L}_b(\tilde{\mathbf{W}})$, then we have

$$
\tilde{L}^{\lambda_r}(\mathbf{W})=2\tilde{L}_0(\mathbf{W})+\frac{1}{2}\tilde{L}_b(\mathbf{W})+\lambda_r\tilde{L}_r(\mathbf{W})=[\mathbf{N}-\mathbf{N}^*,\mathbf{N}-\mathbf{N}^*]_{\mathcal{H}_0}^2+\frac{1}{2}[\mathbf{N},\mathbf{N}]_{\mathcal{G}}^2+\lambda_r\tilde{L}_r(\mathbf{W}).
$$

Then, consider the Fréchet derivative of $\tilde{L}_t(\mathbf{W})$ along $\boldsymbol{\Delta}(t)$, it can be shown that

$$
\begin{aligned}
\langle\nabla\tilde{L}^{\lambda_r}(\mathbf{W}),\boldsymbol{\Delta}\rangle_{\mathcal{I}}&=[\mathbf{N}-\mathbf{N}^*,\boldsymbol{\Delta}\mathbf{W}^\top+\mathbf{W}\boldsymbol{\Delta}^\top]_{2\mathcal{H}_0}+[\mathbf{N},\boldsymbol{\Delta}\mathbf{W}^\top+\mathbf{W}\boldsymbol{\Delta}^\top]_{\mathcal{G}}+\lambda_r\langle\nabla\tilde{L}_r(\mathbf{W}),\boldsymbol{\Delta}\rangle_{\mathcal{I}}\\
&=[\mathbf{N}-\mathbf{N}^*,\boldsymbol{\Delta}\mathbf{W}^\top+\mathbf{W}\boldsymbol{\Delta}^\top]_{2\mathcal{H}_0+\mathcal{G}}+[\mathbf{N}^*,\boldsymbol{\Delta}\mathbf{W}^\top+\mathbf{W}\boldsymbol{\Delta}^\top]_{\mathcal{G}}+\lambda_r\langle\nabla\tilde{L}_r(\mathbf{W}),\boldsymbol{\Delta}\rangle_{\mathcal{I}}\\
&=[\mathbf{N}-\mathbf{N}^*,\boldsymbol{\Delta}\mathbf{W}^\top+\mathbf{W}\boldsymbol{\Delta}^\top]_{2\mathcal{H}_0+\mathcal{G}}+2[\mathbf{N}^*,\mathbf{N}]_{\mathcal{G}}+\lambda_r\langle\nabla\tilde{L}_r(\mathbf{W}),\boldsymbol{\Delta}\rangle_{\mathcal{I}},\quad(40)
\end{aligned}
$$

where the last equality uses the fact that $[\mathbf{N}^*,\mathbf{W}^*\mathbf{W}^\top]_{\mathcal{G}}=[\mathbf{N}^*,\mathbf{W}\mathbf{W}^{*\top}]_{\mathcal{G}}=0$. To see this, notice that

$$
\begin{aligned}
[\mathbf{N}^*,\mathbf{W}^*\mathbf{W}^\top]_{\mathcal{G}}=&\langle AA^\top,\tilde{U}A^\top\rangle+\mathbb{E}_{t,p_t(x)p_t(x')}s^*(x,t)^\top s^*(x',t)s(x,t)^\top s^*(x',t)\\
&-\mathbb{E}_{t,p_t(x)}s^*(x,t)^\top A^\top\tilde{U}s^*(x,t)-\mathbb{E}_{t,p_t(x)}s^*(x,t)^\top A^\top As(x,t)\\
=&\operatorname{Tr}(A(A^\top A-\mathbb{E}_{t,p_t(x)}[s^*(x,t)s^*(x,t)])\tilde{U})+\\
&\mathbb{E}_{t,p_t(x)}[s^*(x,t)^\top(\mathbb{E}_{p_t(x',t)}[s^*(x',t)s^*(x',t)^\top]-A^\top A)s(x,t)]=0,
\end{aligned}
$$

where the last inequality uses the fact that $\tilde{L}_b(\mathbf{W}^*)=0$ and therefore

$$U^{*\top}U^* = \mathbb{E}_{t,p_t(x)}s_Z^*(x,t)s_Z^*(x,t)^\top,$$
$$V^{*\top}V^* = \mathbb{E}_{t,p_t(x)}s_G^*(x,t)s_G^*(x,t)^\top.$$

Similarly, we can show that $[\mathbf{N}^*,\mathbf{W}\mathbf{W}^{*\top}]_{\mathcal{G}}=0$.

Now, consider the Hessian of $\tilde{L}$ along $\boldsymbol{\Delta}$ by taking the Fréchet derivative of Eq. 40,

$$[\boldsymbol{\Delta},\boldsymbol{\Delta}]_{\nabla^2\tilde{L}^{\lambda_r}} = [\boldsymbol{\Delta}\mathbf{W}^\top+\mathbf{W}\boldsymbol{\Delta}^\top,\boldsymbol{\Delta}\mathbf{W}^\top+\mathbf{W}\boldsymbol{\Delta}^\top]^2_{2\mathcal{H}_0+\mathcal{G}}+2[\mathbf{N}-\mathbf{N}^*,\boldsymbol{\Delta}\boldsymbol{\Delta}^\top]_{2\mathcal{H}_0+\mathcal{G}}+$$
$$2[\mathbf{N}^*,\mathbf{W}\boldsymbol{\Delta}^\top+\boldsymbol{\Delta}\mathbf{W}^\top]_{\mathcal{G}}+\lambda_r[\boldsymbol{\Delta},\boldsymbol{\Delta}]_{\nabla^2\tilde{L}_r(\mathbf{W})}$$
$$=\left\|\boldsymbol{\Delta}\mathbf{W}^\top+\mathbf{W}\boldsymbol{\Delta}^\top\right\|^2_{\mathcal{I}}+2[\mathbf{N}-\mathbf{N}^*,\boldsymbol{\Delta}\boldsymbol{\Delta}^\top]_{2\mathcal{H}_0+\mathcal{G}}+2[\mathbf{N}^*,\mathbf{W}\boldsymbol{\Delta}^\top+\boldsymbol{\Delta}\mathbf{W}^\top]_{\mathcal{G}}+\lambda_r[\boldsymbol{\Delta},\boldsymbol{\Delta}]_{\nabla^2\tilde{L}_r(\mathbf{W})},$$

where we use Eq. 38 in the last equality. For the first term of the right-hand side, by the choice of $\boldsymbol{\Delta}$,

$$\|\boldsymbol{\Delta}\mathbf{W}^\top+\mathbf{W}\boldsymbol{\Delta}^\top\|^2_{\mathcal{I}}=\|\mathbf{N}-\mathbf{N}^*+\boldsymbol{\Delta}\boldsymbol{\Delta}^\top\|^2_{\mathcal{I}}$$
$$=\|\boldsymbol{\Delta}\boldsymbol{\Delta}^\top\|^2_{\mathcal{I}}+2\langle\mathbf{N}-\mathbf{N}^*,\boldsymbol{\Delta}\mathbf{W}^\top+\mathbf{W}\boldsymbol{\Delta}^\top\rangle_{\mathcal{I}}-\|\mathbf{N}-\mathbf{N}^*\|^2_{\mathcal{I}}$$
$$=\|\boldsymbol{\Delta}\boldsymbol{\Delta}^\top\|^2_{\mathcal{I}}+2\langle\nabla\tilde{L}^{\lambda_r}(\mathbf{W}),\boldsymbol{\Delta}\rangle_{\mathcal{I}}-4\langle\mathbf{N}^*,\mathbf{N}\rangle_{\mathcal{G}}-2\lambda_r\langle\nabla\tilde{L}_r(\mathbf{W}),\boldsymbol{\Delta}\rangle_{\mathcal{I}}-\|\mathbf{N}-\mathbf{N}^*\|^2_{\mathcal{I}}.$$

For the second term,

$$[\mathbf{N}-\mathbf{N}^*,\boldsymbol{\Delta}\boldsymbol{\Delta}^\top]_{2\mathcal{H}_0+\mathcal{G}}=[\mathbf{N}-\mathbf{N}^*,\mathbf{W}\boldsymbol{\Delta}^\top+\boldsymbol{\Delta}\mathbf{W}^\top]_{2\mathcal{H}_0+\mathcal{G}}-[\mathbf{N}-\mathbf{N}^*,\mathbf{N}-\mathbf{N}^*]^2_{2\mathcal{H}_0+\mathcal{G}}$$
$$=\langle\nabla\tilde{L}^{\lambda_r}(\mathbf{W}),\boldsymbol{\Delta}\rangle_{\mathcal{I}}-2\langle\mathbf{N}^*,\mathbf{N}\rangle_{\mathcal{G}}-\lambda_r\langle\nabla\tilde{L}_r(\mathbf{W}),\boldsymbol{\Delta}\rangle_{\mathcal{I}}-\|\mathbf{N}-\mathbf{N}^*\|^2_{\mathcal{I}}. \quad (41)$$

Combined with the fact that

$$[\mathbf{N}^*,\mathbf{W}\boldsymbol{\Delta}^\top+\boldsymbol{\Delta}\mathbf{W}^\top]_{\mathcal{G}}=2[\mathbf{N}^*,\mathbf{N}]_{\mathcal{G}},$$

we obtain

$$[\boldsymbol{\Delta},\boldsymbol{\Delta}]_{\nabla^2\tilde{L}^{\lambda_r}}=\|\boldsymbol{\Delta}\boldsymbol{\Delta}^\top\|^2_{\mathcal{I}}+4\langle\nabla\tilde{L}^{\lambda_r}(\mathbf{W}),\boldsymbol{\Delta}\rangle_{\mathcal{I}}-3\|\mathbf{N}-\mathbf{N}^*\|^2_{2\mathcal{I}}$$
$$-4[\mathbf{N},\mathbf{N}^*]_{\mathcal{G}}-4\lambda_r\langle\nabla\tilde{L}_r(\mathbf{W}),\boldsymbol{\Delta}\rangle_{\mathcal{I}}+\lambda_r[\boldsymbol{\Delta},\boldsymbol{\Delta}]_{\nabla^2\tilde{L}_r(\mathbf{W})}$$
$$\leq\|\boldsymbol{\Delta}\boldsymbol{\Delta}^\top\|^2_{\mathcal{I}}-3\|\mathbf{N}-\mathbf{N}^*\|^2_{\mathcal{I}}+4\langle\nabla\tilde{L}^{\lambda_r}(\mathbf{W}),\boldsymbol{\Delta}\rangle_{\mathcal{I}}-4\lambda_r\langle\nabla\tilde{L}_r(\mathbf{W}),\boldsymbol{\Delta}\rangle_{\mathcal{I}}+\lambda_r[\boldsymbol{\Delta},\boldsymbol{\Delta}]_{\nabla^2\tilde{L}_r(\mathbf{W})}$$
$$\leq\|\boldsymbol{\Delta}\boldsymbol{\Delta}^\top\|^2_{\mathcal{I}}-3\|\mathbf{N}-\mathbf{N}^*\|^2_{\mathcal{I}}+4\epsilon\|\boldsymbol{\Delta}\|_{\mathcal{I}}-4\lambda_r\langle\nabla\tilde{L}_r(\mathbf{W}),\boldsymbol{\Delta}\rangle_{\mathcal{I}}+\lambda_r[\boldsymbol{\Delta},\boldsymbol{\Delta}]_{\nabla^2\tilde{L}_r(\mathbf{W})}, \quad (42)$$

where the first inequality uses the fact that

$$[\mathbf{N},\mathbf{N}^*]_{\mathcal{G}}=\langle UU^\top,AA^\top\rangle+\mathbb{E}_{t,p_t(x)p_t(x')}s^*(x,t)^\top s^*(x',t)s(x,t)^\top x(x',t)-2\mathbb{E}_{t,p_t(x)}s(x,t)U^\top As^*(x',t)$$
$$=\|U^\top A-\mathbb{E}_{t,p_t(x)}s(x,t)s^*(x,t)^\top\|^2\geq 0,$$

and the second inequality uses the condition that $\mathbf{W}$ is an approximate critical point of $\tilde{L}^{\lambda_r}(\mathbf{W})$. It remains to bound the $\tilde{L}_r$ related terms. Define $\mathbf{N}^Z:=\mathbf{W}_Z\mathbf{W}_Z$, $\mathbf{N}^G:=\mathbf{W}_G\mathbf{W}_G^\top$ and similarly $\mathbf{N}^{Z*}$ and $\mathbf{N}^{G*}$. Notice that

$$\tilde{L}_r(\mathbf{W})=[\mathbf{N}^G-\mathbf{N}^*,\mathbf{N}^G-\mathbf{N}^*]_{\mathcal{H}_0}+\frac{1}{2}[\mathbf{N}^G,\mathbf{N}^G]^2_{\mathcal{G}}$$
$$=[\mathbf{N}^G-\mathbf{N}^{G*},\mathbf{N}^G-\mathbf{N}^{G*}]^2_{\mathcal{H}_0}-2[\mathbf{N}^G-\mathbf{N}^{G*},\mathbf{N}^{Z*}]_{\mathcal{H}_0}+[\mathbf{N}^{Z*},\mathbf{N}^{Z*}]_{\mathcal{H}_0}+\frac{1}{2}[\mathbf{N}^G,\mathbf{N}^G]_{\mathcal{G}}$$
$$=[\mathbf{N}^G-\mathbf{N}^{G*},\mathbf{N}^G-\mathbf{N}^{G*}]_{\mathcal{H}_0}+\frac{1}{2}[\mathbf{N}^G,\mathbf{N}^G]_{\mathcal{G}}=2\tilde{L}_0(\mathbf{W}_G)+\frac{1}{2}\tilde{L}_b(\mathbf{W}_G)=\tilde{L}^0(\mathbf{W}_G),$$

where the second-to-last equality uses the fact that

$$[\mathbf{N}^G-\mathbf{N}^{G*},\mathbf{N}^{Z*}]_{\mathcal{H}_0}=2\mathbb{E}_{t,p_t(x)}\langle Vs_G(g(x),t)-A_Gs_G^*(g(x),t),A_Zs_Z^*(z(x),t)\rangle$$
$$=2\mathbb{E}_{t,p_t(x)}\langle Vs_G(g(x),t),A_Zs_Z^*(z(x),t)\rangle$$
$$=2\langle V\mathbb{E}_{t,p_t(x)}s_G(g(x),t),A_Z\mathbb{E}_{t,p_t(x)}s_Z^*(z(x),t)\rangle=0,$$

where the second-to-last equality uses the independence of the content variable $Z$ and the style variable $G$ and the last equality uses the property of the score function:

$$\mathbb{E}_{p_Z} s_Z^*(z) = \mathbb{E}_{p_Z} \nabla_z \log p_Z(z) = \int \nabla_z p_Z(z) \mathrm{d}z = \nabla_z \int p_Z(z) \mathrm{d}z = 0.$$

Therefore, we can apply Eq. 42 with $\lambda_r = 0$ to obtain

$$[\mathbf{\Delta},\mathbf{\Delta}]_{\tilde{L}_r(\mathbf{W})} = \langle \mathbf{\Delta}_G, \mathbf{\Delta}_G \rangle_{\tilde{L}^0(\mathbf{W}_G)} \tag{43}$$

$$\leq \|\mathbf{\Delta}_G \mathbf{\Delta}_G^\top\|_{\mathcal{I}}^2 - 3\|\mathbf{N}^G - \mathbf{N}^{G*}\|_{\mathcal{I}}^2 + 4\langle \nabla \tilde{L}_r(\mathbf{W}), \mathbf{\Delta} \rangle_{\mathcal{I}}. \tag{44}$$

Plugging Eq. 43 into Eq. 42 yields

$$[\mathbf{\Delta},\mathbf{\Delta}]_{\tilde{L}^{\lambda_r}(\mathbf{W})} \leq \|\mathbf{\Delta}\mathbf{\Delta}^\top\|_{\mathcal{I}}^2 + \lambda_r \|\mathbf{\Delta}_G \mathbf{\Delta}_G^\top\|_{\mathcal{I}} - 3(\|\mathbf{N} - \mathbf{N}^*\|_{\mathcal{I}}^2 + \lambda_r \|\mathbf{N}^G - \mathbf{N}^{G*}\|_{\mathcal{I}}^2) + \epsilon\|\mathbf{\Delta}\|_{\mathcal{I}}$$

$$= \|\tilde{\mathbf{\Delta}}\tilde{\mathbf{\Delta}}\|_{\mathcal{I}}^2 - 3\|\tilde{\mathbf{N}} - \tilde{\mathbf{N}}^*\|_{\mathcal{I}}^2 + \epsilon\|\mathbf{\Delta}\|_{\mathcal{I}},$$

where the last equality uses the definition of $\mathbf{N},\mathbf{N}^*,\mathbf{\Delta}$ and $\mathbf{\Delta}^*$. For example, for $\mathbf{N}$, we have

$$\|\tilde{\mathbf{N}}\|_{\mathcal{I}}^2 = \left\| \begin{bmatrix} \mathbf{W}\mathbf{W}^\top & \sqrt{\frac{\lambda_r}{3}}\mathbf{W}_G\mathbf{W}_G^\top \\ \sqrt{\frac{\lambda_r}{3}}\mathbf{W}_G\mathbf{W}_G^\top & \frac{\lambda_r}{3}\mathbf{W}_G\mathbf{W}_G^\top \end{bmatrix} \right\|_{\mathcal{I}}^2 = \|\mathbf{N}\|_{\mathcal{I}}^2 + \lambda_r\|\mathbf{N}^G\|_{\mathcal{I}}^2 + \left(\frac{\lambda_r^2}{9} - \frac{\lambda_r}{3}\right)\|\mathbf{N}^G\|_{\mathcal{I}}^2$$

$$= \|\mathbf{N}\|_{\mathcal{I}}^2 + 3\|\mathbf{N}^G\|_{\mathcal{I}}^2,$$

where in the last equality we set $\lambda_r = 3$.

## D.6 Proof of Lemma D.6

First, let $\Delta(\mathbf{X}) := U(\mathbf{X}) - Y(\mathbf{X})$, by definition,

$$\mathbb{E}\Delta(\mathbf{X})^\top U(\mathbf{X}) = \mathbb{E}U(\mathbf{X})^\top U(\mathbf{X}) - \mathbb{E}Y(\mathbf{X})^\top U(\mathbf{X}) = \mathbb{E}U(\mathbf{X})^\top \Delta(\mathbf{X}),$$

and

$$\mathbb{E}\|U(\mathbf{X})U(\mathbf{X}')^\top - Y(\mathbf{X})Y(\mathbf{X}')^\top\|^2 = \mathbb{E}\|\Delta(\mathbf{X})U(\mathbf{X}')^\top + U(\mathbf{X})\Delta(\mathbf{X}')^\top - \Delta(\mathbf{X})\Delta(\mathbf{X}')^\top\|^2.$$

Expanding the square norm,

$$\mathbb{E}\|\Delta(\mathbf{X})U(\mathbf{X}')^\top + U(\mathbf{X})\Delta(\mathbf{X}')^\top - \Delta(\mathbf{X})\Delta(\mathbf{X}')^\top\|^2$$

$$= 2\mathbb{E}\|\Delta(\mathbf{X})U(\mathbf{X}')^\top\|^2 + \|\mathbb{E}\Delta(\mathbf{X})^\top\Delta(\mathbf{X})\|^2 + 2\langle\mathbb{E}\Delta(\mathbf{X})^\top U(\mathbf{X}), \mathbb{E}U(\mathbf{X})^\top\Delta(\mathbf{X})\rangle -$$

$$2\mathbb{E}\langle\Delta(\mathbf{X})U(\mathbf{X}')^\top + U(\mathbf{X})\Delta(\mathbf{X}')^\top, \Delta(\mathbf{X})\Delta(\mathbf{X}')^\top\rangle$$

$$= \|\mathbb{E}\Delta(\mathbf{X})^\top\Delta(\mathbf{X})\|^2 + 2\langle\mathbb{E}\Delta(\mathbf{X})^\top U(\mathbf{X}), \mathbb{E}\Delta(\mathbf{X})^\top\Delta(\mathbf{X})\rangle + 2\|\mathbb{E}\Delta(\mathbf{X})^\top U(\mathbf{X})\|^2 -$$

$$4\langle\mathbb{E}\Delta(\mathbf{X})^\top U(\mathbf{X}), \mathbb{E}\Delta(\mathbf{X})^\top\Delta(\mathbf{X})\rangle$$

$$= \frac{1}{2}\|\mathbb{E}\Delta(\mathbf{X})^\top\Delta(\mathbf{X})\|^2 + 2\langle\mathbb{E}U(\mathbf{X})^\top Y, \mathbb{E}\Delta(\mathbf{X})^\top\Delta(\mathbf{X})\rangle +$$

$$\|\sqrt{2}U(\mathbf{X})^\top\Delta(\mathbf{X}) - \frac{1}{\sqrt{2}}\Delta(\mathbf{X})^\top\Delta(\mathbf{X})\|^2$$

$$\geq \frac{1}{2}\|\mathbb{E}\Delta(\mathbf{X})^\top\Delta(\mathbf{X})\|^2 = \frac{1}{2}\mathbb{E}\|(U(\mathbf{X}) - Y(\mathbf{X}))(U(\mathbf{X}') - Y(\mathbf{X}'))^\top\|^2,$$

where the second equality uses the symmetry of $\mathbb{E}U(\mathbf{X})^\top\Delta(\mathbf{X})$ the last inequality uses the PSD of $\mathbb{E}U(\mathbf{X})^\top Y(\mathbf{X})$.

Similarly,

$$\mathbb{E}\|U(\mathbf{X})U(\mathbf{X}')^\top - Y(\mathbf{X})Y(\mathbf{X}')^\top\|^2$$

$$= 2\mathbb{E}\|\Delta(\mathbf{X})U(\mathbf{X}')^\top\|^2 + \|\mathbb{E}\Delta(\mathbf{X})^\top\Delta(\mathbf{X})\|^2 + 2\|\mathbb{E}\Delta(\mathbf{X})^\top U(\mathbf{X})\|^2 -$$

$$4\langle\mathbb{E}\Delta(\mathbf{X})^\top U(\mathbf{X}), \mathbb{E}\Delta(\mathbf{X})^\top\Delta(\mathbf{X})\rangle$$

$$= (2\sqrt{2} - 2)\mathbb{E}\|U(\mathbf{X})\Delta(\mathbf{X}')^\top\|^2 + (4 - 2\sqrt{2})\langle\mathbb{E}U(\mathbf{X})^\top Y(\mathbf{X}), \mathbb{E}\Delta(\mathbf{X})^\top\Delta(\mathbf{X})\rangle +$$

$$\|\sqrt{2}\mathbb{E}U(\mathbf{X})^\top\Delta(\mathbf{X}) - \mathbb{E}\Delta(\mathbf{X})^\top\Delta(\mathbf{X})\|^2$$

$$\geq (2\sqrt{2} - 2)\mathbb{E}\|U(\mathbf{X})\Delta(\mathbf{X}')^\top\|^2 = 2(\sqrt{2} - 1)\mathbb{E}\|(U(\mathbf{X}) - Y(\mathbf{X}))U(\mathbf{X}')^\top\|^2,$$

where the last inequality uses the PSD of $\mathbb{E}U(\mathbf{X})^\top Y(\mathbf{X})$.

## D.7 Proof of Lemma D.7

The proof relies on the following lemma.

**Lemma D.8.** *Suppose the score network parameters are initialized randomly as $\theta_Z(0), \theta_G(0) \sim \mathcal{N}(0,I)$. Then as $d_H \to \infty$, the NTKs $K_Z$ and $K_G$ converge to some kernels $K_Z^*$ and $K_G^*$ fixed during training. Further, define the minimal eigenvalues of operator $K : \mathcal{X} \times [t_0,T] \times \mathcal{X} \times [t_0,T] \mapsto \mathbb{R}^d$ as*

$$\lambda_{\min}(K) := \inf_{v:\mathbb{E}_{t,p(x)}\|v(t,x)\|^2 = 1} \mathbb{E}_{t,t',p(x)p(x')} v(x,t)^\top K(x,t,x',t') v(x',t'),$$

*then the minimal eigenvalues of $K_Z^*$ and $K_G^*$ satisfy $\min\{\lambda_{\min}(K_Z^*), \lambda_{\min}(K_G^*)\} > 0$.*

Define the random gradient of loss $L$ with respect to random matrix $\mathbf{Y}$ with probability density $q_t$ as

$$\nabla_{\mathbf{Y}} L(\mathbf{Y}) = \nabla_{Y(x,t)} L(\mathbf{Y}) \quad \text{w.p.} \quad p(t)q_t(x).$$

By the definition of the gradient flow equations in Eq. 12, we have

$$\dot{\tilde{L}}(\mathbf{W}) = \langle \nabla_{\mathbf{W}} \tilde{L}, \dot{\mathbf{W}} \rangle_{\mathcal{I}}$$
$$= -\|\nabla_U \tilde{L}(\mathbf{W})\|^2 - \|\nabla_V \tilde{L}(\mathbf{W})\|^2 - \|\mathbb{E}_{t,p_t} J_{\theta_Z} s_Z \nabla_{s_Z} \tilde{L}(\mathbf{W})\|^2 - \|\mathbb{E}_{t,p_t} J_{\theta_G} s_G \nabla_{s_G} \tilde{L}(\mathbf{W})\|^2,$$

where the first two terms of the RHS by the property of the gradient flow, vanishes if and only if the gradients $\nabla_U \tilde{L}(\mathbf{W})$ and $\nabla_V \tilde{L}(\mathbf{W})$ become 0. For the third term of the RHS, notice that

$$\|\mathbb{E}_{t,p_t} J_{\theta_Z} s_Z \nabla_{s_Z} \tilde{L}(\mathbf{W})\|^2$$
$$= \mathbb{E}_{t,t',p_t(x)p_{t'}(x')} \nabla_{s_Z(x,t)} \tilde{L}(\mathbf{W})^\top J_{\theta_Z} s_Z(x,t)^\top J_{\theta_Z} s_Z(x',t') \nabla_{s_Z(x',t')} \tilde{L}(\mathbf{W})$$
$$= \mathbb{E}_{t,t',p_t(x)p_{t'}(x')} \nabla_{s_Z(x,t)} \tilde{L}(\mathbf{W})^\top K_Z(x,t,x',t') \nabla_{s_Z(x',t')} \tilde{L}(\mathbf{W})$$
$$\geq \lambda_{\min}(K_Z) \int \|p(t)p_t(x) \nabla_{s_Z(x,t)} \tilde{L}(\mathbf{W})\|^2 \mathrm{d}x \mathrm{d}t.$$

By Lemma D.8, $\lambda_{\min}(K_Z) \xrightarrow{d_H \to \infty} \lambda_{\min}(K_Z^*) > 0$ and thus the term vanishes if and only if

$$\|p(t)p_t(x) \mathbb{E}_{t,p_t(x)} \nabla_{s_Z(x,t)} \tilde{L}(\mathbf{W})\|^2 = 0, \forall x,t.$$

Similarly, we can show that the gradient flow converges only if

$$\|p(t)p_t(x) \mathbb{E}_{t,p_t(x)} \nabla_{s_G(g(x),t)} \tilde{L}(\mathbf{W})\|^2 = 0, \forall x,t.$$

For the second order condition, we use the well-known result that gradient descent (with small noise) is able to escape saddle points almost surely [126].

## D.8 Proof of Lemma D.8

Due to the symmetry in their forms, it suffices to prove the statement for $s_Z$ and $K_Z$ and we omit the subscript when the context is clear. Large of large number and the standard theory on neural tangent kernel [123] yields

$$K_Z \xrightarrow{d_H \to \infty} \mathbb{E}_{\mathrm{vec}(\theta_Z) \sim \mathcal{N}(0,I)} J_{\mathrm{vec}(\theta_Z)} s_Z(x,t)^\top J_{\mathrm{vec}(\theta_Z)} s_Z(x',t'), \tag{45}$$

which stays fixed during training.

Define $\tilde{x} := [x^\top, \mathrm{PE}(t)]^\top$ and $a(\tilde{x}) := (\theta^{(1)\top} \tilde{x})_+$. Later, we will slightly abuse the notation to represent $(x,t)$ when $\tilde{x}$ appears in the true score functions. By definition,

$$K_Z^{(2)}(x,t,x',t') = \mathbb{E}_{\mathcal{N}(0,I)} J_{\mathrm{vec}(\theta_Z^{(2)})} s_Z(\tilde{x})^\top J_{\mathrm{vec}(\theta_Z^{(2)})} s_Z(\tilde{x}')$$
$$= \mathbb{E}_{\mathcal{N}(0,I)} J_{\mathrm{vec}(\theta_Z^{(2)})} \theta_Z^{(2)} a(\tilde{x})^\top J_{\mathrm{vec}(\theta_Z^{(2)})} \theta_Z^{(2)} a(\tilde{x}')$$
$$= \mathbb{E}_{\mathcal{N}(0,I)} (a(\tilde{x})^\top \otimes I_{d_Z})(a(\tilde{x}') \otimes I_{d_X}) = \mathbb{E}_{\mathcal{N}(0,I)} a(\tilde{x})^\top a(\tilde{x}') I_{d_Z},$$

where $\otimes$ is the Kronecker product. Notice that by definition $K_Z^{(2)} \succeq 0$. Similarly,

$$
\begin{aligned}
K_Z^{(1)}(x,t,x',t') &:= \mathbb{E}_{\mathcal{N}(0,I)} J_{\mathrm{vec}(\theta_Z^{(1)})} s_Z(\tilde{x})^\top J_{\mathrm{vec}(\theta_Z^{(1)})} s_Z(\tilde{x}') \\
&= \mathbb{E}_{\mathcal{N}(0,I)} J_{a(\tilde{x})} s_Z(\tilde{x})^\top J_{\theta_Z^{(1)}} a(\tilde{x})^\top J_{\theta_Z^{(1)}} a(\tilde{x}') J_{a(\tilde{x}')} s_Z(\tilde{x}') \\
&=: (\tilde{x}^\top \tilde{x}') \mathbb{E}_{\mathcal{N}(0,I)} \theta_Z^{(2)} S^2(\tilde{x},\tilde{x}') \theta_Z^{(2)\top}, \\
&= (\tilde{x}^\top \tilde{x}') \mathbb{E}_{\mathcal{N}(0,I)} S(\tilde{x},\tilde{x}'),
\end{aligned}
$$

where we use the independence between $(\theta_Z^{(1)}, \theta_Z^{(2)})$ to cancel out $\theta_Z^{(1)}$, and

$$
S_{ij}(\tilde{x},\tilde{x}') := \begin{cases} \mathbb{1}[\theta^{(1),j\top}\tilde{x} \geq 0, \theta^{(1),j\top}\tilde{x}' \geq 0], & \text{if } i = j, \\ 0, & \text{otherwise.} \end{cases}
$$

By assumption, we have $\theta^{(1),j\top}\tilde{x}$'s are zero-mean Gaussians and thus

$$
\Pr[\theta^{(1),j\top}\tilde{x} \geq 0, \theta^{(1),j\top}\tilde{x}' \geq 0] = \frac{1}{4} + \frac{1}{2\pi} \arcsin\left(\frac{\tilde{x}^\top \tilde{x}'}{\|\tilde{x}\| \|\tilde{x}'\|}\right),
$$

where we apply the formula for bivariate Gaussian variable $(N_1, N_2)$: $\Pr[N_1 \geq 0, N_2 \geq 0] = \frac{1}{4} + \arcsin(\rho)$ where $\rho := \frac{\mathrm{Cov}(N_1,N_2)}{\sqrt{\mathrm{Var}(N_1)\mathrm{Var}(N_2)}}$.

As a result,

$$
K_Z^{(1)}(\tilde{x},\tilde{x}') := \frac{1}{4} \tilde{x}^\top \tilde{x}' \left[1 + \frac{2}{\pi} \arcsin\left(\frac{\tilde{x}^\top \tilde{x}'}{\|\tilde{x}\| \|\tilde{x}'\|}\right)\right] I_{d_Z}.
$$

Notice that $K_Z^{(1)}$ is a positive definite operator since for any finite set of distinct samples $\tilde{X} := [\tilde{x}_1, \cdots, \tilde{x}_n]$ with $t_i \neq t_j$ for all $i,j$, the matrix

$$
K := \{K_Z(\tilde{x}_i, \tilde{x}_j)\}_{ij} \succ \frac{1}{4} \tilde{X}^\top \tilde{X} \otimes I_{d_z} \succ 0,
$$

where we use $\|\tilde{x}\| > 0$ and $\mathrm{PE}(t_i)$'s are linearly independent from the condition of the lemma, and thus their Gram matrix $\tilde{X}^\top \tilde{X} \succ 0$. Consequently, $K_Z = K_Z^{(1)} + K_Z^{(2)} \succ 0$ and therefore its minimal eigenvalue is positive.

## D.9 Proof of Lemma D.4

As $d_H, d_T \to \infty$, we have $K_Z = K_Z^*, K_G = K_G^*$. First, we prove the following lemmas in Section D.10 and D.11 respectively.

**Lemma D.9.** *Suppose $n > \max\{d_Z, d_G\}$, and $\min\{d_T, d_H\} \to \infty$, the following holds for the empirical risk minimizer $(\hat{s}_Z, \hat{s}_G, \hat{U}, \hat{V})$ with probability at least $1 - 2\exp(-n)$:*

$$
\hat{L}_Z(\hat{U}, \hat{s}_U) = \hat{L}_G(\hat{V}, \hat{s}_V) = 0,
$$

$$
\|\hat{s}_Z\|_{K_Z} \leq C_Z \sqrt{\frac{\sigma_1(s_Z^*) d_Z^{1/2}}{\lambda_{\min}(K_Z^*)}},
$$

$$
\|\hat{s}_G\|_{K_G} \leq C_G \sqrt{\frac{\sigma_1(s_G^*) d_G^{1/2}}{\lambda_{\min}(K_G^*)}},
$$

*for some constant $C_Z, C_G > 0$.*

**Lemma D.10.** *For the NTK of the estimated content and style score functions $K_Z$ and $K_G$ defined in Lemma D.7 and for any $(x,t) \in \mathcal{X} \times [t_0, T]$, the following holds:*

$$
\|K_(x,t,x,t)\| \leq \frac{3}{2}(\|x\|^2 + \|PE(t)\|^2)
$$

$$
\|K_G(x,t,x,t)\| \leq \frac{3}{2}(\lambda_g^2 \|x\|^2 + \|PE(t)\|^2).
$$

Then we make use of the following properties of sub-gaussian random variables from Lemma 16 of [41].

**Lemma D.11.** *Consider a probability density function $p(x) \leq \exp(-C\|x\|_2^2/2)$ for $x \in \mathbb{R}^d$ and constant $C > 0$. Let $R$ be a fixed radius. Then the following holds*

$$\int_{\|x\| > R} p(x)\mathrm{d}x \leq \frac{2d\pi^{d/2}}{C\Gamma(d/2+1)} R^{d-2}\exp(-CR^2/2),$$

$$\int_{\|x\| > R} \|x\|^2 p(x)\mathrm{d}x \leq \frac{2d\pi^{d/2}}{C\Gamma(d/2+1)} R^d\exp(-CR^2/2).$$

Define $\rho_Z := C_Z \sqrt{\frac{\sigma_1(s_Z^*)d_Z^{1/2}}{\lambda_{\min}(K_Z^*)}}$, and without loss of generality, assume $\sup_{t \in [t_0,T]} \|\mathsf{PE}(t)\| \leq T$. To prove lemma D.4, we first bound the Rademacher average of $\mathcal{S}_Z$ as

$$\mathcal{R}_n(\mathcal{S}_Z) = \mathbb{E}_{\epsilon^n} \sup_{s_Z \in \mathcal{S}_Z(\tilde{x}^{1:n})} \frac{1}{n} \sum_{i=1}^{n} \langle \epsilon_i, s_Z(\tilde{x}^i) \rangle$$

$$= \frac{1}{n} \mathbb{E}_{\epsilon^n} \sup_{c \in \mathcal{C}_Z(\tilde{x}^{1:n})} \sum_{i=1}^{n} \sum_{j=1}^{d_Z} \epsilon_{ij} \langle s_{Z,j}, K_{Z,\tilde{x}^i,j} \rangle_{K_Z} = \frac{1}{n} \mathbb{E}_{\epsilon^n} \sup_{c \in \mathcal{C}_Z(\tilde{x}^{1:n})} \sum_{j=1}^{d_Z} \left\langle s_{Z,j}, \sum_{i=1}^{n} \epsilon_{ij} K_{Z,\tilde{x}^i,j} \right\rangle_{K_Z}$$

$$\leq \frac{\rho_Z}{n} \mathbb{E}_{\epsilon^n} \sqrt{\sum_{j=1}^{d_Z} \left\| \sum_{i=1}^{n} \epsilon_{ij} K_{Z,\tilde{x}^i,j} \right\|_{K_Z}^2} \leq \frac{\rho_Z}{n} \sqrt{\mathbb{E}_{\epsilon^n} \sum_{j=1}^{d_Z} \left\| \sum_{i=1}^{n} \epsilon_{ij} K_{Z,\tilde{x}^i,j} \right\|_{K_Z}^2}$$

$$= \frac{\rho_Z}{n} \sqrt{\sum_{i=1}^{n} p(t^i)p_{t^i}(x^i) \mathbb{E}_{\epsilon_i} \epsilon_i^\top K_Z(\tilde{x}^i,\tilde{x}^i) \epsilon_i} \leq \frac{\rho_Z}{n} \sqrt{\sum_{i=1}^{n} p(t^i)p_{t^i}(x^i) \mathrm{Tr}(K_Z(\tilde{x}^i,\tilde{x}^i))}.$$

Averaging over $\tilde{x}^n$ and applying Cauchy-Schwarz inequality, we can bound the Rademacher complexity of the data-dependent function class $\mathcal{S}_Z$ as

$$\mathbb{E}_{\tilde{x}^n} \mathcal{R}_n(\mathcal{S}_Z) = \mathbb{E}_{\tilde{x}^n} \frac{\rho_Z}{n} \sqrt{\sum_{i=1}^{n} p(t^i)p_{t^i}(x^i) \mathrm{Tr}(K_Z(\tilde{x}^i,\tilde{x}^i))}$$

$$\leq \frac{\rho_Z}{\sqrt{n}} \sqrt{\mathbb{E}_{t,t,p_t(x)p_t(x)} \mathrm{Tr}(K_Z(\tilde{x},\tilde{x}))} =: \frac{\rho_Z C_{K_Z}}{\sqrt{n}},$$

where $C_{K_Z} := \sqrt{\mathbb{E}_{t,t,p_t(x)p_t(x)} \mathrm{Tr}(K_Z(\tilde{x},\tilde{x}))} = O(d_X)$.

To proceed, let the content subspace matrix and score function class learned by solving Eq. 12 given training data $[\tilde{x}^1, \cdots, \tilde{x}^n]$ be $\mathcal{U}(\tilde{x}^{1:n})$ and $\mathcal{S}_Z(\tilde{x}^{1:n})$ respectively, then by Lemma D.6, with probability at least $1 - 2\exp(-\Omega(n))$, the event

$$\mathcal{E} := \mathcal{S}_Z(\tilde{x}^{1:n}) \subseteq \mathcal{S}_Z \tag{46}$$

will occur, which implies with the same probability bound, the empirical risk minimizer over $\mathcal{S}_Z(x^{1:n})$ satisfies

$$\hat{L}_Z(\hat{U}, \hat{s}_Z) = \min_{(U,s_Z) \in \mathcal{U}(\tilde{x}^{1:n}) \times \mathcal{S}_Z(\tilde{x}^{1:n})} \frac{1}{n} \sum_{i=1}^{n} \ell_Z(\tilde{x}^i; U, s_Z) = \min_{U,s_Z \in \mathcal{S}_Z} \frac{1}{n} \sum_{i=1}^{n} \ell_Z(\tilde{x}^i; U, s_Z) = 0.$$

Therefore, let an empirical risk minimizer of $L_Z$ over $\mathcal{S}_Z$ be $\hat{U}', \hat{s}_Z'$, then for any $\epsilon > 0$, we can bound the generalization error probability as

$$\Pr\left[ L_Z(\hat{U}, \hat{s}_Z) \geq \min_{(U,s_Z) \in \mathcal{U} \times \mathcal{S}_Z} L_Z(U,s_Z) + \epsilon \right]$$

$$\leq \Pr(\mathcal{E}) + \Pr\left[ L_Z(\hat{U}', \hat{s}_Z') \geq \min_{(U,s_Z) \in \mathcal{U} \times \mathcal{S}_Z} L_Z(U,s_Z) + \epsilon \right] \tag{47}$$

$$\leq \Pr\left[ L_Z(\hat{U}', \hat{s}_Z') \geq \min_{(U,s_Z) \in \mathcal{U} \times \mathcal{S}_Z} L_Z(U,s_Z) + \epsilon \right] + 2\exp(-\Omega(n)).$$

Next, since the squared loss $\ell_Z$ is not Lipschitz with respect to $x$, we apply a truncation argument on $L_Z$. To this end, define the truncated version of $L_Z$ as

$$L_Z^{\text{trunc}}(U,s_Z):=\mathbb{E}_{t,p_t}\ell(x,t;U,s_Z)\mathbb{1}_{\|x\|\leq R}$$

for some radius $R>0$. Similarly we can define its empirical version as $L_Z^{\text{trunc}}$. Then $L_Z$ admits the following decomposition:

$$
\begin{aligned}
&L_Z(\hat{U}',\hat{s}_Z')-L_Z(A_Z,s_Z^*)\\
=&L_Z^{\text{trunc}}(\hat{U}',\hat{s}_Z')-L_Z^{\text{trunc}}(A_Z,s_Z^*)+L_Z(\hat{U},\hat{s}_Z)-L_Z^{\text{trunc}}(\hat{U}',\hat{s}_Z')+L_Z^{\text{trunc}}(A_Z,s_Z^*)-L_Z(A_Z,s_Z^*)\\
\leq&\underbrace{L_Z^{\text{trunc}}(\hat{U}',\hat{s}_Z')-L_Z^{\text{trunc}}(A_Z,s_Z^*)}_{(i)}+\underbrace{L_Z(\hat{U},\hat{s}_Z)-L_Z^{\text{trunc}}(\hat{U}',\hat{s}_Z')}_{(ii)}
\end{aligned}
$$

To bound $(i)$, notice that for $s_Z\in\mathcal{S}_Z$, with balanced weight $U\in\mathcal{U}_Z$,

$$
\begin{aligned}
&|\ell(x',t';U,s_Z)\mathbb{1}_{\|x'\|\leq R}-\ell(x,t;U,s_Z)\mathbb{1}_{\|x\|\leq R}|\\
\leq&\sup_{x\in\mathcal{X},t\in[t_0,T]}\sup_{(U,s_Z)\in\mathcal{U}\times\mathcal{S}_Z}|\ell(x,t;U,s_Z)\mathbb{1}_{\|x\|\leq R}|\\
\leq&\sup_{x\in\mathcal{X},t\in[t_0,T]}\sup_{(U,s_Z)\in\mathcal{U}\times\mathcal{S}_Z}2(\|Us_Z(x,t)\|^2+\|s_Z^*(x,t)\|^2)\mathbb{1}_{\|x\|\leq R}\\
=&\sup_{x\in\mathcal{X},t\in[t_0,T]}\sup_{(U,s_Z)\in\mathcal{U}\times\mathcal{S}_Z}2\|Us_Z(x,t)\|^2\mathbb{1}_{\|x\|\leq R}+2\lambda_s^2(R^2+T^2)\\
=&(3\rho_Z^4\lambda_{\min}(K_Z^*)+2\lambda_s^2)(R^2+T^2)=:B,
\end{aligned}
$$

where the second-to-last inequality uses the Lipschitz property of $s_Z^*$:

$$\|s_Z^*(x,t)\|^2\mathbb{1}_{\|x\|\leq R}\leq\lambda_s^2\|\tilde{x}\|^2\mathbb{1}_{\|x\|\leq R}\leq\lambda_s^2(R^2+T^2),\tag{48}$$

where the last inequality uses Lemma D.10 as follows:

$$
\begin{aligned}
\|Us_Z(x,t)\|&\leq\|U\|\|s_Z(x,t)\|\\
&\leq\rho_Z\sqrt{\lambda_{\min}(K_Z)}\sqrt{\sum_{j=1}^{d_Z}\langle s_{Z,j},K_{Z,x,t,j}\rangle_{K_Z}^2}\\
&\leq\rho_Z\sqrt{\lambda_{\min}(K_Z)}\|s_Z\|_{K_Z}\|K_{Z,x,t}\|_{K_Z}\\
&\leq\rho_Z^2\sqrt{\lambda_{\min}(K_Z)}\|K_Z(x,t,x,t)\|\leq\rho_Z^2\sqrt{\frac{3}{2}\lambda_{\min}(K_Z)}\|\tilde{x}\|.
\end{aligned}\tag{49}
$$

Using Eq. 48-49, we can show that

$$
\begin{aligned}
\|Us_Z(\tilde{x}^i)-A_Zs_Z^*(\tilde{x}^i)\|\mathbb{1}_{\|x^i\|\leq R}&\leq\|Us_Z(\tilde{x}^i)\mathbb{1}_{\|x^i\|\leq R}\|+\|A_Zs_Z^*(\tilde{x}^i)\mathbb{1}_{\|x^i\|\leq R}\|\\
&\leq(\rho_Z^2\sqrt{3\lambda_{\min}(K_Z^*)/2}+\lambda_s)\sqrt{R^2+T^2}.
\end{aligned}
$$

Let $\lambda_{\text{trunc}}:=2(\rho_Z^2\sqrt{3\lambda_{\min}(K_Z^*)/2}+\lambda_s)\sqrt{R^2+T^2}$ and use the inequality for $\max\{|s|,|t|\}\leq\lambda$,

$$|s^2-t^2|<2\lambda|s-t|,$$

we deduce that

$$|\ell(\tilde{x}^i;U',s_Z')-\ell(\tilde{x}^i;U,s_Z)|\mathbb{1}_{\|x^i\|\leq R}\leq\lambda_{\text{trunc}}|\|U's_Z'(\tilde{x}^i)-A_Zs_Z^*(\tilde{x}^i)\|-\|Us_Z(\tilde{x}^i)-A_Zs_Z^*(\tilde{x}^i)\||,$$

and thus $\ell(\tilde{x}^i;U,s_Z)$ is a $\lambda_{\text{trunc}}$-Lipschitz function of $\|Us_Z(\tilde{x}^i)-A_Zs_Z^*(\tilde{x}^i)\|$. Denote

$$
\begin{aligned}
\mathcal{U}\mathcal{S}_Z&:=\{Us_Z:U\in\mathcal{U},s_Z\in\mathcal{S}_Z\},\\
\ell\circ\mathcal{U}\mathcal{S}_Z(\tilde{x}^n)&:=\{(\ell(\tilde{x}^1;U,s_Z),\cdots,\ell(\tilde{x}^n;U,s_Z)):Us_Z\in\mathcal{U}\mathcal{S}_Z\},
\end{aligned}
$$

and apply the standard generalization bound for Lipschitz function on $L_Z^{\text{trunc}}$ (see, e.g., Theorem 9.1 of [127]) to conclude that with probability at least $1-\delta$,

$$
\begin{aligned}
(i) &\leq 2\mathbb{E}_{\tilde{x}^n}\mathcal{R}_n(\ell\circ\mathcal{US}_Z(\tilde{x}^n))+\frac{B}{2}\sqrt{\frac{8\log(1/\delta)}{n}} \\
&\leq 2\lambda_{\text{trunc}}\left(\mathbb{E}\mathcal{R}_n(\mathcal{US}_Z)+\frac{\lambda_s\sqrt{R^2+T^2}}{\sqrt{n}}\right)+\frac{B}{2}\sqrt{\frac{8\log(1/\delta)}{n}} \\
&\leq 2\lambda_{\text{trunc}}\frac{\rho_Z^2 C_{K_Z}+\lambda_s\sqrt{R^2+T^2}}{\sqrt{n}}+\frac{B}{2}\sqrt{\frac{8\log(1/\delta)}{n}} \\
&=O\left(\frac{\lambda_s\rho_Z^4(R^2+T^2)}{\sqrt{n}}+\lambda_s\rho_Z^4(R^2+T^2)\sqrt{\frac{\log 1/\delta}{n}}\right)=O\left(\frac{\lambda_s d_Z T^2 R^2}{\sqrt{n}}+\lambda_s d_Z T^2 R^2\sqrt{\frac{\log 1/\delta}{n}}\right),
\end{aligned}
$$

where the second inequality uses triangle inequality within the definition of Rademacher complexity followed by the bound in Eq. 48, and the third inequality uses the contraction principle of Rademacher complexity [127].

To bound $(ii)$, notice that

$$
\begin{aligned}
(ii)&=\mathbb{E}_{t,p_t(x)}\ell_Z(x,t;\hat{U}',\hat{s}_Z')\mathbb{1}_{\|x\|>R} \\
&\leq 2\mathbb{E}_{t,p_t(x)}(\|\hat{U}'\hat{s}_Z'(x,t)\|^2+\|s_Z^*(x,t)\|^2)\mathbb{1}_{\|x\|>R} \\
&\leq 2\mathbb{E}_{t,p_t(x)}\left[\|\hat{U}'\hat{s}_Z'(x,t)\|^2+\lambda_s^2(\lambda_z^2\|x\|^2+t^2)\right]\mathbb{1}_{\|x\|>R} \\
&\leq 2\mathbb{E}_{t,p_t(x)}\left[\frac{3}{2}\rho_Z^4\lambda_{\min}(K_Z)d_Z(\|x\|^2+\|\text{PE}(t)\|^2)+\lambda_s^2(\lambda_z^2\|x\|^2+t^2)\right]\mathbb{1}_{\|x\|>R} \\
&=(3\rho_Z^4\lambda_{\min}(K_Z)d_Z+2\lambda_s^2\lambda_z^2)\mathbb{E}_{t,p_t(x)}\|x\|^2\mathbb{1}_{\|x\|>R}+\mathbb{E}_{t,p_t(x)}\left(\lambda_s^2 t^2+\frac{3}{2}\|\text{PE}(t)\|^2\right)\mathbb{1}_{\|x\|>R} \\
&=(3\rho_Z^4\lambda_{\min}(K_Z)d_Z+2\lambda_s^2\lambda_z^2)\frac{\sigma_X^2 d_X 2^{-d_X/2+1}R^{d_X}}{\Gamma(d_X/2+1)}\exp(-R^2/2\sigma_X^2)+ \\
&\quad\left(\frac{3}{2}\rho_Z^4\lambda_{\min}(K_Z)d_Z\mathbb{E}_t\|\text{PE}(t)\|^2+\frac{\lambda_s^2(T^3-t_0^3)}{3(T-t_0)}\right)\frac{\sigma_X^2 d_X 2^{-d_X/2+1}}{\Gamma(d_X/2+1)}R^{d_X-2}\exp(-R^2/2\sigma_X^2) \\
&=O\left(\frac{\lambda_s^2 T^2\sigma_X^2 d_X^3 2^{-d_X/2+1}}{\Gamma(d_X/2+1)}R^{d_X}\exp(-R^2/2\sigma_X^2)\right).
\end{aligned}
$$

where the second-to-last inequality uses Eq. 48 and the last inequality applies Eq. 49.

Take $R:=O(\sqrt{d_X\log d_X+\log n})$ such that $(ii)\leq\frac{\lambda_s^2 d_X T^2}{n}$, and then combining $(i)$ and $(ii)$ yields with probability at least $1-\delta$,

$$
L_Z(\hat{U}',\hat{s}_Z')-L_Z(A_Z,s_Z^*)\leq O\left(\frac{1+\sqrt{\log 1/\delta}}{\sqrt{n}}(d_X^{5/2}+\log n)\right). \tag{50}
$$

Setting $\delta:=\frac{1}{n}$ and combining Eq. 47 and Eq. 50 yields with probability at least $1-O\left(\frac{1}{n}\right)$,

$$
L_Z(\hat{U},\hat{s}_Z)-L_Z(A_Z,s_Z^*)\leq O\left(\frac{1+\sqrt{\log n}}{\sqrt{n}}(d_X^{5/2}+\log n)\right)=O\left(\sqrt{\frac{d_X^5\log^3 n}{n}}\right).
$$

Combining this bound with Eq. 47 yields the desired bound. And similarly, we can prove the bound for $\hat{s}_G$.

## D.10 Proof of Lemma D.9

As $d_H, d_T \to \infty$, we have $K_Z = K_Z^*, K_G = K_G^*$. By the optimality of $\hat{s}_Z$ and $\hat{s}_G$ and Theorem D.3, we have

$$
\begin{aligned}
\hat{U}\hat{s}_Z(x^i,t^i) &= A_Z s_Z^*(x^i,t^i), \\
\hat{V}\hat{s}_G(g(x^i),t^i) &= A_G s_G^*(g(x^i),t^i), \\
\frac{1}{n}\sum_{i=1}^n \hat{s}_Z(x^i,t^i)\hat{s}_Z(x^i,t^i) &= \hat{U}^\top \hat{U}, \\
\frac{1}{n}\sum_{i=1}^n \hat{s}_G(g(x^i),t^i)\hat{s}_G(g(x^i),t^i) &= \hat{V}^\top \hat{V}, \forall i = 1,\cdots,n.
\end{aligned}
\tag{51}
$$

The first two equalities immediately yield

$$
\hat{L}_Z(\hat{U},\hat{s}_U) = \hat{L}_G(\hat{V},\hat{s}_V) = 0.
$$

To prove the last two inequalities in the lemma, let $\mathbb{E}_{\hat{p}_n}[f(x,t)] := \frac{1}{n}\sum_{i=1}^n f(x^i,t^i)$. Then Eq. 51 implies that

$$
\begin{aligned}
&\mathbb{E}_{\hat{p}_n}\|A_Z s_Z^*(x,t)\|^2 \\
&= \mathbb{E}_{\hat{p}_n}\text{Tr}(\hat{s}_Z^\top(x,t)\hat{U}^\top \hat{U}\hat{s}_Z(x,t)) = \text{Tr}\left(\mathbb{E}_{\hat{p}_n}\hat{s}_Z(x,t)\hat{s}_Z^\top(x,t)\hat{U}^\top\hat{U}\right) \\
&= \text{Tr}\left(\mathbb{E}_{\hat{p}_n}\hat{s}_Z(x,t)\hat{s}_Z^\top(x,t)\mathbb{E}_{\hat{p}_n}\hat{s}_Z(x,t)\hat{s}_Z^\top(x,t)\right) = \|\mathbb{E}_{\hat{p}_n}\hat{s}_Z(x,t)\hat{s}_Z^\top(x,t)\|^2 \geq \left(\mathbb{E}_{\hat{p}_n}\|\hat{s}_Z(x,t)\|^2\right)^2,
\end{aligned}
\tag{52}
$$

Let $\Sigma_n(s) := \mathbb{E}_{\hat{p}_n} s(x,t)s(x,t)^\top$ and $\lambda_i(M)$ be the $i$-th eigenvalue of the matrix $M$, then we have

$$
\begin{aligned}
\mathbb{E}_{\hat{p}_n}\|\hat{s}_Z(x,t)\|^2 &\leq \sqrt{\mathbb{E}_{\hat{p}_n}\|A_Z s_Z^*(x,t)\|^2} = \sqrt{\mathbb{E}_{\hat{p}_n}\|s_Z^*(x,t)\|^2} \\
&= \sqrt{\sum_{i=1}^{d_Z}\lambda_i(\Sigma_n(s_Z^*))} \leq \sqrt{\|\Sigma_n(s_Z^*)\|_{\text{op}}d_Z}.
\end{aligned}
\tag{53}
$$

Further, suppose $\hat{s}_Z(\tilde{x}) = \frac{1}{n}\sum_{i=1}^n K_Z(\tilde{x},\tilde{x}^i)c_Z(\tilde{x}^i)$ and define

$$
\bar{K}_Z := \begin{bmatrix} K_Z(\tilde{x}^1,\tilde{x}^1) & \cdots & K_Z(\tilde{x}^1,\tilde{x}^n) \\ \vdots & \ddots & \vdots \\ K_Z(\tilde{x}^n,\tilde{x}^1) & \cdots & K_Z(\tilde{x}^n,\tilde{x}^n) \end{bmatrix}, \bar{c}_Z := [c_Z(\tilde{x}^1)^\top,\cdots,c_Z(\tilde{x}^n)^\top]^\top,
$$

then we have

$$
\begin{aligned}
\|\hat{s}_Z\|_{K_Z}^2 &= \frac{1}{n}\bar{c}_Z^\top\bar{K}_Z\bar{c}_Z \leq \frac{\bar{c}_Z^\top\bar{K}_Z^2\bar{c}_Z}{n\lambda_n(\bar{K}_Z)} = \frac{1}{n\lambda_n(\bar{K}_Z)}\sum_{j=1}^n\left\|\sum_{i=1}^n\bar{K}_Z(\tilde{x}^i,\tilde{x}^j)c_Z(\tilde{x}^j)\right\|^2 \\
&= \frac{1}{\lambda_n(\bar{K}_Z)}\mathbb{E}_{\hat{p}_n}\|\hat{s}_Z(\tilde{x})\|^2 \leq \frac{1}{\lambda_{\min}(K_Z^*)}\mathbb{E}_{\hat{p}_n}\|\hat{s}_Z(\tilde{x})\|^2 \leq \frac{\sqrt{\|\Sigma_n(s_Z^*)\|_{\text{op}}d_Z}}{\lambda_{\min}(K_Z^*)}.
\end{aligned}
$$

It remains to prove the concentration of the operator norm $\|\Sigma_n(s_Z^*)\|_{\text{op}}$ around $\|\Sigma_\infty(s_Z^*)\|_{\text{op}} =: \|\Sigma(s_Z^*)\|_{\text{op}}$. To this end, we use the assumptions that $s_Z^*(x,t)$ is $\lambda_s$-Lipschitz and $(z,t)$ is sub-gaussian with sub-gaussian norm at most $\sigma_Z^2 d_Z + T^2$ to conclude that $s_Z^*(x,t)$ is sub-gaussian with sub-gaussian norm at most $\lambda_s(\sigma_Z^2 d_Z + T^2)$. Then Theorem 4.7.1 of [128] yields with probability at least $1 - 2\exp(-u)$,

$$
\|\Sigma_n(s_Z^*) - \Sigma(s_Z^*)\|_{\text{op}} \leq C\left(\sqrt{\frac{r+u}{n}} + \frac{r+u}{n}\right)\|\Sigma(s_Z^*)\|_{\text{op}},
\tag{54}
$$

where $r := \frac{\text{Tr}(\Sigma(s_Z^*))}{\|\Sigma(s_Z^*)\|_{\text{op}}}$ is the stable rank of $\Sigma(s_Z^*)$. As a result, with probability at least $1 - 2\exp(-n)$,

$$
\|\hat{s}_Z\|_{K_Z} \leq \frac{C_Z\|\Sigma(s_Z^*)\|_{\text{op}}^{1/4}d_Z^{1/4}}{\lambda_{\min}(K_Z^*)^{1/2}} = C_Z\sqrt{\frac{\sigma_1(s_Z^*)d_Z^{1/2}}{\lambda_{\min}(K_Z^*)}},
$$

where $C_Z := 1 + C^{1/2}(\sqrt{r+1}+r+1)^{1/2}$ suffices.

Finally, using a similar argument we can show that

$$\|\hat{s}_G\|_{K_G} \leq C_G \sqrt{\frac{\sigma_1(s_G^*)d_G^{1/2}}{\lambda_{\min}(K_G^*)}}$$

with probability at least $1 - 2\exp(-n)$.

## D.11 Proof of Lemma D.10

As $d_H, d_T \to \infty$, we have $K_Z = K_Z^*, K_G = K_G^*$. By definition of the NTKs,

$$
\begin{aligned}
\|K_Z(x,t,x,t)\| &= \left\|\mathbb{E}_{\mathcal{N}(0,I)}\|a(x,t)\|^2 I_{d_Z} + \frac{1}{2}\|\tilde{x}\|^2 I_{d_Z}\right\| \\
&= \left\|\mathbb{E}_{\mathcal{N}(0,I)}\|a(x,t)\|^2 + \frac{1}{2}\|\tilde{x}\|^2\right\| d_Z \\
&= \left\|\lim_{d_H \to \infty} \frac{1}{d_H}\sum_{j=1}^{d_H} \mathbb{E}_{\mathcal{N}(0,I)}(\tilde{x}^\top \Theta_j^{(1)})_+ (\Theta_j^{(1)\top}\tilde{x})_+ + \frac{1}{2}\|\tilde{x}\|^2\right\| d_Z \\
&\leq \left\|\lim_{d_H \to \infty} \frac{1}{d_H}\sum_{j=1}^{d_H} \mathbb{E}_{\mathcal{N}(0,I)}(\tilde{x}^\top \Theta_j^{(1)}\Theta_j^{(1)\top}\tilde{x})_+ + \frac{1}{2}\|\tilde{x}\|^2\right\| d_Z = \left\|\|\tilde{x}\|^2 + \frac{1}{2}\|\tilde{x}\|^2\right\| = \frac{3}{2}\|\tilde{x}\|^2.
\end{aligned}
$$

Similarly, we can prove the bound for $K_G$.

Table 2: **Quantitative results for image disentanglement on MNIST and CIFAR10 test sets**. MSE, PSNR and SSIM stands for mean squared error, peak signal-to-noise ratio and structural similarity respectively between the generated and target samples. LPIPS [75] is a perceptual metric based on features from deep image classifiers. The results are averaged over two random trials.

| | MNIST | | | | CIFAR10 | | | |
|---|---|---|---|---|---|---|---|---|
| | MSE($\downarrow$) | PSNR($\uparrow$) | SSIM($\uparrow$) | LPIPS($\downarrow$) | MSE($\downarrow$) | PSNR($\uparrow$) | SSIM($\uparrow$) | LPIPS($\downarrow$) |
| $\lambda_r = 0$ | $0.19_{\pm 0.01}$ | $5.2_{\pm 0.2}$ | $0.42_{\pm 0.04}$ | $0.30_{\pm 0.02}$ | $0.44_{\pm 0.03}$ | $3.5_{\pm 0.2}$ | $0.05_{\pm 0.00}$ | $0.62_{\pm 0.01}$ |
| $\lambda_r = 0.03$ | $0.07_{\pm 0.04}$ | $9.6_{\pm 2.3}$ | $0.53_{\pm 0.06}$ | $0.18_{\pm 0.04}$ | $0.35_{\pm 0.02}$ | $4.4_{\pm 0.1}$ | $0.05_{\pm 0.00}$ | $0.61_{\pm 0.02}$ |
| $\lambda_r = 0.3$ | $0.01_{\pm 0.00}$ | $16.0_{\pm 0.3}$ | $0.66_{\pm 0.00}$ | $0.1_{\pm 0.00}$ | $0.18_{\pm 0.06}$ | $7.3_{\pm 0.4}$ | $0.06_{\pm 0.00}$ | $0.53_{\pm 0.02}$ |
| $\lambda_r = 3$ | $\mathbf{0.01}_{\pm 0.00}$ | $\mathbf{17.3}_{\pm 0.3}$ | $\mathbf{0.66}_{\pm 0.01}$ | $\mathbf{0.1}_{\pm 0.00}$ | $\mathbf{0.11}_{\pm 0.01}$ | $\mathbf{9.3}_{\pm 0.2}$ | $\mathbf{0.07}_{\pm 0.00}$ | $\mathbf{0.49}_{\pm 0.00}$ |

## E Experiment details

### E.1 Latent subspace GMM disentanglement

For the synthetic disentanglement experiment, we choose the subspace dimension to be $d_Z = d_G = 5$ and sample the content variable via $Z \sim \frac{1}{2}\mathcal{N}(\mu_1^Z, \sigma_0^2 \mathbf{I}_{d_Z}) + \frac{1}{2}\mathcal{N}(\mu_2^Z, \sigma_0^2 \mathbf{I}_{d_Z})$ and the style variable via $G \sim \frac{1}{2}\mathcal{N}(\mu_1^G, \sigma_0^2 \mathbf{I}_{d_G}) + \frac{1}{2}\mathcal{N}(\mu_2^G, \sigma_0^2 \mathbf{I}_{d_G})$, where $\sigma_0 = 0.1$. In this way, we generate i.i.d $4000$ samples for training.

We follow the network architecture shown in Figure 2 with $d_H = 512$. For the time embedding $\text{PE}(\cdot)$, we opt for a *Gaussian Fourier projection layer* to encode temporal information between $(0,1]$ defined as:

$$\text{PE}(t) := \begin{bmatrix} \sin(2\pi\Omega t) \\ \cos(2\pi\Omega t) \end{bmatrix},$$

where $\Omega \sim \mathcal{N}(\mathbf{0}_{512}, 9000 I_{512})$. We train the models for 10,000 steps with an Adam [131] optimizer with learning rate $10^{-5}$ and batch size equal to the entire training set.

Table 10: **The default noise schedule hyperparameters** for the synthetic data experiments. Continuous time ($t \in [10^{-5}, 1]$) is used in the expression.

| Name | $\alpha(t)$ | $\sigma^2(t)$ |
|---|---|---|
| VE | $1$ | $\frac{25^{2t}-1}{2\log 25}$ |
| VP | $e^{-0.05t-4.975t^2}$ | $1 - e^{-0.1t-9.95t^2}$ |
| sub-VP | $e^{-0.05t-4.975t^2}$ | $(1 - e^{-0.1t-9.95t^2})^2$ |
| VP (cosine) | $e^{-\frac{t}{2}-\frac{1}{\pi}\sin(\frac{t}{2})}$ | $1 - e^{-t-\frac{2}{\pi}\sin(\frac{t}{2})}$ |

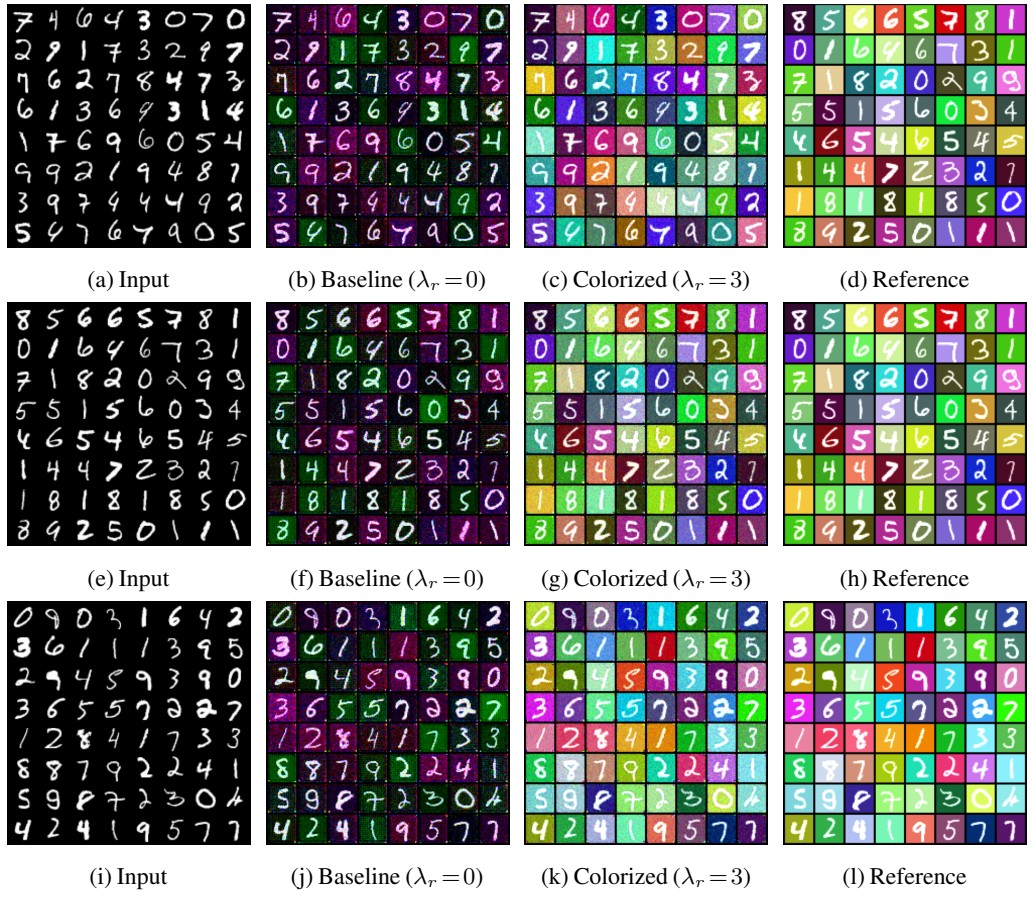

|  |  |  |  |
|---|---|---|---|
| (a) Input | (b) Baseline ($\lambda_r = 0$) | (c) Colorized ($\lambda_r = 3$) | (d) Reference |
| (e) Input | (f) Baseline ($\lambda_r = 0$) | (g) Colorized ($\lambda_r = 3$) | (h) Reference |
| (i) Input | (j) Baseline ($\lambda_r = 0$) | (k) Colorized ($\lambda_r = 3$) | (l) Reference |

Figure 6: More image colorization results on MNIST

To ensure convergence, we pretrained the speaker score network $\hat{s}^G$ We experiment with various noise schedulers, including the variance exploding (VE), vanilla variance preserving (VP) [33], sub-VP [74] and cosine VP [132]. The detailed schedule hyperparameters are listed in Table 10 and are chosen based on rules of thumbs in [133, 74]. To evaluate the subspace recovery, we use the subspace recovery error defined as:

$$d(U,V,A_Z,A_G) := \frac{1}{2d_X}\big(\|P_U - P_{A_Z}\|_F^2 + \|P_V - P_{A_G}\|_F^2\big), \tag{55}$$

whose range is $[0,1]$.

### E.2 Image disentanglement

For all experiments, we use a Gaussian Fourier projection layer to encode temporal information between $(0,1]$. Further, the U-Net architecture used as the score function for MNIST and CIFAR10 are shown in Table 3 and Table 4 respectively. To capture the content information of the image, we use the $16 \times 16$ feature map from a pretrained `vit-small-patch16-224-dino` variant of the DINO model [56]. The feature map is then resampled to the same size as the image.

For both datasets, we train the DM using an Adam optimizer [131] with a batch size of 128 and a learning rate $10^{-4}$ for 50 epochs. A VE schedule is used during conditional score matching. During inference, we use probability flow [74] with 500 steps to perform sampling. For the CIFAR10 denoising experiment, we feed a all-zero matrix as the noise map. All models are implemented in Pytorch [134] on two A5000 GPUs. The training time is approximately an hour for both datasets and the inference is approximately 10 seconds for 64 samples.

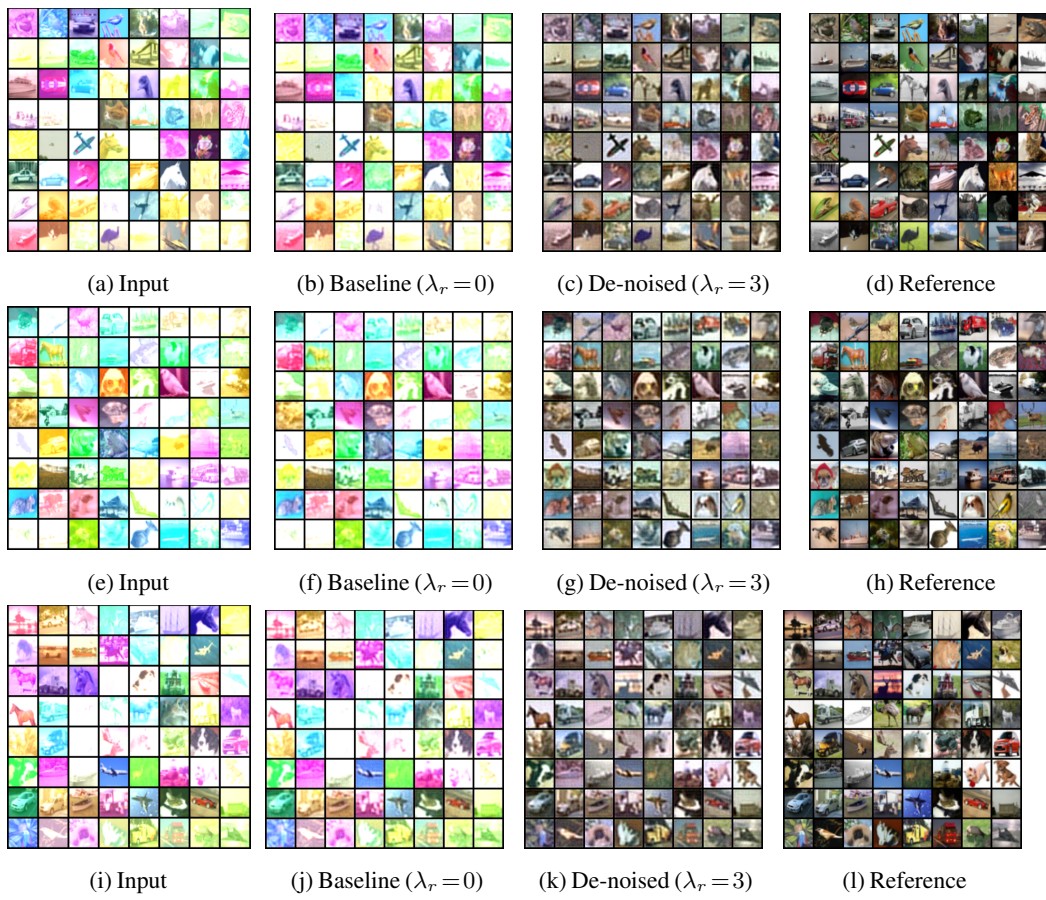

| (a) Input | (b) Baseline ($\lambda_r = 0$) | (c) De-noised ($\lambda_r = 3$) | (d) Reference |

| (e) Input | (f) Baseline ($\lambda_r = 0$) | (g) De-noised ($\lambda_r = 3$) | (h) Reference |

| (i) Input | (j) Baseline ($\lambda_r = 0$) | (k) De-noised ($\lambda_r = 3$) | (l) Reference |

Figure 7: More image denoising results on CIFAR10

Full quantitative results on MNIST and CIFAR10 are shown in Table 2. And more examples are shown in Figure 6 for MNIST and Figure 7 for CIFAR10.

Table 3: **U-Net score network used in the MNIST colorization experiment**. The input is $9 \times 28 \times 28$ created by stacking the image, the projected and resized DINO feature map to $3 \times 28 \times 28$ and the background color vector broadcasted to $3 \times 28 \times 28$. $a+b$ denotes that component $a$ accepts hidden embedding from the previous layer and component $b$ accepts the time embedding, and the outputs of $a$ and $b$ are added with proper broadcasting. "Conv2d" refers to 2-D convolutional layer, "GroupNorm" stands for group normalization layer, and "ConvTrans2d" stands for 2-D transposed convolutional layer.

| MNIST U-Net |
| --- |
| $384 \times 3 \times 1 \times 1$ Conv2d with stride 1 (DINO projection layer) |
| $9 \times 32 \times 3 \times 3$ Conv2d with stride 2 + $1024 \times 32$ Linear |
| GroupNorm with 4 groups 
 Swish activation |
| $32 \times 64 \times 3 \times 3$ Conv2d with stride 2 + $1024 \times 64$ Linear |
| GroupNorm with 32 groups 
 Swish activation |
| $64 \times 128 \times 3 \times 3$ Conv2d with stride 2 + $1024 \times 128$ Linear |
| GroupNorm with 32 groups 
 Swish activation |
| $128 \times 256 \times 3 \times 3$ Conv2d with stride 2 + $1024 \times 256$ Linear |
| GroupNorm with 32 groups 
 Swish activation |
| $256 \times 128 \times 4 \times 4$ ConvTrans2d with stride 2 + $1024 \times 128$ Linear |
| GroupNorm with 32 groups 
 Swish activation |
| $256 \times 64 \times 4 \times 4$ ConvTrans2d with stride 2 + $1024 \times 64$ Linear |
| GroupNorm with 32 groups 
 Swish activation |
| $128 \times 32 \times 4 \times 4$ ConvTrans2d with stride 2 + $1024 \times 32$ Linear |
| GroupNorm with 32 groups 
 Swish activation |
| $64 \times 3 \times 3 \times 3$ ConvTrans2d with stride 1 |

### E.3 Speech disentanglement

The dataset statistics are shown in Table 5. The overall results for realistic datasets are shown in Table 6

For the IEMOCAP dataset, we use a system available on SpeechBrain [135] that finetunes on the wav2vec 2.0 backbone [136] with a multi-layer perceptron classifier (MLP) [137]. The classifier is trained using Adam optimizer for 30 epochs with a batch size of $4$ and a learning rate of $10^{-4}$ for the MLP and the $10^{-5}$ learning rate for wav2vec 2.0 weights. The system is then evaluated using the standard classification accuracy metric and 5-fold cross validation [76, 138]. For each fold, we use all 8 speakers from the training set as target speakers.

On the ALS and ADReSS, we use whisper-medium [129] features, as they have shown to be the most effective for speech impairment classification [139]. To avoid unfair comparison, We concatenate hidden representations over *all* layers of the whisper-medium encoder rather than selecting a particular layer and perform mean pooling over the frame-level features. For both datasets, we follow the standard splits used in previous works [79] to have no overlaps between speaker in the training and test sets. And for both datasets, we use the 15 most frequent speakers from the training set as target speakers for the VC to achieve maximize conversion quality via better speaker representation.

Table 4: **U-Net score network used in the CIFAR10 denoising experiment**. The input is $12 \times 32 \times 32$ created by stacking the image, the projected and resized DINO feature map to $3 \times 32 \times 32$ and the noise map broadcasted to $3 \times 32 \times 32$. A dual encoder with separate U-Nets for the content and the style variables is used and $a+b+c$ denotes that component $a$ accepts the content embedding from the previous layer, component $b$ accepts the previous style embedding and $c$ accepts the time embedding. Further, the outputs of $a,c$ and the outputs of $b,c$ are added separately with proper broadcasting. $a+b$ here denotes that $a$ accepts the previous content embedding, $b$ accepts the previous style embedding and the two outputs are added.

| CIFAR10 U-Net |
| :---: |
| $384 \times 6 \times 1 \times 1$ Conv2d with stride 1 (DINO projection layer) |
| $6 \times 32 \times 3 \times 3 + 3 \times 32 \times 3 \times 3$ Conv2d with stride 2 + $1024 \times 32$ Linear |
| GroupNorm with 4 groups
Swish activation |
| $32 \times 64 \times 3 \times 3 + 32 \times 64 \times 3 \times 3$ Conv2d with stride 2 + $1024 \times 64$ Linear |
| GroupNorm with 32 groups
Swish activation |
| $64 \times 128 \times 3 \times 3 + 64 \times 128 \times 3 \times 3$ Conv2d with stride 2 + $1024 \times 128$ Linear |
| GroupNorm with 32 groups
Swish activation |
| $128 \times 256 \times 3 \times 3 + 128 \times 256 \times 3 \times 3$ Conv2d with stride 2 + $1024 \times 256$ Linear |
| GroupNorm with 32 groups
Swish activation |
| $256 \times 128 \times 4 \times 4 + 256 \times 128 \times 4 \times 4$ ConvTrans2d with stride 2 + $1024 \times 128$ Linear |
| GroupNorm with 32 groups
Swish activation |
| $256 \times 64 \times 4 \times 4 + 256 \times 64 \times 4 \times 4$ ConvTrans2d with stride 2 + $1024 \times 64$ Linear |
| GroupNorm with 32 groups
Swish activation |
| $128 \times 32 \times 4 \times 4 + 128 \times 32 \times 4 \times 4$ ConvTrans2d with stride 2 + $1024 \times 32$ Linear |
| GroupNorm with 32 groups
Swish activation |
| $64 \times 3 \times 3 \times 3 + 64 \times 3 \times 3 \times 3$ ConvTrans2d with stride 1 |

Table 5: **Datasets and VC-adapted classifiers used during realistic data experiments**

| | $|\mathcal{Y}|$ | Feature | Classifier | #Classifiers | Reference | VC | DM-based | Reference |
| --- | --- | --- | --- | --- | --- | --- | --- | --- |
| IEMOCAP | 4 | wav2vec 2.0 base | MLP | 8 | [76] | TriAAN-VC | No | [77] |
| ADReSS | 2 | whisper-medium | SVM | 15 | [78] | KNN-VC | No | [65] |
| ALS-TDI | 5 | whisper-medium | SVM | 15 | [79] | Diff-VC | Yes | [38] |

We apply the VCs in mostly a zero-shot, plug-and-play fashion, and leave finetuning to specific datasets for future works. For the Diff-VC, we use the publicly available score network and vocoder checkpoints trained on LibriTTS and adopt the original inference hyperparameter settings for all experiments. Similarly, we use the pretrained models and for other VC models. Further, we use a maximum of 120 second speech from the target speaker to compute the target speaker embedding for all models except KNN-VC, where we use all the target speech as the pool for nearest neighbor search. We also compare VC adaptation with common data augmentation technique such as pitch shifting, where we shift the pitch of all the speech utterances to equally spaced pitch levels over the $F0$ range of the training speech

Table 6: **Overall results on realistic datasets**. All metrics are between 0-100. A: single (average); B: single (best); MV: majority vote; SMV: soft majority vote.

| VC type | Impairment | | | | | | | | Emotion | | | |
| | ALS-TDI, F1↑ | | | | ADReSS, F1↑ | | | | IEMOCAP, Acc. (5-fold)↑ | | | |
| | A | B | MV | SMV | A | B | MV | SMV | A | B | MV | SMV |
|---|---|---|---|---|---|---|---|---|---|---|---|---|
| No VC | 54.9 | 54.9 | 54.9 | 54.9 | 70.6 | 70.6 | 70.6 | 70.6 | 71.5 | 71.5 | 71.5 | 71.5 |
| Pitch shifting | 55.8 | 60.3 | 57.6 | 61.5 | 71.2 | 77.1 | 77.1 | 68.8 | 60.6 | 55.1 | 61.1 | 61.1 |
| KNN-VC | 55.8 | 61.7 | 64.8 | 49.9 | 71.5 | 79.2 | 79.2 | 83.3 | 70.4 | 69.3 | 71.4 | 71.5 |
| TriAAN-VC | 55.7 | 60.7 | 61.7 | 53.3 | 72.4 | 75.0 | 77.1 | 83.3 | 65.1 | 64.1 | 66.8 | 67.2 |
| Diff-VC | 47.0 | 51.2 | 50.3 | 49.2 | 65.6 | 69.4 | 66.7 | 70.8 | 87.0 | 94.3 | 96.5 | 97.2 |

Table 7: **Emotion recognition results on IEMOCAP**. 8 speakers in the training set of each fold are used as target speakers. WER stands for word error rate computed using the `whisper-large-v2` model [129] and lower is better. MOS stands for mean opinion score computed using UTMOS [130] and higher is better.

| VC type | Voting type | Accuracy | | | | | | WER (↓) | MOS (↑) |
| | | 1 | 2 | 3 | 4 | 5 | Avg. | | |
|---|---|---|---|---|---|---|---|---|---|
| No VC | - | 72.6 | 76.6 | 68.9 | 68.9 | 70.3 | 71.5 | 9.0 | 2.3 |
| Pitch shifting | single (best) | 64.0 | 65.3 | 57.4 | 57.7 | 58.6 | 60.6 | 36.4 | 1.3 |
| | single (avg) | 61.3 | 62.1 | 50.2 | 48.9 | 53.0 | 55.1 | | |
| | majority | 61.2 | 65.3 | 58.5 | 57.8 | 62.5 | 61.1 | | |
| | soft majority | 60.8 | 65.4 | 57.8 | 57.5 | 61.5 | 61.1 | | |
| KNN-VC | single (best) | 71.2 | 75.4 | 68.3 | 71.9 | 69.1 | 70.4 | 9.8 | 1.6 |
| | single (avg) | 69.6 | 72.6 | 67.0 | 69.9 | 67.4 | 69.3 | | |
| | majority | 70.3 | 75.6 | 68.5 | 72.8 | 70.0 | 71.4 | | |
| | soft majority | 70.3 | 76.1 | 68.5 | 72.8 | 69.9 | 71.5 | | |
| TriAAN-VC | single (best) | 65.5 | 66.9 | 63.0 | 67.5 | 62.6 | 65.1 | 16.6 | 1.9 |
| | single (avg) | 64.6 | 66.3 | 61.1 | 65.6 | 63.1 | 64.1 | | |
| | majority | 66.9 | 69.0 | 63.9 | 67.9 | 66.5 | 66.8 | | |
| | soft majority | 66.6 | 69.9 | 63.6 | 68.8 | 67.0 | 67.2 | | |
| Diff-VC | single (avg) | 87.0 | 88.3 | 86.2 | 87.6 | 86.1 | 87.0 | 22.9 | 2.5 |
| | single (best) | 94.4 | 94.8 | 94.2 | 95.2 | 92.9 | 94.3 | | |
| | majority | 97.5 | 96.7 | 95.2 | 98.1 | 94.9 | 96.5 | | |
| | soft majority | 97.7 | 97.6 | 96.3 | 98.7 | 95.6 | 97.2 | | |

data with levels equal to the number of target speakers and train separate classifiers for each level as in the case of using VC adaptation.

For ALS severity classification as shown in Table 6, KNN-VC achieves the best performance among the VCs, reaching 65% macro-F1 with 15 target speakers and hard majority voting, compared to 54.9% when training without VC adaptation and 61.7% with pitch shifting. For cognitive impairment detection as shown in Table 6, TriAAN-VC performed the best followed by the KNN-VC method, both achieved 83.3% macro-F1 with soft majority voting, which is 12.7% better than methods without VC adaptation and 14.5% and 6.2% better than the pitch shifting adaptation using hard and soft majority voting respectively. On IEMOCAP, we found that Diff-VC performs the best, reaching an average of 97.2% accuracy, which is 25.7% better than the no-VC classifier and 36.1% than the pitch shifting adaptation. Though a phenomenon out of the scope of predictions by our theory, we hypothesized that such "specialization" of the VC methods is due to the different level of generalization ability of different VCs to latent variables *other than* the speaker identity, such as recording conditions and health conditions of the speaker. For instance, Diff-VC does not perform well on ALS compared to KNN-VC, probably due to the domain mismatch between the health conditions of its training set, which contains little pathological speech, compared to KNN-VC which uses the WavLM representation trained on much larger speech dataset with diverse speech.

Table 8: Alzheimer detection results on ADReSS

| VC type | Voting type | Precision | Recall | F1 | Accuracy |
|---------|-------------|-----------|--------|-----|----------|
| No VC | - | 71.4 | 70.8 | 70.6 | 70.8 |
| Pitch shifting | single (avg) | 71.8 | 71.4 | 71.4 | 71.2 |
| | single (best) | 77.1 | 77.1 | 77.1 | 77.1 |
| | majority | 77.1 | 77.1 | 77.1 | 77.1 |
| | soft majority | 68.8 | 68.8 | 68.7 | 68.8 |
| KNN-VC | single (avg) | 71.8 | 71.5 | 71.4 | 71.5 |
| | single (best) | 80.0 | 79.2 | 79.1 | 79.2 |
| | majority | 79.4 | 79.2 | 79.1 | 79.2 |
| | soft majority | 83.6 | 83.3 | 83.3 | 83.3 |
| TriAAN-VC | single (avg) | 72.5 | 72.4 | 72.4 | 72.4 |
| | single (best) | 75.2 | 75.0 | 75.0 | 75.0 |
| | majority | 77.5 | 77.1 | 77.0 | 77.1 |
| | soft majority | 83.3 | 83.3 | 83.3 | 83.3 |
| Diff-VC | single (avg) | 65.7 | 65.4 | 65.4 | 65.6 |
| | single (best) | 69.4 | 69.4 | 69.4 | 69.4 |
| | majority | 66.7 | 66.7 | 66.7 | 66.7 |
| | soft majority | 72.2 | 70.8 | 70.4 | 70.8 |

Table 9: ALS severity classification results on ALS-TDI with a whisper-medium+SVM classifier

| VC type | Voting type | Precision↑ | Recall↑ | F1↑ |
|---------|-------------|-----------|---------|-----|
| No VC | - | 59.8 | 53.7 | 54.9 |
| Pitch shifting | single (avg.) | 60.5 | 54.1 | 55.8 |
| | single (best) | 67.4 | 57.7 | 60.3 |
| | majority | 73.0 | 54.9 | 57.6 |
| | soft majority | 68.4 | 59.0 | 61.5 |
| KNN-VC | single (avg.) | 58.5 | 54.6 | 55.8 |
| | single (best) | 65.7 | 59.5 | 61.7 |
| | majority | 67.9 | 62.9 | 64.8 |
| | soft majority | 51.1 | 49.6 | 49.9 |
| TriAAN-VC | single (avg.) | 60.2 | 54.5 | 55.7 |
| | single (best) | 68.0 | 58.2 | 60.7 |
| | majority | 69.9 | 59.1 | 61.7 |
| | soft majority | 54.1 | 52.8 | 53.3 |
| Diff-VC | single (avg.) | 48.2 | 47.0 | 47.0 |
| | single (best) | 53.0 | 51.0 | 51.2 |
| | majority | 49.8 | 50.9 | 50.3 |
| | soft majority | 50.1 | 48.8 | 49.2 |

As to the advantage of hard vs. soft voting, we observe different trends across different datasets and VC methods. On ALS-TDI, hard voting works better than soft voting by 8.4% and 16% for the best two methods Diff-VC and KNN-VC, though worse by 3.9% and 1.3% for pitch shifting and Diff-VC. On IEMOCAP, the gap between soft and hard voting is negligible, with soft majority voting shows a 0.1%-0.7% edge over hard majority voting across VC methods. On ADReSS, we found soft voting methods to be better than hard voting for all the VC methods by $4.1\% - 6.2\%$, while worse for the pitch shifting method by 8.3% (68.8% vs. 77.1%). Since soft voting uses a random classifier for voting, it tends to perform well when the model is "confidently" correct and "hesitantly" wrong, as it puts more weights on confident classifiers than hesitant ones. This suggests that the average confidence score estimated in terms of the classifier posteriors on incorrect examples will be high for classifier

ensembles that perform well with hard voting than soft voting. Table 7 8 9 show the complete results for the realistic dataset experiments.

