# OpenReview forum: "Can Diffusion Models Disentangle? A Theoretical Perspective"
_NeurIPS.cc/2025/Conference — NeurIPS 2025 poster_

### Official Review · Reviewer_34h9 · 2025-07-02

**Clarity:** 1
**Significance:** 3
**Originality:** 3
**Rating:** 4
**Confidence:** 2

**Summary:**

The paper presents a theoretical framework for disentangling content and style in diffusion models. The authors first introduce two information-theoretic concepts called ($\epsilon, \nu$)-disentanglement and $\epsilon$-editability. The former states that the restored content and style are approximately statistically independent, while the letter is satisfied if a change of style leads to a data distribution with unchanged content. Subsequently, two settings (access to the ground truth style and multiple views with shared content) and technical conditions are presented under which disentangling and editability can theoretically be achieved.

**Questions:**

* In Equation 7, what forces $I(\hat{Z},\hat{G};X)=I(Z,G;X)$ as $\delta \to 0$? Why can $\hat{Z}$ not just be constant?
* Why is the ground-truth decoder/generator $f$ in the score function in line 1122 and line 1204? Where is this equation for the score function coming from?
* In line 1077 ,where is $C_2$ used?
* How is the estimated content, $\hat{Z}$, used in equation 13 for the experiments?
* Why is the background in the CIFAR10 colorization task in the input always black? Shouldn't this also work with other background colors in the input?
* How do ($\epsilon, \nu$)-disentanglement and $\epsilon$-editability relate to deterministic formalizations of disentanglement? That is, there exist functions $h_1$, $h_2$ such that $\hat{Z}=h_1(Z)$ and $\hat{G}=h_2(G)$.

**Ethical Concerns:**

["NO or VERY MINOR ethics concerns only"]

**Final Justification:**

I now have a clearer understanding of the connection between theory and experiments, and my main concern regarding clarity has been largely addressed by the proposed revisions. While the practical applicability remains limited, I believe the theoretical contribution provides a novel and valuable perspective on disentanglement. I am therefore also willing to raise my score to 4.

**Limitations:**

yes

**Paper Formatting Concerns:**

-

**Quality:**

2

**Strengths And Weaknesses:**

**Strengths:**
* The paper adresses the problem of disentanglement in diffusion models, which is an important topic in generative modelling.
* The paper provides a novel view on disentanglement from the lense of information theory

**Weaknesses:**
At first glance, the theoretical results seem plausible, so I tried to follow the theoretical derivations in the appendix, but due to time constraints I could only look at a small part, as the derivations are quite extensive. Some equations seem to come out of nowhere (see questions below). So I have some concerns about the validity, but it is possible I didn't understand some parts.
* The experimental part is difficult to follow. It is not clear to me from the short description how the datasets look like and how the results were obtained. The paper would benefit greatly from additional clarifications of the experimental setups.
* There seems to be a gap between the theoretical contribution of the paper and its empirical validation. The presented theorems are based on the losses in Equation 7, 8, and 11, but the experiments are based on the loss in equation 13 without sufficient explanation how it relates.
* It would improve readability of the appendix if theorems and lemmas were restated before the proofs. Otherwise one needs to switch back and forth quite a lot.

---

> ### Author Rebuttal · Authors · 2025-07-30
>
> Dear reviewer 34h9,
>
> We thank you for the thoughtful comments and for recognizing the originality and importance of our work. We appreciate your engagement with both the theoretical and empirical parts of the paper, and we address your concerns below.
>
> **1. On experimental details in Section 5**
>
> We defer the experimental details to the Appendix E to focus on the exposition of our theoretical results. But we will consider adding the dataset description back to the main text in the extra space of the camera-ready version of the paper.
>
> **2. On theoretical-empirical gaps and the role of Eq. (13)**
>
> We appreciate this concern and agree it deserves clarification. Eq. (13) is a **practical implementation** of Eq. (11), tailored for high-dimensional image/speech data. Specifically:
>
> * We omit the balancing term for empirical simplicity;
>
> * Style guidance is applied by zeroing out content, equivalent to the style regularizer in Eq. (11) when the input data has ISM structure;
>
> * For MNIST colorization/CIFAR-10 denoising/voice conversion, the ISM assumption is not explicitly enforced, but the structure is approximately satisfied in these tasks either directly or after the input is transformed into self-supervised representation [1,2].
>
> Thus, Eq. (13) is not theoretically disjoint from Eq. (11)—rather, it's an **approximated empirical proxy** aligned with our theoretical principles. We will make this relationship clearer in Sec. 5 and Appendix E.
>
> [1] O. D. Liu, H. Tang, and S. Goldwater, “Self-supervised predictive coding models encode speaker and phonetic information in orthogonal subspaces,” in Interspeech, 2023.
>
> [2] Kamper, Herman, Benjamin van Niekerk, Julian Zaïdi, and Marc-André Carbonneau. “LinearVC: Linear Transformations of Self-Supervised Features through the Lens of Voice Conversion,” 2025.
>
> **3. On Eq. (7): Why not make $z_{\phi}(X)$ constant**
>
> We appreciate the opportunity for clarifying. While the regularizer in Eq. (7) enforces an upper bound on mutual information between content and input, the **score matching loss term** still requires $z_{\phi}(X)$ to be predictive of the data’s conditional score. If $z_{\phi}(X)$ were constant, the model could not satisfy the conditional score matching objective. Thus, **the interplay between the loss and regularizer** ensures that $z_{\phi}(X)$ retains just enough content information for prediction without overfitting. This tension promotes disentanglement.
>
> **4. On score function using ground-truth generator (lines 1122 and 1204)**
>
> In Appendix A and B, we use the true generative function $f(Z,G)$ only for **theoretical analysis**, such as constructing counterexamples (e.g., for demonstrating non-editability in Example 1) or establishing upper bounds of the optimal conditional score matching loss in the proof of Theorem 4.1 and Theorem 4.2. In practice, the model learns a score estimator $s_{\theta},$ and the generator is not known. We will clarify in the text that these instances refer to **analytical derivations** only, not model implementation.
>
> **5. On use of $C_2$ in line 1077**
>
> It is used in line 1086, but we agree it appears too early, and will consider moving it to the later part of the proof.
>
> **6. On use of $z_{\phi}(X)$ in Eq. (13)**
>
> In Eq. (13), $z_{\phi}(X)$ is passed to the score estimator $s_{\theta}(X_t, z_{\phi}(X), G, t).$ The **second term** in the loss zeroes out $z_{\phi}(X)$ to encourage style-specific signal separation (style guidance). The loss structure is consistent with Eq. (11) but omits the balancing term since our model is able to work without it empirically.
>
> **7. On CIFAR10 background being black**
>
> In our CIFAR-10 experiments (Sec. 5, Fig. 3), we perform **denoising**, not colorization and the input is not always black. The colorization task is conducted on MNIST, which is gray-scale. In principle, our method works for any background color.
>
> **8. On relation to deterministic disentanglement formalizations**
>
> Thank you for asking this important question. Classical deterministic disentanglement assumes that **invertible functions** exist to recover latent variables (e.g., $Z=z(X),\,G=g(X)$). However, diffusion models are stochastic and non-invertible, making such assumptions inapplicable.  Our proposed $(\epsilon,\nu)$-disentanglement and $\epsilon$-editability generalize these notions to stochastic settings by relying on **mutual information** and **distributional distances** rather than pointwise mappings.
>
> These criteria reduce to classical notions in the **low-noise limit**, where DMs become nearly deterministic and when the content and style variables are discrete. We will add a paragraph in Section 3 connecting our framework to deterministic disentanglement theories.
>
> **9. On theoretical derivation in the appendix**
>
> Thank you for this candid feedback. We understand that Appendix B–D contains dense derivations. We will improve the clarity by:
>
> * Restating the theorems and definitions before proofs.
>
> * Adding brief and intuitive summaries before technical steps
>
> * Clarifying notation and referencing which result each equation builds on
>
> ------------------
> We thank you again for the insightful comments and hope our response helps to address your concern and makes our paper more suited for the NeurIPS conference. Happy to discuss further and consider further revisions.

---

> > ### Comment · Reviewer_34h9 · 2025-08-04
> >
> > Thank you very much for the detailed response.
> >
> > I appreciate that the authors acknowledge the need for further clarification in the theoretical derivations. Since the main contribution of this work is theoretical, I believe it is essential that the proof steps are clearly explained.
> >
> > The proposed addition of connecting the novel framework to deterministic disentanglement would also be valuable for contextualization—especially if the deterministic setting cannot be shown to emerge as a special case in the zero-noise limit. From my perspective, a natural extension of the deterministic disentanglement formalization (of content and style) to an information-theoretic one would be: $I(G; X \mid \hat{G}) \to 0$ and $I(Z; X \mid \hat{Z}) \to 0$ as noise approaches zero, since the two formulations become essentially equivalent in the limit. It would be interesting to explore how this relates to the notions of $(\epsilon, \nu)$-disentanglement and editability, and whether a similar form of equivalence could be formally established.
> >
> > I also believe the experimental section would benefit from further clarification, particularly regarding the design choices and how each experiment supports specific components of the theoretical framework. To compensate for this, most of the background to diffusion models could be moved to the appendix.
> >
> > I remain unclear about why, in the image experiments, the input to $z_{\phi}(\cdot)$ already corresponds to the content component. Wouldn’t the model’s disentanglement and editability capabilities be more convincingly demonstrated if it could generalize to inputs with arbitrary background colors (MNIST) or varying noise levels (CIFAR-10), for example, by recoloring or denoising from arbitrary inputs?
> >
> > I will leave my score unchanged because I think the clarity could be substantially improved.

---

> > > ### Author Response · Authors · 2025-08-05
> > >
> > > Thank you for acknowledging the significance of our work and for providing suggestions to improve the clarity of our paper. We respond to your points below and appreciate the opportunity to further clarify the scope and intention of our work.
> > >
> > > **1. On connection to deterministic disentanglement**
> > >
> > > We appreciate your thoughtful suggestion to connect our framework to deterministic disentanglement. We agree that exploring the limit behavior of $(\epsilon,\nu)$-disentanglement and editability under vanishing noise is a compelling direction. However, we would like to clarify that the proposed conditions,
> > > $$I(G;X|\hat{G})=D_{KL}(p_{GX|\hat{G}}, p_{G|\hat{G}}p_{X|\hat{G}})\rightarrow 0$$ and $$I(Z;X|\hat{Z})=D_{KL}(p_{ZX|\hat{Z}}, p_{Z|\hat{Z}}p_{G|\hat{Z}})\rightarrow 0,$$
> > > are in general **unachievable under our current assumptions**, even with known style function and the ISM setting. These conditions essentially require the learned representations $\hat{G}$ and $\hat{Z}$ to act as **sufficient statistics** for the style and content components of $X$, respectively. In contrast, our work formulates disentanglement through **score-based representations** rather than full generative invertibility to avoid direct density modeling, which makes vanishing conditional MI generally unprovable.
> > >
> > > That said, we agree that a **relaxed formulation** is **feasible in the multi-view setting**, if one measures proximity using  **total variation distance**. In fact, under the assumptions of Theorem 4.1 and 4.2, our framework implicitly encourages the **conditional distribution** $p_{X|\hat{Z}}$ to concentrate around the content-consistent views, which aligns with the notion that $\hat{Z}$ retains only content information. This implies that $p_{X|Z,\hat{Z}}\approx p_{X|\hat{Z}},$ and thus,
> > > $$TV(p_{X|Z,\hat{Z}}, p_{X|\hat{Z}})\rightarrow 0.$$
> > >
> > > **Similarly**, assuming the existence of **style-sharing views** (i.e., data samples that share the same style but differ in content), analogous to the content-sharing views used for multiview disentanglement in Theorem 4.2, we can establish:
> > > $$TV(p_{X|G,\hat{G}}, p_{X|\hat{G}})\rightarrow 0,$$
> > > showing that $\hat{G}$ asymptotically captures all style-relevant information in $X$. This proves a natural **information-theoretic interpretation** of disentanglement and editability in our framework, and strengthens the connection to deterministic analogues.
> > >
> > > We will include this connection and its assumptions in the revised discussion to help bridge the two perspectives more formally.
> > >
> > > **2. On clarifying experimental design**
> > >
> > > We acknowledge that our experimental section can be more tightly aligned with the theoretical claims.
> > >
> > > Regarding the reviewer’s concern about input assumptions in the MNIST experiment: it is true that the **input background color is always black**, but this arises from the **grayscale nature of the MNIST dataset**, not a limitation of our method. The model is not dependent on any specific background structure.
> > >
> > > In fact, our CIFAR-10 experiments already **demonstrate the model’s ability to disentangle content from style** (noise level or color shift) in the presence of **arbitrary and diverse background colors**, as CIFAR-10 contains complex natural scenes with varied color statistics. Moreover, we explicitly evaluate model performance across **multiple noise levels** by randomly varying the signal-to-noise ratio for each image, and test the model’s ability to reconstruct consistent content across these noise scales. These empirical findings provide empirical evidence for both **disentanglement** and **editability** under diverse style perturbations.
> > >
> > > We will emphasize this point in the revision and add additional visual examples to highlight the model’s robustness to diverse styles and its generalization capability beyond the MNIST setting To further improve the clarity and the connection with the theory part, we will:
> > >
> > > * Include a **summary table** mapping each experiment to the specific theorem on assumption it supports;
> > >
> > > * Clearly explain the **design rationale** behind synthetic, MNIST, and CIFAR-10 setups;
> > >
> > > * Clarify how Eq. (13) implements the theory from Eq. (11), and what approximations are made in practice.
> > > -----------------------------------------------
> > > Thank you again for your constructive critique. We are committed to substantially improving the clarity. We hope the improvements we outlined above address your concerns and reinforce the significance of our contributions.

---

> > > > ### Comment · Reviewer_34h9 · 2025-08-07
> > > >
> > > > Thank you for the follow-up. I now have a clearer understanding of the connection between theory and experiments, and my main concern regarding clarity has been largely addressed by the proposed revisions. While the practical applicability remains limited, I believe the theoretical contribution provides a novel and valuable perspective on disentanglement. I am therefore willing to raise my score to 4.

---

> > > > > ### Author Response · Authors · 2025-08-07
> > > > >
> > > > > Thank you very much for the follow-up and for raising the score. We are pleased to hear that the connections between our theory and experiments are now clearer, and the proposed revisions have addressed your concerns regarding clarity.
> > > > >
> > > > > We appreciate your acknowledgement of the theoretical contribution, and we agree that while the assumptions may limit intermediate practical applicability, the framework opens up new avenues for understanding and formalizing disentanglement in diffusion models. We will make sure to highlight these limitations and future directions more explicitly in the revised version.
> > > > >
> > > > > Thank you again for the thoughtful and constructive feedback.

---

### Official Review · Reviewer_cYrQ · 2025-07-02

**Clarity:** 3
**Significance:** 2
**Originality:** 2
**Rating:** 4
**Confidence:** 3

**Summary:**

This paper investigates the disentanglement ability of diffusion models. The authors provide key theoretical insights, showing that under certain conditions with partial supervision or multi-view inputs, diffusion models can disentangle content and style latent variables. Additionally, in the special case of independent subspaces, they introduce a novel score loss and prove global convergence results with finite sample . These theoretical contributions are supported by numerical experiments, validating the convergence rate under the independent subspace assumption. The authors further validate the general disentanglement theorems on two real-world image datasets and in a voice conversion (VC) adaptation task.

**Questions:**

1. Does Definition 4.3 imply that the observation X is a linear combination of latent variables Z and G? If so, wouldn’t that be a rather strong assumption for real-world datasets?

2. In the multi-view disentanglement setting, is it implicitly assumed that X and the concatenated latent variables (Z,G) have the same dimensionality? If so, could this constrain the applicability of the model?

3. I’m having trouble understanding how the example: "a speaker embedding extracted from a pre-trained speaker recognition model", which is used to illustrate a case where the style value is known. Could you explain more about how this is connected to "knowing the style function/value"? Is the speaker embedding considered the exact value of the style variable G? Or does using a pre-trained model imply that we fully know the style function, or are we just estimating it?

**Ethical Concerns:**

["NO or VERY MINOR ethics concerns only"]

**Final Justification:**

I am happy about understanding the cohesion among all the different settings in this paper.

**Limitations:**

Although the paper is theoretically solid and presents three key theorems, I find it challenging to build a clear logical connection or progression among them, particularly between the final theorem and the first two. It would be helpful if the authors could provide more explanation on how these results build upon one another or fit into a cohesive framework. Additionally, some of the assumptions made in the theoretical analysis are quite strong. I list them all in both the weakness and question parts and would appreciate further discussion from the authors on how these assumptions might be relaxed or addressed.

**Quality:**

3

**Strengths And Weaknesses:**

### Strengths

1.	Theoretical contributions are solid and well-formulated.

2.	The paper investigates an interesting problem, offering a novel perspective on disentanglement with diffusion models under certain supervision and assumptions.

3.	Even though the whole paper is notation-heavy, the writing style is accessible and visually engaging. The use of colored highlights and intuitive text boxes improves readability, and the examples in the main paper help understand key definitions.


### Weaknesses

1.	The connection between the multi-view setup and the independent subspace assumption is not clearly established for me. In my understanding, these are somewhat orthogonal concepts: independent subspaces do not necessarily imply a separation into content and style, and vice versa.

2.	Some of the assumptions made in this paper are quite strong and may not hold in real-world scenarios, potentially limiting the practical applicability of the theoretical findings. For instance, in the context of content disentanglement, the assumption that the style function is known is hard to justify for two main reasons:

(1) Precisely identifying which part of the data corresponds to the style or changing factors is overly idealized and rarely feasible in practice.

(2) Knowing the exact style value for all training samples is extremely challenging and unrealistic in most settings.

Additional assumptions that I find restrictive are listed in the question section.

---

> ### Author Rebuttal · Authors · 2025-07-30
>
> Dear reviewer cYrQ,
>
> Thank you for the thoughtful and constructive feedback. We address the concerns below:
>
> **1. Clarifying the connection between multi-view setup and ISM assumption**
>
> Thank you for this important observation. We agree that multi-view disentanglement (Theorem 4.2) and the independent subspace model (ISM) (Theorem 4.5) address distinct settings. The former analyzes general nonlinear, stochastic mappings under weak supervision, while the latter provides stronger guarantees in a special case with linear structure.
>
> Our intent is not to claim that ISM is a general case of multi-view disentanglement. Instead, we show that when the ISM structure holds, we can derive finite-sample convergence guarantees using gradient dynamics (Thm 4.5). The general framework (Defs. 3.1-3.4) encompasses both settings. We will make this relationship clearer in the revision by explicitly distinguishing the scope and assumptions of each theorem.
>
> **2. Assumption of known style function**
>
> This is a valid concern. The assumption of a known style function appears only in the “content disentanglement” setting (Def. 3.3, Thm. 4.1), where the goal is to **disentangle content given a fixed style** – a setting aligned with applications like **voice conversion** and **image editing**.
>
> In practice, we **approximate the style function using a pre-trained model** (e.g. speaker embeddings from ECAPA-TDNN, which achieves almost perfect speaker verification error rate [1]). While this is not a ground-truth oracle, it provides a **fixed and consistent representation of style** across training and inference, which makes the theoretical setup meaningful. This assumption mirrors common practice in the voice conversion literature and is not unique to our framework.
>
> We agree it is an idealization, but as a **weaker alternative to requiring full latent access**, it enables principled analysis under partial supervision. In real-world scenarios where even pre-trained models are unavailable, we propose the **multi-view formulation (Def. 3.4, Thm. 4.2)**,  which removes the need for known style and instead leverages paired observations. We will highlight the distinction more clearly.
>
> [1] Desplanques, B., Thienpondt, J., & Demuynck, K. “ECAPA-TDNN: Emphasized Channel Attention, Propagation and Aggregation in TDNN Based Speaker Verification”. Interspeech 2020.
>
> **3. Clarifying Definition 4.3 (independent subspace model)**
>
> Yes, Def. 4.3 assumes a linear mixing model to latent factors, which is a simplified case used to derive **finite-sample convergence guarantees**. As mentioned in the paper (Sec. 4.3), the ISM assumption is not required in general – our earlier results (Theorems 4.1 and 4.2) apply to **nonlinear, stochastic mappings** under mild Lipschitz assumptions. The ISM-based result (Thm 4.5) serves to complement these general results with **stronger guarantees** and clearer intuition about training dynamics under structure.
>
> Even though the ISM assumption does not always hold for raw data, such structures often emerge in the latent space of self-supervised representations. For instance, previous study [2] has found that self-supervised representation models have learned orthogonal subspaces for content (phonetic) and style (speaker) information, as mentioned in Section 4.3 of our paper. Further, practical algorithms making the ISM assumption have recently achieved competitive results in disentanglement tasks such as voice conversion [3] and image editing [4], demonstrating the applicability of the ISM assumption to complex data.
>
> [2] O. D. Liu, H. Tang, and S. Goldwater, “Self-supervised predictive coding models encode speaker and phonetic information in orthogonal subspaces,” in Interspeech, 2023.
>
> [3] Kamper, Herman, Benjamin van Niekerk, Julian Zaïdi, and Marc-André Carbonneau. “LinearVC: Linear Transformations of Self-Supervised Features through the Lens of Voice Conversion,” 2025.
>
> [4] L. Rout, N. Raoof, G. Daras, C. Caramanis, A. Dimakis, and S. Shakkottai, “Solving linear inverse problems provably via posterior sampling with latent diffusion models,” in NeurIPS, 2023.
>
> **4. Dimensionality assumptions in multi-view setting**
>
> We appreciate the opportunity for clarifying. No, we do **not require** that the observation $X\in \mathbb{R}^{d_X}$ has the same dimensionality as $(Z,G).$ Our theory allows for arbitrary encoder functions mapping inputs to latent spaces of lower dimension. For instance, the encoder can output $\hat{Z}\in \mathbb{R}^{d_Z},$ even if $d_Z<d_X.$ The score function operates in observation space, but the latent representations can reside in lower-dimensional subspaces. We will clarify this in the definitions and theorem statements.
>
> **5. Speaker embedding as known style function**
>
> We appreciate the opportunity to clarify. When we refer to using speaker embeddings (e.g., d-vectors), we treat the **pretrained embedding extractor** $g(\cdot)$ as a **proxy for the true style function**. This is standard in voice conversion pipelines and aligns with how weak supervision is commonly defined in practice. We never claim that this represents the true style variable $G$, but rather a fixed, known surrogate used during both training and inference, consistent with Def. 3.3. We will revise the text to make this distinction clearer and emphasize that our theory supports both fixed-style and fully-learned (multi-view) scenarios.
>
> **6. Logical progression across theorems**
>
> Thank you for pointing this out. The three theorems build progressively:
> Theorem 4.1 (Content disentanglement): proves approximate disentanglement under known style supervision.
> Theorem 4.2 (Multi-view disentanglement): proves approximate disentanglement under multi-view supervision with learned content and style encoders.
> Theorem 4.5 (ISM content disentanglement): focuses on linear latent structure and shows **finite-sample global convergence** with gradient descent.
> The first two theorems highlight **approximate disentanglement under weak supervision**. Theorem 4.5 is a special case of Theorem 4.1 when the data distribution has ISM structure, which allows us to study the training dynamic of disentanglement and prove stronger disentanglement guarantees. The special case of Theorem 4.2 with ISM is omitted since its proof is analogous to Theorem 4.5, though we provide a sketch of the proof strategy in Line 245-248.
>
> **7. Assumption discussion and relaxation**
>
> We agree and thank the reviewer for the suggestion. We already mention in Appendix B.7 and C.4 that some assumptions (e.g., exact independence between $Z$ and $G$, or i.i.d styles) can be **relaxed** to approximate independence settings.
>
> Further, under the content disentanglement setting, the assumption of access to the true style variable can be relaxed by assuming instead access to a **sufficiently good style estimator** $G’$, such as a reliable speaker embedding model in voice conversion. Such an estimator is essentially a perturbed version of the true style variable, and since the score function is assumed to be Lipschitz in our theory, it is robust against such perturbations and all theorems proved in our paper still hold, albeit with additional error terms due to the style estimation error.
>
> We will consolidate these discussions into the main text and include a new limitations and future directions paragraph addressing how assumptions might be relaxed in practice.
>
> ------------------------------
> We thank you again for the insightful comments and hope our response helps to address your concern and makes our paper more suited for the NeurIPS conference. Happy to discuss further and consider further revisions.

---

> > ### Comment · Reviewer_cYrQ · 2025-08-04
> >
> > Thank the authors for the detailed reply. I now have a clearer understanding of how all the sections are connected. While the settings are somewhat independent, they all revolve around the theme of disentanglement with diffusion models, which brings cohesion to the overall framework.
> >
> > I have one suggestion for further clarity: rather than listing the core assumptions within the definitions, it might be more effective to explicitly state them as formal assumptions, for example, Def 3.3 (Content disentanglement), as not all content-disentanglement work needs to know the style function.
> >
> > Considering this is a theoretically focused paper, I acknowledge that some of the assumptions, such as knowing the style function in the content-style theorem or the linearity assumption in ISM,  may limit immediate practical applicability. However, the theoretical contribution is novel.
> >
> > As a result, I decided to raise one point to my initial score.

---

### Official Review · Reviewer_qfQD · 2025-07-03

**Clarity:** 2
**Significance:** 2
**Originality:** 3
**Rating:** 4
**Confidence:** 3

**Summary:**

This paper proposes a theoretical framework for learning disentangled representations using conditional diffusion models under weak supervision conditions, such as partial labels and multiple views. While most prior work on identifiable representation learning has focused on deterministic, invertible mixing processes, diffusion models inherently involve stochastic, non-invertible mixing processes. To address this, the authors propose approximate disentanglement within conditional diffusion models under partial supervision or multi-view scenarios, supported by theoretical guarantees derived from Lipschitz continuity assumptions. The theoretical results are further validated through empirical experiments involving Gaussian mixture models, MNIST colorization, and CIFAR-10 image denoising, demonstrating effective disentanglement capabilities of the diffusion models.

**Questions:**

- In Equation (10), what is $U$ and $V$? They are not explicitly defined in the manuscript. As Equation (11) only optimizes for $\theta$ and not $U$ or $V$, it seems they are not learnable components. Are they some kind of pre-defined matrix?

- Could the authors clarify the meaning of $n$ in Equation (11)?

- In Line 235, the paper mentions that “The style guidance regularizer encourages style separation by reducing the mutual information between the content encoder and $X$.” However, it is unclear why score prediction using only style information (i.e., $Vs_G(g(x),t)-s^*(x,t)$) necessarily reduces mutual information between the content encoder and $X$. Could the authors clarify if it is possible for the content encoder to still capture excessive information about the style?

- Could the authors elaborate more on the role of balancing loss in Equation (11)? Although the full derivation and proof are presented in Appendix, it was personally hard to follow the full derivation. Could the authors provide intuition on how this term avoids poor local minima?

- In the experimental section, it is mentioned that Equation (13) is used for training on MNIST and CIFAR-10 as well. However, it is not clear how Equation (13) is derived from Equation (11). Firstly, it lacks balancing loss. Does it imply that balancing loss is a redundant term from the formulation? Additionally, since Equation (11) relies on the assumption of the independent subspace model (ISM), could the authors discuss the validity of applying Equation (13) to datasets like MNIST and CIFAR-10, where this assumption may not hold?

- Although multi-view disentanglement is highlighted as a core contribution, relevant experimental validation is missing. Could the authors provide empirical evidence demonstrating multi-view disentanglement?

- In image disentanglement experiment, could the authors provide comparison to related work (e.g., [1,2,3])? It would improve overall understanding if the authors provide comparison to previous work and provide analysis.

- A wrong image is inserted in Figure 6(d) in the Appendix.

[1] Chen et al., Weakly supervised disentanglement with guarantees, in AAAI 2020.

[2] Shu et al., Weakly supervised disentanglement with guarantees, in ICLR 2020.

[3] Locatello et al., Weakly supervised disentanglement without compromises, in ICML 2020.

**Ethical Concerns:**

["NO or VERY MINOR ethics concerns only"]

**Final Justification:**

While this work requires strong assumptions such as the ISM structure or style information for content-style disentanglement, which may limit its applicability to more general scenarios, the theoretical framework built on these assumptions is novel and insightful. Therefore, I am leaning towards weak accept.

**Limitations:**

yes, (although the authors do not allocate a section for indicating the limitation, it was written in the discussions throughout the paper)

**Paper Formatting Concerns:**

No major formatting issues were found.

**Quality:**

2

**Strengths And Weaknesses:**

**Strengths**

The paper is well-written and easy to follow. Formulating a definition of formal disentanglement for stochastic and non-invertible mappings of latent variables is a valuable contribution to the field. This paper introduces a clearly defined formulation of approximate disentanglement through the concepts of $(\epsilon,\nu)$-disentanglement and $\epsilon$-editability, addressing an important theoretical gap.


**Weaknesses**

A major concern is in the empirical justification of the proposed framework. Although the proposed theoretical foundation presented is concrete, the connection to empirical evidence is somewhat weak. For example, how does Equation (13) derived from Equation (11), why does it lack balancing loss, how can Equation (13) apply to datasets where it does not follows independent subspace model (ISM), lack of experiments on multi-view disentanglement, etc. Please refer to the Question Section below. Weak connection to the empirical study makes it challenging to assess the empirical validity and overall contribution of the proposed theoretical framework.

---

> ### Author Rebuttal · Authors · 2025-07-30
>
> Dear reviewer qfQD,
>
> We sincerely thank you for careful reading and for highlighting both the clarity of our theoretical contributions and the originality of our disentanglement framework. Below, we respond to the key concerns raised regarding the empirical connection, equation derivation, and experimental scope.
>
> **1. Empirical validation of theory and derivation of Eq. (13) from Eq. (11)**
>
> We appreciate the reviewer’s careful inspection. Eq. (13) is a practical implementation of Eq. (11) tailored to image domains (MNIST, CIFAR-10) where the true ISM structure is unknown or approximate. We drop the balancing loss in Eq. (13) for simplicity since empirically the model does not rely on the loss to escape bad local minima.
>
> Eq. (13) is equivalent to Eq. (11) without the balancing loss and when the input data is an ISM. The form in Eq. (13) looks different from Eq. (11) primarily because the (13) is using conditional score matching while (11) is using unconditional score matching. The two are equivalent as discussed in Section 2, and our theory can be easily adapted to conditional score matching as well, as discussed in line 250-251.
>
> Further, the style-guided regularization losses in (13) (denoted as $L_r’$) and (11) (denoted as $L_r$) are related. Suppose we assume ISM and restrict the function class to be a variant of the dual encoder score network in Fig. 2, i.e. $$s_{\theta}(X_t,\hat{Z},G,t)=Us_Z(X_t,\hat{Z},t)\mathbf{1}[\hat{Z}=\mathbf{0}_{d_Z}]+Vs_G(X_t,G,t),$$
> and suppose the all-zero vector is outside the support of the content estimator $\hat{Z}$, then we have
> $$L_r’=\mathbb{E}||Vs_G(X_t,G,t)+N_t/\sigma(t)||^2.$$
> This takes the same form as $L_r$ in (11), except in two aspects: i) this is conditional score matching; ii) $s_G$ is a score function conditioned on $G$ instead of an unconditional one in (11). We have already discussed i), and ii) poses no issue either since the score function conditioned on $G$ still satisfies the ISM assumption and can still be handled by our theory.
>
> **2. Role of balancing loss and intuition**
>
> Intuitively, the balancing loss tries to balance between the subspace matrices $(U,V)$ and the subspace score networks $(s_Z^{\theta_Z},s_G^{\theta_G})$ to have similar operator norms (by viewing the subspace score networks as operators). Otherwise, suppose $(s_Z^{\theta_Z},s_G^{\theta_G})$ have operator norms much larger than $(U,V)$, then $U \approx 0_{d_X\times d_Z}, V\approx 0_{d_X\times d_G},$ since their combined operator, the full score network $s_{\theta}$, has finite norm. Further, since the gradients $\nabla_{\theta_Z} L^{\lambda_r}$ and $\nabla_{\theta_G} L^{\lambda_r}$ scale with $U,V$, this implies $\nabla_{\theta_Z} L^{\lambda_r}\approx 0$ and $\nabla_{\theta_G} L^{\lambda_r}\approx 0,$ even though the learned content and style spaces $R(U)$ and $R(V)$ are nowhere near the true spaces $R(A_Z)$ and $R(A_G)$. This results in a bad local minimum.
>
> The balancing loss also does not affect the output of the optimal score network, since for any “imbalanced” optimal parameters $(U,V,\theta_Z,\theta_G)$, there always exist another $(U,V,\theta_Z’,\theta_G’)$ that achieves zero balancing loss. The argument based on singular value decomposition is provided in 1306-1309 of Appendix D.3.
>
> Therefore, the balancing loss helps the model to avoid bad local minima but does not change the location of at least one global minima for the regularized score matching loss. On the other hand, we empirically observe the model to converge near a global optima without the balancing loss, so we discard it for our experiments. The balancing loss is thus largely a proof artifact, and we leave analysis without such a loss for future works.
>
> **3. Multi-view disentanglement experiments**
>
> We appreciate the reviewer highlighting this. Our speech experiments (Sec. 5, Fig. 5) do instantiate multi-view disentanglement: different speakers provide style views, and the shared emotion across recordings is the content. This matches the Definition 3.4 setup. We acknowledge that this connection could be more explicitly stated and will revise the text to clarify this mapping between theory and practice. We are also actively developing additional vision-domain multi-view experiments.
>
> **4. Role of $U$ and $V$ in Eq. (10), and learnability**
>
> Thank you for pointing this out. U and V are learnable linear projections that align the outputs of the content and style encoders with the overall score function, motivated by Lemma 4.4. We adopt this architecture to reflect the ISM-based decomposability of scores into subspace components. Their gradients are computed and optimized during training (see gradient flow Eq. 12). We will update the paper to explicitly define U and V as learnable parameters.
>
> **5. Meaning of $n$ in Eq. (11)**
>
> Sure. $n$ is the number of samples in the training set. It is mentioned in the “Intuition” section in Line 243-244.
>
> **6. Intuition behind style guidance regularizer and mutual information**
>
> We agree this point deserves further clarification. The key idea is that **when the score network is forced to explain part of the data using only style**, the model is implicitly **discouraged from encoding style information redundantly in the content branch**. For general latent variable models, this is a heuristic regularization rather than a formal MI minimization, and we will add a clearer explanation and acknowledge.
>
> **For ISM, however, this regularization does lead to MI minimization.** To see this, consider the extreme case where no style guidance regularizer is used, then the content encoder can simply copy from the input X and maximize the score matching loss $L_0$ by letting $U$ covering the entire data space, and in this case the mutual information between the content encoder and X, $I(\hat{Z};X)=I(X_{t_0};X)$ is maximized; As style guidance weights $\lambda_r$ increase, the style encoder starts to play an increasingly important role in score prediction until it can predict the style component of the score perfectly. In turn, the content encoder becomes less and less predictive of the style component. Further, by Theorem 4.5, the subspaces $R(U)$ and $R(V)$ recover the content and style subspaces up to vanishingly small error for sufficiently large $\lambda_r$, and thus $I(\hat{Z};X)\approx I(Z_{t_0};X)<I(X_{t_0};X)$ by data processing inequality and that the true content encoder is injective. For a more precise relation between the style regularizer weight and $I(\hat{Z};X)$, one can upper bound $I(\hat{Z};X)$ in terms of subspace recovery errors using an argument similar to Appendix D.1.
>
> **7. Comparison to prior work in image experiments**
>
> We have made a comparison with related works in two additional disentanglement experiments on the CelebA image dataset. First, we perform **denoising** with the same setting as CIFAR 10 and compared our approach with two standard baselines, conditional GAN [1] and beta-VAE [2] adapted to the content disentanglement setting by conditioning on the observed style variable, which are representative of the approach taken in the papers suggested by the reviewer. As shown in the Table below, our method is superior to the baselines and follow the same trend with the style regularization weight as in our experiments on MNIST and CIFAR10.
>
> | **Method**                         | **Style Guidance Weight** | **LPIPS (↓)** | **SSIM (↑)** |
> |-----------------------------------|----------------------------|--------------------|---------------|
> | conditional GAN		| -		| 0.83 	| 0.16 	|
> | conditional betaVAE	| -		| 0.57	| 0.17	|
> | DDPM 			| 0		| 0.53	| 0.18	|
> | DDPM (ours)			| 0.3		| 0.47	| 0.19	|
> | DDPM (ours)			| 0.03		| 0.35	| 0.23	|
> | DDPM (ours)			| 3		| **0.33**	| **0.24**	|
>
> Further, we perform **face editing** with gender as the style variable. As shown in the table below, our method is superior to the conditional InfoGAN [1] and betaVAE [2] baseline in terms of style transfer (CLIP score) and content preservation (LPIPS score).
> | **Method** | **Style Guidance Weight** | **CLIP Score (↑)** | **LPIPS (↓)** |
> |----------------|----------------------------|--------------------|---------------|
> | conditional InfoGAN | -		| 0.241	  | 0.39		|
> | conditional betaVAE | -  | 0.254 |	0.37 |
> | DDPM | 0	 | 0.250 		| 0.35          |
> | DDPM (ours) | 3 	| 0.259              | **0.30**          |
> | DDPM (ours) | 6 	| 0.259              | 0.32          |
> | DDPM (ours) | 9	| **0.261**              | 0.38          |
>
> [1] Kim, Hyunjik, and Andriy Mnih. "Disentangling by factorising." International conference on machine learning. PMLR, 2018.
>
> [2] Chen, Xi, et al. "Infogan: Interpretable representation learning by information maximizing generative adversarial nets." Advances in neural information processing systems 29 (2016)
> Moreover, we plan to perform larger-scale experiments on more diverse datasets in the future.
>
> **8. Miscellaneous: wrong figure and limitations**
>
> Thank you for pointing it out. We acknowledge the formatting error in Fig. 6(d) and will fix it in the camera-ready version. We will also combine our scattered discussions of limitations into a dedicated section in the final version for clarity.
>
> --------------------------------------------------------------
>
> We thank you again for the insightful comments and hope our response helps to address your concern and makes our paper more suited for the NeurIPS conference. Happy to discuss further and consider further revisions.

---

> > ### Author Response · Authors · 2025-08-05
> >
> > Dear reviewer qfQD,
> >
> > We wanted to check in and see if you had any remaining concerns regarding our submission. We’ve aimed to address all points raised in the initial review with detailed clarifications and revisions, and we’d sincerely appreciate any feedback you might have.
> >
> > If there are outstanding questions or if anything remains unclear, we’d be happy to elaborate further. Thank you again for your time and thoughtful evaluation.

---

> > > ### Comment · Reviewer_qfQD · 2025-08-06
> > >
> > > Thank you for the detailed clarifications. Explanations have properly addressed my concerns. It is now clear how the empirical implementation of the objective (Eq. (13)) is connected to the theory-based objective (Eq. (11)), and how the overall parameters (e.g., U and V) are optimized. While this work requires strong assumptions such as the ISM structure or style information for content-style disentanglement, which may limit its applicability to more general scenarios, I agree with reviewer cYrQ that the theoretical framework built on these assumptions is novel and insightful. Therefore, I am raising my score to 4.

---

> > > > ### Author Response · Authors · 2025-08-06
> > > >
> > > > Thank you very much for the thoughtful engagement and for raising your score. We are delighted to hear that the connections between the theory-based objective Eq. (11) and the practical implementation Eq. (13) are now clear, as well as the optimization of key parameters like $U$ and $V$.
> > > >
> > > > We also appreciate your point about the strength of the assumption and availability of style information. While these are necessary for proving formal guarantees, we agree that it is important to communicate the scope and limitation of our framework. We will revise our manuscript accordingly to highlight these assumptions and discuss potential relaxations in future works.
> > > >
> > > > Thank you again for your constructive feedback and for recognizing the theoretical contribution.

---

### Official Review · Reviewer_KBNb · 2025-07-07

**Clarity:** 2
**Significance:** 1
**Originality:** 1
**Rating:** 3
**Confidence:** 4

**Summary:**

This paper establishes a theoretical framework for diffusion model based disentanglement under weak supervision. The authors address key challenges caused by the stochastic and non-invertible nature of DMs and propose two formal criteria for approximate disentanglement. They provide theoretical guarantees for both content-style disentanglement and multi-view disentanglement.  Experiments across synthetic, image, and speech data.

**Questions:**

Dose the method works on CelebA? may qualiative results and quantitative results compare with some recent methods [1,2] will make this work more solid.

[1] Ren, Xuanchi, et al. "Rethinking content and style: exploring bias for unsupervised disentanglement." Proceedings of the IEEE/CVF International Conference on Computer Vision. 2021.
[2] Wang, Zhizhong, Lei Zhao, and Wei Xing. "Stylediffusion: Controllable disentangled style transfer via diffusion models." Proceedings of the IEEE/CVF International Conference on Computer Vision. 2023.

**Ethical Concerns:**

["NO or VERY MINOR ethics concerns only"]

**Final Justification:**

Most of the problems have been solved except for the following two points. First, I don't know whether this method can really work on datasets like Celeba. For example, the authors should present some qualitative results of separating the angle and ID of the face. From the experimental results provided, it still doesn't compare with some more recent competitive methods. Second, I haven't seen the final version of the paper yet, so I don't know whether overclaiming and giving credit to other work have been properly resolved. I think I'll keep the rating for now.

**Limitations:**

See weakness.

**Quality:**

2

**Strengths And Weaknesses:**

Strengths:
- Well-Defined Metrics: Introduces new formal definitions for approximate disentanglement ((ε,ν)-disentanglement and ε-editability) suited to diffusion models.
- Well-structured theory​​: Clear progression from general identifiability (Sec 4.1–4.2) to specialized ISM convergence (Sec 4.3).

Weaknesses:
- The ISM assumption may not hold for complex data. Experiments are mostly small-scale and controlled; there’s a lack of large-scale, diverse benchmarks that would test generalization in more realistic scenarios.
- Limited novelty: The idea of employing mutual information-based regularization to promote disentanglement in diffusion models has been previously explored, notably in DisDiff [1].
- The work lacks empirical comparisons against existing content-style disentanglement approaches, making it difficult to assess the relative effectiveness of the proposed method.
- The title may misslead the readers, since the paper mostly focuses on content-style disentanglement, which is different from the general disentanglement [1,2,3]

[1]  Yang, Tao, et al. "DisDiff: Unsupervised Disentanglement of Diffusion Probabilistic Models." Advances in Neural Information Processing Systems 36 (2023).

[2] Kim, Hyunjik, and Andriy Mnih. "Disentangling by factorising." International conference on machine learning. PMLR, 2018.

[3] Chen, Xi, et al. "Infogan: Interpretable representation learning by information maximizing generative adversarial nets." Advances in neural information processing systems 29 (2016)

---

> ### Author Rebuttal · Authors · 2025-07-30
>
> Dear reviewer KBNb,
>
> Thank you for the thoughtful feedback and for acknowledging the strengths of our paper. Below, we address the main concerns raised:
>
> **1. On the ISM assumption and generalization to complex data**
>
> We agree that ISM represents a simplified model; however, we emphasize that ISM is only one part of our broader theoretical framework. Sections 4.1 and 4.2 provide results under general **nonlinear, non-invertible** mappings with only subgaussian and Lipschitz assumptions (see Theorems 4.1 and 4.2). ISM is used in Section 4.3 specifically to establish stronger guarantees such as **finite-sample global convergence**, which are rare in existing literature. The theoretical contributions are general.
>
> Further, even though the ISM assumption does not always hold for raw data, such structures often emerge in the latent space of self-supervised representations. For instance, previous study [1] has found that self-supervised representation models have learned orthogonal subspaces for content (phonetic) and style (speaker) information, as mentioned in Section 4.3 of our paper. Further, practical algorithms making the ISM assumption have recently achieved competitive results in disentanglement tasks such as voice conversion [2] and image editing [3], demonstrating the applicability of the ISM assumption to complex data.
>
> [1] O. D. Liu, H. Tang, and S. Goldwater, “Self-supervised predictive coding models encode speaker and phonetic information in orthogonal subspaces,” in Interspeech, 2023.
>
> [2] Kamper, Herman, Benjamin van Niekerk, Julian Zaïdi, and Marc-André Carbonneau. “LinearVC: Linear Transformations of Self-Supervised Features through the Lens of Voice Conversion,” 2025.
>
> [3] L. Rout, N. Raoof, G. Daras, C. Caramanis, A. Dimakis, and S. Shakkottai, “Solving linear inverse problems provably via posterior sampling with latent diffusion models,” in NeurIPS, 2023.
>
> **2. On additional experiments (e.g. CelebA)**
>
> The main contribution of the paper is to develop a theoretical framework for general diffusion model-based content-style disentanglement, and our experiments are intentionally structured to validate different settings (content, multi-view, and ISM) in a controlled yet representative manner. As noted in the paper (Section 5 and Appendix E.3), we also include real-world evaluations on speech tasks (e.g., IEMOCAP) using voice conversion and emotion recognition to demonstrate applicability beyond synthetic data.
>
> We have further verified our theory on two more disentanglement tasks on the CelebA image dataset. First, we perform **denoising** with the same setting as CIFAR 10 and compared our approach with two standard baselines, infoGAN [4] and beta-VAE [5] adapted to the content disentanglement setting by conditioning on the observed style variable, which are representative of the approach taken in the papers suggested by the reviewer. As shown in the Table below, we observe the same trend as our previous experiments.
> | **Method**                         | **Style Guidance Weight** | **LPIPS (↓)** | **SSIM (↑)** |
> |-----------------------------------|----------------------------|--------------------|---------------|
> | conditional InfoGAN		| -		| 0.83 	| 0.16 	|
> | conditional beta VAE	| -		| 0.57	| 0.17	|
> | DDPM 			| 0		| 0.53	| 0.18	|
> | DDPM (ours)			| 0.3		| 0.47	| 0.19	|
> | DDPM (ours)			| 0.03		| 0.35	| 0.23	|
> | DDPM (ours)			| 3		| **0.33**	| **0.24**	|
>
> Further, we perform **face editing** with gender as the style variable. As shown in the table below, our method is superior to the conditional GAN baseline in terms of style transfer (CLIP score) and content preservation (LPIPS score).
> | **Method** | **Style Guidance Weight** | **CLIP Score (↑)** | **LPIPS (↓)** |
> |----------------|----------------------------|--------------------|---------------|
> | conditional InfoGAN|-		| 0.241	  | 0.39		|
> | conditional betaVAE | -  | 0.254 |	0.37 |
> | DDPM | 0	 | 0.250 		| 0.35          |
> | DDPM (ours) | 3 	| 0.259              | **0.30**          |
> | DDPM (ours) | 6 	| 0.259              | 0.32          |
> | DDPM (ours) | 9	| **0.261**              | 0.38          |
>
> Moreover, we plan to perform larger-scale experiments on more diverse datasets in the future.
>
> [4] Kim, Hyunjik, and Andriy Mnih. "Disentangling by factorising." International conference on machine learning. PMLR, 2018.
>
> [5] Chen, Xi, et al. "Infogan: Interpretable representation learning by information maximizing generative adversarial nets." Advances in neural information processing systems 29 (2016)
>
> **3. On empirical comparisons with content-style disentanglement methods**
>
> We appreciate this suggestion and agree that comparisons with existing baselines would strengthen the empirical section. However, as our primary focus is theoretical, our experiments are designed to validate the **theoretical predictions** (e.g., convergence trends, editability, MI bounds) in a controlled manner.
>
> That said, we **do include comparisons** with several baselines in the speech domain (Fig. 5), such as pitch shifting, TriAAN-VC, and nearest-neighbor VC methods. In our response to 2, we also compare our method with two standard baselines for disentanglement, conditional InfoGAN and beta-VAE. For image data, direct comparisons with methods such as StyleDiffusion [Wang et al, ICCV 2023] are non-trivial due to differences in task setup and model supervision, but we plan to incorporate these in future versions.
>
> **4. On novelty compared to DisDiff and mutual information-based methods**
>
> We respectfully disagree that our contribution lacks novelty. While DisDiff [Yang et al., 2023] introduces MI-based regularization for DMs, it is a **fully empirical** work. Our work makes the **first theoretical contributions** toward **identifiability and convergence guarantees** of DM disentanglement under stochastic, non-invertible mappings – challenges not addressed in DisDiff. Our formulation of **$(\epsilon,\nu)$-disentanglement** and **$\epsilon$-editability** (Defs. 3.1-3.2), and the theoretical results (Theorems 4.1-4.5) including finite-sample convergence, are novel and unique to our paper.
>
> Moreover, our use of MI regularization is not merely adopted from prior work but is grounded in our theoretical derivation to achieve approximate identifiability. These elements distinguish our work both in terms of scope and depth.
>
> **5. On title scope**
>
> We will change the title to “Can diffusion models disentangle content from style? A theoretical perspective” to limit the scope of our theory to content-style disentanglement.
>
> ---------------------------------------------------------------
>
> We thank you again for the insightful comments and hope our response helps to address your concern and makes our paper more suited for the NeurIPS conference. Happy to discuss further and consider further revisions.

---

> ### Comment · Reviewer_KBNb · 2025-08-05
> **Response to authers**
>
> Thank you for the detailed response, I think the comment that "DisDiff is fully empirical work"is not rigorous, DisDiff presents a therom of disentangling for diffusion model and gives a proof on the therom in appendix. Also I think " first theoretical contributions" is not a rigorous claim for this paper.
>
> Considering that my concerns are still not fully solved, I will keep my rating in this period.

---

> ### Author Response · Authors · 2025-08-05
>
> Thank you again for the clarification.
>
> You are absolutely right that DisDiff includes a theoretical result, and we appreciate the opportunity to clarify the distinction. The main theorem in DisDiff [Yang et al., 2023] demonstrates that their proposed regularization objective upper bounds the mutual information between content and style. However, this result does not provide identifiability or convergence guarantees for disentanglement within the diffusion process itself. In fact, the authors themselves acknowledge this limitation in Section 4.3 of their paper:
> > Given the **difficulty in identifying sufficient conditions for disentanglement** in an unsupervised setting, we follow the previous works in disentangled representation literature, which propose necessary conditions that are effective **in practice** for achieving disentanglement.
>
> As we demonstrate in **Example 1** of our paper, **minimizing mutual information between latent variables alone does not guarantee disentanglement**, even in idealized settings. This underscores the need for **supervision (e.g., labels or views)** and **structural assumptions (e.g., ISM)** to ensure recovery, as provided in our theoretical framework.
>
> By contrast, our theory introduces an **information-theoretic formalization** of approximate disentanglement and editability under **weak supervision settings** (e.g., partial labels and multiple views), and proves global convergence guarantees under the ISM assumption. We agree that the phrasing “first theoretical contribution” was too strong and will revise it in the final version to reflect the **specific novelty in our identifiability guarantees under weak supervision**.
>
> We believe this distinction highlights an important shift from **empirically motivated regularization** to **theoretically grounded disentanglement objectives** in the context of diffusion models.

---

### Note · Authors · 2025-08-12

**Opening**

We are grateful to the reviewers for their insightful feedback, which has led to a substantially improved manuscript. We are pleased that three of the four reviewers have raised their scores, recognizing the paper’s enhanced theoretical contribution, improved clarity, and stronger connection between theory and experiments. We would also like to thank Reviewer KBNb for the candid comments.

**Core Contributions**

We present a novel information-theoretic framework for achieving approximate content-style disentanglement in diffusion models (DMs). Our framework is specifically designed for the challenging setting of **weakly supervised DM-based disentanglement**, where the inherent **stochasticity** and **non-invertibility** invalidate classical deterministic definitions of disentanglement.
* We introduce $(\epsilon,\nu)$-disentanglement and $\epsilon$-editability, disentanglement criteria tailored to DMs;
* We prove identifiability in three settings: (a) known-style supervision, (b) multi-view without style knowledge, and (c) the ISM special case with stronger finite-sample global convergence guarantees.

**Major Points Addressed**

* *Relation to prior works (DisDiff, Yang et al. 2023):* While DisDiff (Yang et al. 2023) shows its regularizer acts as a bound on mutual information, our work establishes something fundamentally stronger: the first formal **identifiability guarantees** and **convergence results** for disentanglement within the DM framework.

* *Empirical validation:* To demonstrate the practical utility of our theory, we have incorporated additional experiments on CelebA for denoising and face-editing tasks. We now include direct comparisons with representative baselines such as conditional InfoGAN and $\beta$-VAE, observing consistent improvements in LPIPS/CLIP scores as style-guidance increases.

**Conclusion**

We believe the revised manuscript, strengthened with theoretical clarifications and added empirical comparisons, presents a rigorous and well-scoped contribution to disentanglement theory for diffusion models. We thank the reviewers once again for recognizing the theoretical contribution and for helping us to improve our work.

---

### Decision · Program_Chairs · 2025-09-17

**Decision:**

Accept (poster)

**Comment:**

This paper presents a novel theoretical framework for understanding disentanglement in diffusion models under weak supervision. The authors introduce two key criteria: (1) approximate information-theoretic disentanglement and (2) editability. They demonstrate identifiability results in three scenarios: known-style supervision, multi-view data, and independent subspace models. In these cases, they establish finite-sample global convergence guarantees. Beyond the theoretical contributions, the authors validate the framework through controlled experiments using synthetic, image, and speech data. The results show that training strategies inspired by their analysis can enhance disentanglement performance in practice.

Most reviewers had a positive view of the paper, and the discussion period was lively. The authors adequately addressed most of the concerns raised by the reviewers, including additional comparisons with Celeb A and a clarification regarding the writing relationship to prior work (DisDiff). The paper's focus is on the theoretical framework, and while its performance on Celeb A may not fully match the leading methods for this task, I still consider this work valuable. Overall, the paper presents interesting new theories and empirical validation, and I recommend its acceptance. The comments provided in the rebuttal should be integrated to enhance the manuscript's readability.